# How Transformers Represent Hierarchies: A Local-to-Global Mechanism

Zhiling Zhou [1]   Tianhao Wang [2]   Zhuoran Yang [1]

## Abstract

Large language models built on autoregressive Transformers excel at next-token prediction, but it is unclear how their internal computations capture the latent hierarchical dependencies that often underlie language. We study this question in a controlled formal-language setting based on probabilistic context-free grammars (PCFGs), where sequences are generated by a latent hierarchical process. Empirically, standard autoregressive Transformers can be trained to accurately match the grammar-induced next-token distribution. Using probing analyses, we find that Transformer hidden states contain information used by classical parsing algorithms. Moreover, this information emerges through a layer-wise progression, revealing a local-to-global mechanism: early layers accumulate local patterns, while later layers aggregate them into a compact summary for next-token prediction. Complementing these empirical findings, we provide an explicit construction of Transformers that can parse binary PCFGs with depth *logarithmic* in the grammar's sequence length. Surprisingly, trained Transformers in this setting exhibit prediction behavior and internal representations that closely mirror our construction. Together, our results offer a mechanistic account of how Transformers integrate hierarchical parsing with autoregressive generation, enabling them to closely approximate the grammar-induced next-token distribution. Code is available at our GitHub repository.

## 1. Introduction

The unprecedented capabilities of Transformer-based LLMs across a wide range of downstream tasks are driven in large part by large-scale generative pretraining on massive text corpora (Vaswani et al., 2017; Radford et al., 2019). At inference time, they generate text left-to-right by repeatedly mapping a prefix to a distribution over the next token using a fixed stack of attention and MLP blocks. Despite this success, we still do not have a concrete understanding of what information is represented in the hidden states and how it is updated across layers to produce accurate next-token probabilities. This gap is especially salient for hierarchical structures in language: many regularities are naturally described in terms of constituent-level organization rather than surface proximity. While mechanistic interpretability has begun to identify meaningful features and circuits in trained Transformers (Wang et al., 2022; Nanda et al., 2023; Wang et al., 2024; Chen et al., 2025), we still do not have a clear account of how hierarchical syntactic information is computed and maintained during autoregressive generation.

To study representations of hierarchies in a controlled and analyzable setting, we focus on context-free grammars (CFGs) (Chomsky, 1956). CFGs provide a clean abstraction of hierarchical structures that enables precise algorithmic analysis. We consider autoregressive Transformers trained on sequences generated by probabilistic CFGs (PCFGs). In this setting, the model must assign correct next-token probabilities conditioned on a prefix, so success requires maintaining an appropriate prefix-conditioned summary of hierarchical constraints throughout generation.

Prior work has shown that Transformers can succeed on a range of CFG-related tasks, but existing evaluations often leave the underlying internal computation underdetermined. Some studies focus on recognition, deciding whether a full string belongs to a formal language (Ebrahimi et al., 2020; Hahn, 2020; Weiss et al., 2021; Delétang et al., 2022; Jerad et al., 2026). There are also studies of masked language modeling objectives that require structural inference from partially observed sequences (Zhao et al., 2023; Garnier-Brun et al., 2024). In autoregressive settings, constructive results show that attention-based architectures can generate restricted context-free languages under suitable assumptions (Yao et al., 2021), yet correct generation alone does not reveal what structured intermediate states the model learns to track, and it need not correspond to a canonical parsing computation (Wen et al., 2023). More recent work provides suggestive evidence of structured representations for richer grammar families, for example depth-bounded

---

[1]Yale University, New Haven, CT, USA [2]University of California, San Diego, La Jolla, CA, USA. Correspondence to: Zhiling Zhou <zhiling.zhou@yale.edu>.

PCFGs (Allen-Zhu & Li, 2023), and observations of parallel learning of subgrammars (Schulz et al., 2025). Complementary studies examine hierarchical language learning through last-token prediction, emphasizing random-hierarchy settings, representation dynamics, and scaling behavior during training (Cagnetta et al., 2024; Cagnetta & Wyart, 2024; Cagnetta et al., 2025). Our focus is instead on fully autoregressive generation at every prefix and on characterizing the algorithmic computation learned by the trained model. A remaining gap is to characterize what prefix-conditioned information is present in Transformers' hidden states that is sufficient for correct next-token prediction, and how that information is organized across layers.

In this work, we address this gap by explicitly connecting trained Transformers to prefix-conditioned parsing. Our empirical analyses are based on autoregressive Transformers trained on a class of depth-bounded PCFGs. After training, the resulting Transformers closely approximate the next-token distribution induced by the grammar. We interpret the resulting representations through the lens of classical chart parsing. Bottom-up chart algorithms such as CYK (Kasami, 1966; Younger, 1967; Cocke, 1969) and inside-style dynamic programs (Lari & Young, 1990) build larger constituents from smaller ones, while incremental algorithms such as Earley parsing maintain prefix-conditioned chart states that support prefix probabilities (Earley, 1970; Stolcke, 1995; Jelinek & Lafferty, 1991). Using probing-based analyses, we first test whether PCFG-trained autoregressive Transformer representations encode the information needed to reconstruct prefix-conditioned chart structure. We find that later-layer hidden states contain sufficient information, closely related to Earley-style chart states, for a classical incremental parser to continue parsing and generating from a prefix, even though the Transformer is not explicitly executing Earley operations step by step.

These results raise a natural mechanistic question: how can an Earley-like prefix summary arise from a single forward pass in a shallow stack of layers? The Earley algorithm processes prefixes sequentially: the parse state after $n$ tokens is obtained through $O(n)$ sequential updates, with each update potentially retrieving information from the prefix history. In contrast, the depth of our Transformers scales with the grammar's derivation depth, not with sequence length. Because the grammar family we consider generates sequences that grow exponentially with derivation depth, even a shallow Transformer must handle sequences far longer than its layer count would permit for step-by-step simulation. This rules out naive sequential emulation and suggests that Transformers represent hierarchies via a local-to-global mechanism across layers.

To test this hypothesis, we then probe for bottom-up structure in the same trained models. We recover CYK-like

dynamic programming signals from intermediate representations and observe a consistent local-to-global pattern across layers: evidence about short spans and local constituents is accessible in earlier layers, and deeper layers progressively compose this information into summaries corresponding to larger spans and global syntactic structure. This provides a concrete mechanistic picture that complements the Earley-style result. Local span-level evidence is accumulated and composed across layers, and the resulting global summary is rich enough to support accurate next-token prediction, including the recovery of Earley-like chart states.

Finally, we complement these empirical findings with theory. We provide a theoretical construction of a Transformer that implements local-to-global parsing for a canonical class of hierarchical grammars, namely balanced full-binary PCFGs, and produces the corresponding next-token distributions. We also train Transformers on this full-binary grammar and find that the learned parameters exhibit striking agreement with the construction. In particular, analyses of attention maps reveal patterns consistent with the predicted hierarchical information flow. This links the proposed local-to-global mechanism to both an explicit algorithmic realization and the behavior of trained models.

**Main contributions**. Our main contributions are as follows:

- We analyze autoregressive Transformers trained on a depth-bounded PCFG family and show via probing that hidden states encode sufficient information, closely related to Earley-style prefix chart states, for a classical incremental parser to continue parsing and generating from a prefix.
- We show that small Transformers can model this PCFG family even when string length can be exponential in derivation depth, and we argue that the recovered Earley-like information is unlikely to arise from sequentially simulating Earley operations within a forward pass.
- We recover CYK-like bottom-up dynamic programming signals from hidden states and identify a local-to-global mechanism across layers in which local span evidence is composed into global syntactic structure that supports autoregressive prediction.
- We provide a theoretical construction of a Transformer that implements local-to-global parsing for a balanced full-binary PCFG, and we empirically show that trained models exhibit attention patterns that qualitatively match the theoretical construction.

## 2. Preliminaries

### 2.1. Probabilistic Context-Free Grammar

Probabilistic context-free grammars (PCFGs) (Booth & Thompson, 2006) provide a canonical model for language hierarchies. Specifically, we consider a class of depth-

bounded PCFGs defined as follows.

**Definition 2.1** (Depth-bounded PCFG). A finite-depth probabilistic context-free grammar (PCFG) is a tuple $\mathcal{G} = (\mathcal{N}, \Sigma, \mathcal{R}, S, p)$, where $\mathcal{N}$ is a finite set of *nonterminal* symbols, $\Sigma$ is a finite set of *terminal* symbols, and $S \in \mathcal{N}$ is the start symbol. The depth bound is specified by an integer $K \geq 1$ together with a disjoint level decomposition $\mathcal{N} = \sqcup_{k=1}^{K} \mathcal{N}_k$ with $\mathcal{N}_0 = \Sigma$ and $\mathcal{N}_K = \{S\}$. The *production rule* set $\mathcal{R}$ is finite and consists of rules $r : A \to \gamma$ such that $A \in \mathcal{N}_k, \gamma \in (\mathcal{N}_{k-1})^*$, for some $k \in \{1, \ldots, K\}$. Finally, $p : \mathcal{R} \to [0, 1]$ assigns a probability to each rule, normalized so that for every nonterminal $A \in \mathcal{N}$, the probabilities of all rules expanding $A$ sum to one: $\sum_{r \in \mathcal{R}_A} p(r) = 1$, where $\mathcal{R}_A = \{r \in \mathcal{R} : r \text{ has left-hand side } A\}$.

Under this layered formulation, derivations are organized as finite trees with a fixed maximum height, making the hierarchical structure of constituent composition explicit. In our experiments, we sample random grammars from this family with depth 4 and terminal alphabet $\{1, 2, 3\}$. Figure 1 presents CFG-1, one sampled example grammar; additional sampled examples (CFG-2 through CFG-5) are provided in Appendix C.1. These grammars define a vast combinatorial space, where the potential language size scales double-exponentially with the grammar's depth. For instance, the number of possible terminal sequences generated by CFG-1 is upper-bounded by $2^{48}$.

Closely related grammar formalisms have been studied in prior work, including Allen-Zhu & Li (2023); Cagnetta et al. (2024); Cagnetta & Wyart (2024); Favero et al. (2025). These settings also avoid potential shortcuts in which the current depth can be recovered by simple bracket-counting, as in the bounded Dyck languages considered in (Yao et al., 2021; Wen et al., 2023).

## 2.2. Classical Parsing Algorithms for PCFGs

Parsing algorithms provide procedures for constructing the corresponding latent hierarchies (derivation trees) from observed strings. Below, we discuss two classical approaches—CYK/inside and Earley—that will serve as interpretive lenses for understanding Transformer computations. Details and examples are provided in Appendix B.

**Bottom-Up Parsing (CYK / Inside)**. The CYK algorithm (Cocke, 1969; Younger, 1967; Kasami, 1966) and the closely related inside algorithm (Lari & Young, 1990) use dynamic programming to build parse trees from the bottom up: they first identify which nonterminals can generate each individual token, then determine which nonterminals can generate each pair of adjacent spans, then triples, and so on. This process is inherently parallelizable—larger spans depend only on smaller spans, with no backward dependencies. Formally, let $\alpha_{i,j}^A$ denote the probability that nonterminal $A$ derives

Layer 4 $\{ 16 \to 15\,13 \mid 13\,14\,15 \mid 13\,15\,14$

Layer 3 $\begin{cases} 13 \to 11\,11 \mid 11\,12\,11 \mid 11\,11\,10 \\ 14 \to 12\,11\,11 \mid 10\,12\,10 \mid 12\,12\,10 \mid 10\,11\,11 \\ 15 \to 12\,11 \mid 12\,12\,10 \end{cases}$

Layer 2 $\begin{cases} 10 \to 9\,9 \mid 7\,8 \mid 7\,7 \mid 8\,8 \\ 11 \to 7\,8\,7 \mid 7\,9\,7 \mid 9\,8 \mid 8\,7\,8 \\ 12 \to 9\,9 \mid 8\,7\,8 \mid 7\,9 \end{cases}$

Layer 1 $\begin{cases} 7 \to \text{'2'}\,\text{'1'}\,\text{'2'} \mid \text{'2'}\,\text{'2'}\,\text{'2'} \mid \text{'2'}\,\text{'3'} \\ 8 \to \text{'3'}\,\text{'3'} \mid \text{'3'}\,\text{'2'}\,\text{'1'} \mid \text{'1'}\,\text{'3'} \\ 9 \to \text{'1'}\,\text{'1'} \mid \text{'2'}\,\text{'3'}\,\text{'3'} \mid \text{'3'}\,\text{'2'} \mid \text{'1'}\,\text{'3'} \end{cases}$

*Figure 1.* CFG-1, one sampled grammar from our random layered PCFG family with depth 4, where nonterminals are partitioned into four layers (annotated on the left) and each production expands symbols from one layer to the immediately lower layer. Each line corresponds to a production rule, where the left-hand side expands to one of several candidate right-hand sides, separated by |. The terminal alphabet is $\{1, 2, 3\}$.

the substring $x_{i:j}$ (the *inside probability*). The recursion combines adjacent subspans:

$$\alpha_{i,j}^A = \sum_{A \to BC \in \mathcal{R}} p(A \to BC) \sum_{k=i}^{j-1} \alpha_{i,k}^B \, \alpha_{k+1,j}^C, \quad (1)$$

with initialization $\alpha_{i,i}^A = p(A \to x_i)$ for terminal rules. A nonzero entry $\alpha_{i,j}^A > 0$ indicates that $A$ can derive the substring $x_{i:j}$, which we call a *complete constituent*. However, CYK only tells us *what* constituents exist—it does not track where we are in the parsing process or what tokens might come next.

**Left-to-right Parsing (Earley)**. The Earley algorithm (Earley, 1970; Stolcke, 1995) reads the input left-to-right, tracking all possible partial parses that are consistent with the prefix seen so far. At each position, it maintains a set of *Earley states* that record which production rules are in progress and how far each has been recognized. This allows Earley to know not just what constituents exist, but *where we are* in the overall parse—and therefore what tokens can legally come next. However, this capability comes at a cost: the COMPLETE operation, which incorporates finished constituents into waiting partial parses, requires looking back through the entire prefix history. This makes Earley parsing inherently sequential.

**The key contrast**. CYK and Earley offer complementary strengths. CYK efficiently identifies *what* constituents exist via parallelizable bottom-up computation, but does not track parsing progress. Earley tracks *where* we are in the parse and can predict next tokens, but requires sequential processing. As we will see, Transformers appear to combine the best of both: early layers perform CYK-style bottom-up aggregation to progressively build constituent information, while the final layer produces Earley-style representations sufficient for prefix-conditioned next-token prediction.

## 2.3. Transformers and Probing

We train decoder-only Transformers on sequences generated by PCFGs, and study their internal representations through a suite of probing analyses.

**Transformer.** Let $\mathbf{x} = (x_1, \ldots, x_t)$ be a sequence of tokens from vocabulary $\mathcal{V}$. Input tokens are mapped to hidden states $H^{(0)} \in \mathbb{R}^{t \times d}$ via embedding matrix $E \in \mathbb{R}^{|\mathcal{V}| \times d}$ and positional encodings $P \in \mathbb{R}^{t \times d}$, where $H_i^{(0)} = E_{x_i} + P_i$. The Transformer consists of $L$ layers, each performing:

$$\tilde{H}^{(l)} = \text{LN}\big(H^{(l-1)} + \text{MHA}(H^{(l-1)})\big),$$
$$H^{(l)} = \text{LN}\big(\tilde{H}^{(l)} + \text{FFN}(\tilde{H}^{(l)})\big).$$

The Multi-Head Attention (MHA) with $A$ heads is $\text{MHA}(H) = [\text{head}_1, \ldots, \text{head}_A]W^O$, with each head $k$ computed as $\text{head}_k = \text{softmax}((HW_k^Q)(HW_k^K)^T/\sqrt{d_k} + M)(HW_k^V)$, where $W_k^{Q,K,V} \in \mathbb{R}^{d \times d_k}$, $d_k = d/A$, and $M$ is the causal mask ($M_{ij} = -\infty$ for $i < j$). The FFN is defined as $\text{FFN}(h) = \sigma(hW_1 + b_1)W_2 + b_2$ with $W_1 \in \mathbb{R}^{d \times 4d}$ and $W_2 \in \mathbb{R}^{4d \times d}$. Finally, the next-token distribution is $P(x_{t+1} \mid x_{1:t}) = \text{softmax}(H_t^{(L)}W_U)$, where $W_U \in \mathbb{R}^{d \times |\mathcal{V}|}$ is the readout matrix.

**Probing.** To examine the internal mechanisms of Transformers, we utilize probing—a diagnostic technique designed to identify whether specific structural properties are encoded within a model's latent representations (Alain & Bengio, 2016; Adi et al., 2016; Hewitt & Manning, 2019; Tenney et al., 2019). Formally, let $h_i^{(l)}$ denote the hidden state at layer $l$ and position $i$. A probe is defined as a classifier $f_\theta$ that maps the hidden state to a target property $z_i$, i.e., $\hat{z}_i = f_\theta(h_i^{(l)})$. During probing, the pre-trained Transformer parameters remain frozen; we pass input sequences through the model to extract hidden states $h_i^{(l)}$, which, paired with their corresponding labels $z_i$, form the probing dataset. Since our target involves a multi-label classification task (where multiple Earley items may be active at a single position), we employ a Multi-Layer Perceptron (MLP) as the probing architecture to capture potential nonlinear dependencies.

# 3. Layer-Wise Parsing Representations in PCFG-Trained Transformers

In this section, we conduct controlled experiments to examine whether Transformers trained for autoregressive next-token prediction on finite-depth PCFGs exhibit internal representations consistent with these parsing computations.

## 3.1. Training Autoregressive Transformers on PCFGs

**Model architecture.** For each grammar, we train a GPT-style decoder-only Transformer with 5 layers, 4 attention heads per layer, and a hidden dimension of 512. The vocabulary consists of the terminal symbols $\{1, 2, 3\}$ together with the special tokens $[\texttt{BOS}]$, $[\texttt{EOS}]$, and $[\texttt{PAD}]$, all of which are represented using trainable token embeddings. Given a sequence $x_{1:T}$ sampled from a PCFG, we prepend and append boundary tokens $x_0 = [\texttt{BOS}]$ and $x_{T+1} = [\texttt{EOS}]$, respectively. At each position $t$, the model predicts the next token conditioned on the prefix $x_{1:t}$, inducing a distribution $p_\theta(x_{t+1} \mid x_{1:t})$. Compared to prior work such as Allen-Zhu & Li (2023), which trains a 12-layer, 12-head Transformer with hidden dimension 768 on sequences generated from a deeper grammar, our model depth is intentionally chosen to be comparable to, and not substantially larger than, the depth of the underlying grammars.

**Training configuration.** We consider a family of randomly generated finite-depth layered PCFGs of height 4 (Section 2), in which each nonterminal expands to 2-4 uniformly weighted rules of length 2 or 3 over the immediately lower layer. Figure 1 shows one sampled example, and Appendix C.1 gives the full sampling procedure and additional grammars. For the probing experiments below, we sample 50 grammars from this family and generate sequences by recursively expanding the start symbol until only terminals remain. For each grammar, we train a separate model with cross-entropy loss and AdamW (Loshchilov & Hutter, 2017) for 80,000 steps, using a fresh mini-batch of 256 independently sampled sequences at each step; further details are deferred to Appendix C.2.

**Evaluation.** We evaluate whether a trained Transformer matches the grammar-induced next-token distribution, computed via the probabilistic Earley algorithm, using held-out sequences sampled from the grammar. As shown in Figure 2, we visualize the next-token distributions induced by the grammar and by the Transformer on a representative sequence from $\texttt{CFG-1}$, one sampled example grammar from the 50-grammar pool. The two distributions exhibit highly similar structures. We further assess generative correctness by sampling 20,000 sequences from each trained Transformer and measuring the fraction that satisfy the corresponding grammar. Across the sampled grammars, the trained models achieve accuracies exceeding 99%. Quantitative comparisons based on the KL divergence between the grammar-induced and Transformer-predicted next-token distributions are reported in Appendix C.3.

## 3.2. Probing Earley Prefix Chart States in PCFG-Trained Transformers

As described in Section 3.1, we train the Transformer models using only a small fraction of the sequences generated by each grammar (e.g., less than $0.01\%$ of the data for the sampled example grammar $\texttt{CFG-1}$). Despite this limited supervision, the trained models exhibit low KL divergence

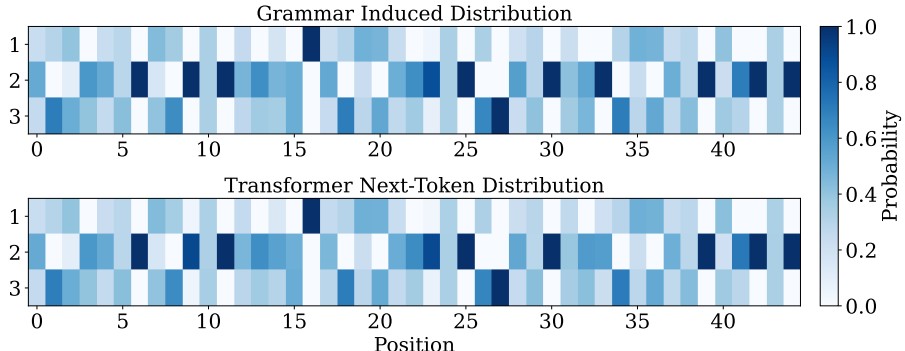

*Figure 2.* Visualization of the next-token distributions at each position of a sample sequence from `CFG-1`, a representative grammar from the 50-grammar pool. We compare the grammar-induced distribution with the Transformer-predicted distribution.

from the grammar-induced next-token distribution on unseen sequences, indicating that their performance cannot be attributed to mere memorization. Accurate next-token prediction under a context-free grammar requires resolving the latent parse state conditioned on the observed prefix. These observations naturally raise the question of how the trained Transformers internally achieve this capability.

Allen-Zhu & Li (2023) use multi-head linear probes to argue that Transformers may learn dynamic-programming-like computations akin to classical inside–outside parsing, while also noting that it remains difficult to precisely characterize which dynamic program is learned. Our analysis is designed to answer a different question. Rather than probing only the *final* layer with targets defined from the *full derivation tree*, we probe *every layer at every prefix position*. This applies both to the present Earley-state analysis and to the subsequent probing experiments, making it possible to trace the information flow across layers and to relate the recovered structure directly to the information required for next-token prediction.

Motivated by this gap, we examine classical parsing algorithms and analyze the Transformer's hidden representations through this lens. Given the autoregressive nature of Transformers, we first consider classical parsing algorithms that support prefix-conditioned generation.

> **Question 1.** Do PCFG-trained Transformers represent the Earley chart state needed for next-token prediction?

Among such algorithms, Earley (Earley, 1970; Jelinek & Lafferty, 1991) provides a canonical framework for prefix-conditioned generation under context-free grammars. Concretely, we perform probing analyses to examine whether the Transformer's internal representations progressively encode information sufficient to recover Earley chart states in a layer-by-layer manner.

**Probing Earley states**. We ask whether the representation

at position $t$ encodes the Earley chart state at that position. An Earley item is written as $(i, A \to \alpha \bullet \beta, w)$, where $i$ is the origin position, $A \to \alpha\beta$ is a production rule, the dot $\bullet$ marks progress through that rule, and $w$ denotes the item weight in the probabilistic setting. For a sequence $x_{1:n}$, let $\mathcal{E}_t$ be the set of Earley items at position $t$. We define a finite vocabulary of item types as tuples $(A, \alpha \bullet \beta)$, ignoring the start index $i$ and weight $w$. Then, for each layer $\ell$, we use the hidden state $h_t^{(\ell)}$ to predict a multi-hot label vector $y_t \in \{0, 1\}^{|\mathcal{I}|}$, where $y_t[\tau] = 1$ iff an item of type $\tau$ appears in $\mathcal{E}_t$. This gives a multi-label probing task of predicting the item types present in the chart state from $h_t^{(\ell)}$.

**Training the probing model**. For each grammar and each Transformer layer, we train a two-layer MLP with hidden size 256 and sigmoid outputs using binary cross-entropy. Probes are optimized with AdamW (batch size 256) for 100 epochs on position-wise input–target pairs from 16k sequences, with a learning rate of 0.003, and evaluated on input–target pairs from 4k held-out sequences (see Appendix C.2). We use MLP probes because the multi-label structured targets require probes with sufficient expressivity. Appendix C.6 shows that linear probes recover the same qualitative layer-wise trend but with consistently lower accuracy, supporting this choice. For the chosen MLP probes, the close match between training and test losses indicates good generalization rather than overfitting. A prediction is counted as correct only if all active labels in the target multi-hot vector are identified, and we report layer-wise probing accuracy averaged across the 50 sampled grammars.

**Probing results**. As shown in the left panel of Figure 3, the probing accuracy averaged across the 50 sampled grammars increases monotonically across the first four layers, exceeding 90% at its peak, before decreasing at the final layer. This pattern matches the sampled grammar family, whose derivation height is four: the first four layers progressively accumulate the information needed to represent Earley chart states, while the final layer appears to primarily reorganize

that information for next-token prediction, leading to lower probing accuracy.

> **Answer 1.** Transformer representations progressively accumulate sufficient information for Earley chart states, which enables accurate next-token prediction.

However, the COMPLETE operation in the Earley algorithm induces recursively nested sequential dependencies: when a constituent is completed at position $t$, it must be propagated to all previously created items that were awaiting it, which may originate from arbitrarily earlier positions. For a sequence of length $L$, this propagation requires $\mathcal{O}(L)$ sequential steps in the worst case. In contrast, as shown in Section 3.1, a Transformer that succeeds in autoregressive generation has only $\mathcal{O}(K + 1) = \mathcal{O}(\log L)$ layers, where $K$ denotes the grammar depth. This mismatch suggests that next-token prediction in Transformers is unlikely to be implemented via a purely Earley-style computation.

By contrast, if completed constituents are already available, incomplete states can be expanded locally to support next-token prediction. This observation motivates our focus on CYK or inside-style parsing, which recognizes completed constituents in a parallel, bottom-up manner.

### 3.3. Layer-Wise Emergence of CYK/Inside Constituents in PCFG-Trained Transformers

If the model uses bottom-up parsing, its intermediate representations should encode completed constituents. For instance, in the sampled example grammar `CFG-1` (Figure 1), the substring [1, 2, 3] forms constituent 7 at a higher layer. We thus investigate the following question.

> **Question 2.** Do PCFG-trained Transformers construct CYK/inside-style *bottom-up* constituents across layers?

To answer this question, we probe the hidden states of a trained Transformer for evidence of CYK-style chart construction across layers and at every position. Relatedly, prior work (Allen-Zhu & Li, 2023) probes for the ancestors of terminals, but restricts probing to the final Transformer layer and to positions corresponding to nonterminal (NT) boundaries, assuming full access to the global derivation tree. This setting differs fundamentally from autoregressive generation: when only a prefix is observed, the true ancestor of each terminal is generally *ambiguous*, with multiple valid possibilities. In contrast, we consider a more general *prefix-based* setting, in which probes are applied at every position and targets are defined solely from the observed prefix using CYK/inside-style algorithms, which operate without prior knowledge of constituent boundaries.

**Probing completed constituents**. We run the CYK algorithm to record completed constituents at their ending positions (see Appendix C.4). Let $\mathcal{C}_t$ denote the set of nonterminals $A \in \mathcal{N}$ such that $\alpha_{i,t}^A \neq 0$ for some $i \leq t$. We use the nonterminals in $\mathcal{N}$ as the label vocabulary, and for each layer $\ell$, we probe whether $h_t^{(\ell)}$ predicts the multi-hot vector $z_t \in \{0, 1\}^{|\mathcal{N}|}$, where $z_t[A] = 1$ if and only if $A \in \mathcal{C}_t$. This gives a multi-label probing task for predicting which completed constituents end at position $t$. We train a two-layer MLP probe independently for each layer using the same protocol as in Section 3.2; Appendix C.6 provides the training and test losses, along with corresponding linear-probe comparison.

The middle panel of Figure 3 shows that the probing accuracy for completed constituents, averaged across the 50 sampled grammars, increases monotonically across layers. This indicates that the Transformer gradually accumulates information corresponding to bottom-up completed constituents, consistent with CYK-style parsing behavior, and reaches high accuracy in the final layer across the sampled grammar family we test. Moreover, the CYK-derived information is sufficient to determine the final Earley state, since CYK provides exactly the completed constituent summaries required by Earley's COMPLETE operation.

**Span-restricted probing**. Probing all completed constituents at each position only establishes recoverability; it does not reveal how bottom-up information is constructed across layers. Motivated by the divide-and-conquer structure of CYK and inside parsing, which build larger-span constituents from smaller ones, we introduce span-restricted probing based on window size.

For each nonterminal layer $k$, let $L_k$ denote the maximum span length of nonterminals in $\mathcal{N}_k$. Given a window size $L$, we restrict labels to completed constituents whose span length satisfies $l \leq L$. Formally, at position $t$, a nonterminal $A$ is included in $\mathcal{C}_t^{(L)}$ if and only if $C[i, t, A] = 1$ for some $i \geq t - L + 1$. We then construct four multi-hot label sets using $\mathcal{C}_t^{(L)}$ with $L = L_1, L_2, L_3, L_4$, corresponding to the max-span windows induced by nonterminal layers 1, 2, 3, and 4, respectively, and probe every Transformer layer using the same MLP architecture. In the right panel of Fig. 3, the horizontal axis is indexed by nonterminal layer rather than by the numerical window size itself.

The right panel of Fig. 3 reports the resulting probing accuracy heatmap averaged across the 50 sampled grammars. Stars ($\star$) mark the first Transformer layer at which accuracy exceeds 90%, while dots ($\bullet$) indicate the layer achieving the highest accuracy for each nonterminal-layer-specific max-span window. Two clear trends emerge. First, as the nonterminal layer increases, the corresponding max-span window requires deeper Transformer layers before probing accuracy exceeds 90%. Second, the layer achieving peak accuracy also moves to higher Transformer layers for higher nonterminal layers.

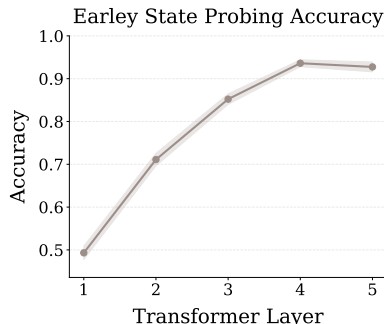
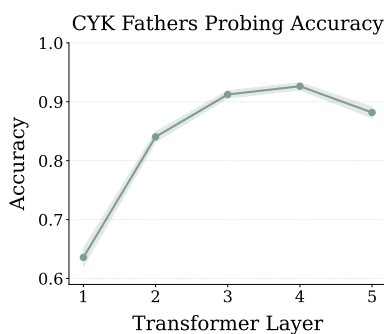
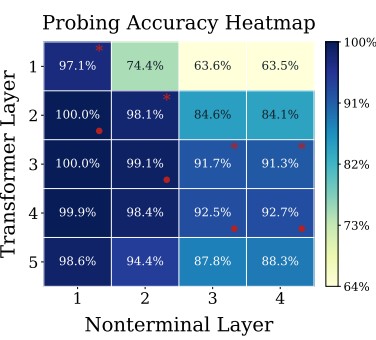

*Figure 3.* **Left.** Layer-wise probing accuracy for Earley-style prefix states. **Middle.** Layer-wise probing accuracy for bottom-up completed constituents. **Right.** Span-restricted probing accuracy for completed constituents, with columns corresponding to nonterminal layers 1–4, represented by their maximum span windows. All results are averaged over 50 sampled grammars; shaded regions in the left and middle panels denote the standard error of the mean (SEM) across grammars. In the right panel, $\star$ marks the first Transformer layer whose accuracy exceeds $90\%$, and $\bullet$ marks the layer with the highest accuracy for each nonterminal layer.

These patterns indicate that the Transformer encodes shorter-span constituents in earlier layers and progressively constructs larger-span constituents in deeper layers. In the final layer, probing accuracy decreases across all window sizes, suggesting that the model transitions from representing parsing structure to aggregating information for next-token prediction. (We note that labels corresponding to the largest spans are relatively rare, which may explain the similar performance observed for the two largest window sizes; this does not affect the overall conclusion.)

> **Answer 2.** Transformer layers progressively build bottom-up CYK-style representations via divide-and-conquer parallelism.

Overall, these results provide evidence consistent with Transformers learning a bottom-up parsing-like mechanism in a hierarchical, divide-and-conquer manner closely resembling CYK and inside algorithms. This behavior is consistent with observations reported in Liu et al. (2022).

> **Conclusion.** Transformers progressively implement bottom-up CYK-style parsing via divide-and-conquer parallelism (*local* aggregation). The resulting boundary representations are sufficient to advance Earley-style prefix states (*global* parsing), thereby supporting grammar-consistent next-token prediction.

## 4. Local-to-Global Parsing in Transformers: A Theoretical Account

To complement the preceding empirical findings, we further seek stronger theoretical support.

> **Question 3.** Can a Transformer architecture *provably* implement local-to-global parsing?

We provide an affirmative answer in this section.

### 4.1. Main Results

To formalize the local-to-global mechanism, we focus on balanced full-binary PCFGs, a clean base case for isolating the computational role of Transformer layers.

**Definition 4.1** (Balanced Full-Binary PCFG). Following Definition 2.1, a *balanced full-binary PCFG* of depth $K$ is a depth-bounded grammar $\mathcal{G} = (\mathcal{N}, \Sigma, \mathcal{R}, S, p)$ where each production expands a nonterminal into exactly two children from the subsequent layer: $\mathcal{R}_k := \{C \rightarrow AB \mid C \in \mathcal{N}_k, \ A, B \in \mathcal{N}_{k-1}\}, \mathcal{R} = \sqcup_{k=1}^{K} \mathcal{R}_k$.

This grammar class admits a finite and deterministic span structure. Every level-$k$ constituent spans exactly $2^k$ terminals (perfect balance), and a level-$k$ constituent ending at position $t$ exists if and only if $t \equiv 0 \pmod{2^k}$, which uniquely determines its split into two children. Such positions are referred to as *dyadic boundaries*. For example, if $k = 2$, then the boundaries are $4, 8, 12, \cdots$.

Within this setting, we show that there exists a Transformer architecture that realizes the local-to-global mechanism described above, with an illustration in Figure 4.

**Theorem 4.2** (Informal). *For any balanced full-binary PCFG of depth $K$, there exists a $(K+1)$-layer Transformer that: (i) computes bottom-up CYK-style representations in parallel at dyadic boundaries across the first $K$ layers; and (ii) aggregates these boundary representations in the final layer to perform Earley-style prefix rollout and produce the correct next-token distribution.*

For a balanced full-binary PCFG of depth $K$ with a constant number (e.g., three) of production rules per nonterminal, the generated sequences have length $\mathcal{O}(2^K)$. In contrast, autoregressive generation can be implemented by a Transformer with only $\mathcal{O}(K)$ layers, i.e., logarithmic in the length of the

sequences generated by the grammar.

*Remark* 4.3. Although our construction is presented for balanced full-binary grammars, the underlying intuition extends to more general bounded-depth grammars. In such grammars, the admissible span lengths and ending positions are still deterministic and finite, which allows a prefix to be decomposed in an analogous, albeit more complex, manner. While this requires additional attention heads to account for multiple possible decompositions, the overall aggregation principle remains the same.

Our construction differs from prior constructive results in masked or bidirectional settings in a fundamental way: it targets causal autoregressive generation, where the model sees only a prefix and must maintain a sufficient statistic for all valid continuations. Zhao et al. (2023) study inside–outside-style inference where each position has access to the full string, and Garnier-Brun et al. (2024) analyze tree-like attention patterns under a masked language modeling objective. In contrast, our result makes the causal computation explicit through an Earley-style prefix summary, capturing how a causal Transformer can perform prefix-conditioned rollout for next-token prediction, albeit under the balanced full-binary restriction above.

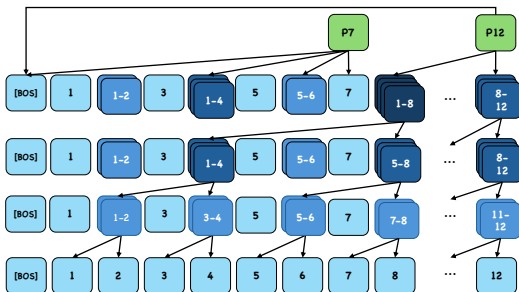

*Figure 4.* Illustration of token-level attention and information flow in the theoretical model, which first performs progressive local aggregation (blue) and then carries out Earley-style global prefix parsing (green). Darker blue indicates aggregation over a longer context window. The arrow denotes the attention pattern at the critical point. `[BOS]` encodes the initial Earley state.

### 4.2. Proof Sketch

**Local Aggregation (Bottom-Up Parsing)**. Theorem 4.2 shows that the early layers of the Transformer can implement CYK-style bottom-up parsing at dyadic boundaries. This computation follows a divide-and-conquer paradigm, in which short spans are recursively composed into longer spans in parallel, mirroring the inside recursion in Eq. (1). By constructing two attention heads that aggregate information from the current position and from previous dyadic boundaries, we prove by induction that the Transformer exactly simulates the inside dynamic program, with inside probabilities stored at dyadic boundaries (proofs in Appendix D.6). For instance, the inside probability $\alpha_{1,4}^A$ can

be computed by attending to the dyadic subspans $\alpha_{1,2}^B$ and $\alpha_{3,4}^C$, corresponding to a single binary production $A \rightarrow BC$. This construction aligns with our empirical findings in Section 3.3, where Transformer layers are observed to progressively encode CYK-style bottom-up representations via divide-and-conquer parallelism.

**Global Prefix Parsing (Earley-Style Rollout)**. As discussed in Section 3, given the bottom-up representations of completed spans, these local constituents can be aggregated to form a global parsing state conditioned on the observed prefix, which is sufficient for next-token prediction. Formally, we demonstrate that the Earley state at prefix position $t$ can be derived using only PREDICT and SCAN operations, bypassing COMPLETE entirely. This is achieved by consuming maximal dyadic constituents in a canonical order; due to its technical complexity, the formal statement and proof of this result (Lemma D.2) are deferred to Appendix D.8.2.

Intuitively, this consumption order corresponds to decomposing the parsed prefix into a sequence of maximal completed subtrees in a balanced binary parse tree. Because each subtree is already complete, no further completion is required, and the parsing state can be updated deterministically. As illustrated in Figure 4, constructing the Earley state at $t = 7$ merely requires consuming the maximal constituents $\alpha_{1,4}^{A_2}$, $\alpha_{5,6}^{A_1}$, and $\alpha_{7,7}^{A_0}$, where $A_k \in \mathcal{N}_k$.

In the final Transformer layer, the model aggregates the representations of maximal completed constituents using attention, and a subsequent MLP maps this aggregated representation to the exact Earley state (proof provided in Appendix D.8). This theoretical result aligns with our empirical findings in Section 3, which show that complete constituent information is sufficient to recover Earley-style prefix states.

### 4.3. Experimental Evidence

To empirically validate our construction, we train a 5-layer GPT-style Transformer (1 head, $d = 256$) on sequences from a 4-layer balanced full-binary grammar. We then analyze the resulting attention patterns, with full implementation details deferred to Appendix C.2.

**Local aggregation pattern**. As shown in Figure 5, the attention maps of the first four Transformer layers exhibit clear subtree structures. Lower layers primarily attend to smaller windows, while higher layers progressively focus on larger spans. This hierarchical pattern is consistent with our theoretical construction, in which the first $K$ layers implement a local aggregation process that progressively builds constituents of increasing size. This qualitative picture is also supported quantitatively: Table 3 in Appendix C.8 shows that, across layers, most attention mass falls inside the theory-predicted local window.

**Causal intervention**. To move beyond visual alignment,

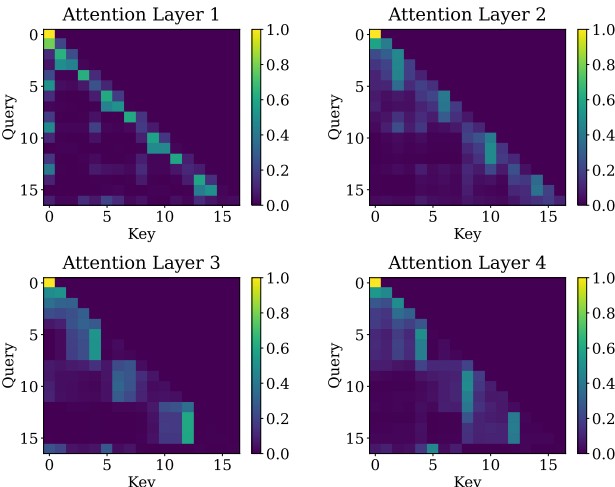

*Figure 5.* Attention maps from the first $K$ Transformer layers show a progressively expanding effective attention window. Lower layers concentrate on local neighborhoods, while higher layers attend over increasingly larger spans of the input sequence, reflecting hierarchical aggregation.

we perform a targeted causal test for the first four layers, which in our construction implement bottom-up CYK-style aggregation. As defined above, a level-$k$ constituent ends at dyadic boundaries satisfying $t \equiv 0 \pmod{2^k}$, and its representation is formed by combining information from the preceding dyadic boundary at offset $2^{k-1}$. Thus, the predicted offsets for Layers 1–4 are $1, 2, 4, 8$, respectively. We therefore mask each such offset separately and measure the resulting generation accuracy. As shown in Table 1, the largest accuracy drop occurs exactly at the theory-predicted offset for Layers 1–3, producing the diagonal pattern anticipated by the construction. This goes beyond recoverability: removing the dyadic-boundary connection singled out by the theory degrades performance in precisely the way the theory predicts. The weaker effect in deeper layers is also expected, since higher-level dyadic boundaries are rarer and therefore have more limited influence. In particular, at the deepest level there is only one such boundary in the whole sequence, so masking it can affect only the prediction at the end of sequences.

| Layer | Offset = 1 | Offset = 2 | Offset = 4 | Offset = 8 |
|-------|-----------|-----------|-----------|-----------|
| 1 | **0.6596** | 0.9995 | 0.9995 | 0.9995 |
| 2 | 0.9995 | **0.8220** | 0.9995 | 0.9995 |
| 3 | 0.9995 | 0.9995 | **0.9849** | 0.9995 |
| 4 | 0.9995 | 0.9995 | 0.9995 | 0.9995 |

*Table 1.* Generation accuracy under fixed-offset attention masking in the first four layers. Boldface marks the lowest accuracy in each row, equivalently the largest accuracy drop; the unmasked baseline is 0.9995.

**Global aggregation pattern**. Figure 6 provides direct evidence for the final-layer rollout mechanism. For prefixes ending at $t = 7, 9, 12$, the binary decomposition pre-

dicts maximal completed constituents ending at positions $\{4, 6, 7\}$, $\{8, 9\}$, and $\{8, 12\}$, respectively. The learned attention patterns closely match these predictions: the clearest peaks occur at the predicted non-endpoint positions, while the current endpoint $t$ still receives nonzero attention. Overall, attention concentrates on the boundary summaries identified by the dyadic decomposition, suggesting that the final layer globally aggregates them to form the prefix representation. The close alignment with the theoretical construction in Figure 4 further supports our theory.

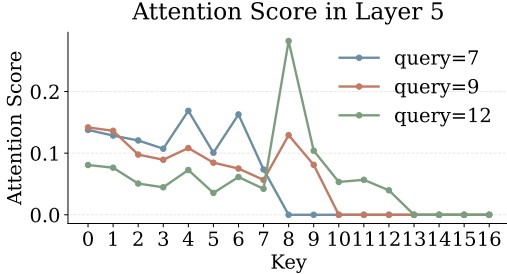

*Figure 6.* Attention map of the last Transformer layer for positions 7, 9, and 12. The dominant peaks align with the dyadic boundary summaries predicted by the final-layer Earley-style rollout.

Due to approximation effects during training, attention patterns are not always perfectly sharp: relevant boundary positions often appear as clear peaks rather than exact maxima, which nevertheless suffices to reflect the intended hierarchical structure. Therefore, we can answer the question posed at the beginning.

**Answer 3.** There exists a Transformer that realizes a local-to-global parsing mechanism for balanced full-binary grammars, combining bottom-up span aggregation with prefix-conditioned parsing, consistent with the behavior observed in trained models.

## 5. Conclusion and Discussion

In this paper, we provide a unified account of how Transformers autoregressively generate sequences from probabilistic context-free grammars (PCFGs) via a local-to-global mechanism that incorporates hierarchical parsing structure. Our probes reveal that hidden representations encode both Earley-style prefix information and CYK-like bottom-up signals, organized in a progressive local-to-global manner across layers. These representations enable accurate next-token prediction under PCFGs with a Transformer of logarithmic depth. We present a theoretical construction of a Transformer implementing this local-to-global mechanism, consistent with observed attention patterns. Together, our results provide a principled basis for understanding how Transformers capture latent hierarchical structure and motivate future studies of their training dynamics.

## Impact Statement

This work clarifies how Transformers can represent and update hierarchical syntactic structure during left-to-right generation by relating depth-bounded PCFGs and classical parsing algorithms to layerwise computation. These insights support more principled probing and evaluation of structure-sensitive generalization. While our analysis uses controlled grammar-based settings and may not fully transfer to unrestricted natural language, we expect primarily positive impact through improved transparency and diagnostics for widely deployed language models.

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

## A. Related Work

**Context-Free Grammars and Parsers**. Context-free grammars and probabilistic context-free grammars are classical objects in computational linguistics and programming language theory. Standard parsing algorithms include CYK and inside–outside parsing, which compute span-based representations bottom-up (Cocke, 1969; Younger, 1967; Kasami, 1966; Lari & Young, 1990), and the Earley algorithm, which supports left-to-right parsing and prefix probability computation (Earley, 1970; Stolcke, 1995; Jelinek & Lafferty, 1991). The parallelization of the CYK algorithm (Bernardy & Claessen, 2013) and the Earley algorithm (Ahrens et al., 2018; Kim et al., 2019; Nowak & Cotterell, 2023) has also been an active area of research.

**Transformers and Context-Free Language Generation**. Prior work on context-free languages (CFLs) has primarily focused on recognition tasks (Ebrahimi et al., 2020; Hahn, 2020; Weiss et al., 2021; Delétang et al., 2022; Jerad et al., 2026) or diffusion-based generative models (Favero et al., 2025). Yao et al. (2021) show that self-attention networks can generate bounded-depth Dyck languages, while Wen et al. (2023) demonstrate that correct generation does not imply interpretable internal algorithms. In contrast, we investigate *autoregressive* generation of context-free languages and characterize the internal algorithmic structure the trained model has learned.

The most closely related empirical work is Allen-Zhu & Li (2023), who probe Transformer representations on finite-depth layered PCFGs and propose an inside–outside interpretation. We extend their approach in two respects: we probe *every layer at every prefix position* rather than the final layer alone, and we use *prefix-conditioned* targets—Earley states and CYK-father information—rather than the full derivation tree, which is unavailable during generation. This allows us to make their multi-stage hypothesis precise: CYK-style bottom-up constituent building in early layers followed by Earley-style prefix rollout in the final layer, supported by probing, attention analysis, causal interventions, and a matching theoretical construction.

Cagnetta & Wyart (2024); Cagnetta et al. (2024; 2025) and Schulz et al. (2025) study hierarchical language models from the perspectives of learning dynamics and sample complexity, with a focus on how representations emerge during training. Their focus is the *data* side—how hierarchical input structure shapes learning dynamics—while ours is the *model* side: what algorithm the trained model implements, analyzed through full autoregressive generation at every prefix position and formalized via an explicit $O(\log L)$-depth Transformer construction.

Zhao et al. (2023) construct a Transformer implementing the Inside-Outside algorithm, and Garnier-Brun et al. (2024) provide a Belief Propagation–based construction accounting for tree-like attention patterns; both work in the *masked language modeling* (MLM) setting. We study the causal, autoregressive setting, where next-token prediction requires tracking running prefix summaries rather than performing full-string inference—a structurally different problem our construction addresses via Earley-style states within $O(\log L)$ depth (versus $O(L)$ in Zhao et al. (2023)). Extending our result beyond balanced full-binary PCFGs to general PCFGs remains open.

**Expressive Power of Transformers**. Pérez et al. (2019); Merrill (2021); Strobl et al. (2024); Jerad et al. (2026) investigate the theoretical properties of the Transformer architecture. These analyses demonstrate that Transformers can compress certain sequential computations into logarithmic depth by leveraging parallelism, achieved either through parallel divide-and-conquer constructions (Liu et al., 2022) or large-scale parallel computation (Sanford et al., 2024). Furthermore, Merrill & Sabharwal (2023) characterize the inherent trade-offs associated with this parallelism.

**Mechanistic Interpretation.** Our work contributes to the mechanistic interpretability literature by studying internal computations in Transformers trained on structured data. See, e.g., (Belrose et al., 2023; Sakarvadia et al., 2023; Stolfo et al., 2023; Allen-Zhu & Li, 2023; Ferrando et al., 2024; He et al., 2024; Chen et al., 2024; Ball et al., 2024; Orgad et al., 2024; Prakash et al., 2024; Quirke & Barez, 2023; Ameisen et al., 2025; Chen et al., 2025; Li et al., 2025; Kharlapenko et al., 2025) and the references therein for details, and (Zhang et al., 2026) for a recent survey. Most existing mechanistic interpretability studies focus on different data generation schemes (e.g., arithmetic, copying, or simple pattern completion), which involve different underlying mechanisms and thus cannot be directly compared to our setting.

## B. Classical Parsing Algorithms: Details and Examples

This appendix provides additional details on the CYK and Earley parsing algorithms. As shown in the main text, Transformers trained on PCFGs appear to recover key aspects of these classical algorithms, motivating a self-contained review here.

## B.1. Background: What is Parsing?

A context-free grammar (CFG) defines a set of rules for generating strings. To illustrate, consider a simple grammar with three rules: $S \rightarrow AB$, $A \rightarrow a$, and $B \rightarrow b$. These rules specify that the start symbol $S$ can be rewritten as the sequence $AB$, the nonterminal $A$ can be rewritten as the terminal $a$, and $B$ can be rewritten as $b$. Starting from $S$, we can derive the string "ab" through the sequence of rewrites $S \Rightarrow AB \Rightarrow aB \Rightarrow ab$.

Parsing is the reverse problem: given a string like "ab", we want to determine *how* it was generated—that is, to reconstruct the derivation tree that produced it. For probabilistic CFGs (PCFGs), where each rule has an associated probability, parsing also involves computing the probability of the string under the grammar.

A key concept is that of a *constituent*: a contiguous substring that corresponds to a single nonterminal in the parse tree. In our running example, "a" is a constituent generated by $A$, "b" is a constituent generated by $B$, and the entire string "ab" is a constituent generated by $S$. We say a constituent is *complete* when a nonterminal has finished deriving an entire span, meaning the derivation from that nonterminal down to terminals is fully determined.

## B.2. CYK / Inside Algorithm

The CYK algorithm answers a fundamental question: *for each substring of the input, which nonterminals can generate it?* We present the probabilistic version, often called the Inside algorithm, which computes the total probability summed over all derivations. The standard CYK algorithm instead uses a `max` operator to find the single best parse (Viterbi parsing). Since we care about probability distributions over next tokens, the summing version is the relevant one for this paper.

**The key idea.** CYK employs dynamic programming to build up answers from small substrings to large ones. The algorithm proceeds in phases organized by span length. In the first phase, it determines which nonterminals can generate each individual token (length-1 spans). In the second phase, it determines which nonterminals can generate each pair of adjacent tokens (length-2 spans) by considering all ways to combine the length-1 results. This process continues for length-3, length-4, and so on, until the algorithm reaches the full string.

**Formal description.** We present the algorithm for grammars in Chomsky Normal Form, where every rule is either binary ($A \rightarrow BC$) or terminal ($A \rightarrow a$). Grammars with longer right-hand sides can be converted to this form, or the algorithm can be extended with additional split points.

Let $\alpha_{i,j}^A$ denote the probability that nonterminal $A$ generates the substring from position $i$ to position $j$ (using 1-based indexing, so position 1 is the first token). For the base case of length-1 spans, if there is a terminal rule $A \rightarrow a$ and the token at position $i$ is $a$, then $\alpha_{i,i}^A = p(A \rightarrow a)$. For longer spans, we compute $\alpha_{i,j}^A$ by trying all ways to split the span $[i, j]$ into two parts:

$$\alpha_{i,j}^A = \sum_{A \rightarrow BC \in \mathcal{R}} p(A \rightarrow BC) \sum_{k=i}^{j-1} \alpha_{i,k}^B \cdot \alpha_{k+1,j}^C.$$

In words, the probability that $A$ generates $[i, j]$ equals the sum over all ways to split $[i, j]$ at some position $k$, have $B$ generate the left part $[i, k]$, have $C$ generate the right part $[k + 1, j]$, and multiply by the rule probability $p(A \rightarrow BC)$.

**Concrete example.** Consider parsing the string "ab" with our grammar $S \rightarrow AB$, $A \rightarrow a$, $B \rightarrow b$ (all rules with probability 1). In the first phase, we process length-1 spans: position 1 contains "a", so $\alpha_{1,1}^A = 1$ since $A \rightarrow a$; position 2 contains "b", so $\alpha_{2,2}^B = 1$ since $B \rightarrow b$. In the second phase, we ask whether $S$ can generate positions 1-2. Trying the only possible split at $k = 1$, we check whether $A$ can generate $[1, 1]$ (yes, $\alpha_{1,1}^A = 1$) and $B$ can generate $[2, 2]$ (yes, $\alpha_{2,2}^B = 1$). Combining these, we obtain $\alpha_{1,2}^S = p(S \rightarrow AB) \cdot \alpha_{1,1}^A \cdot \alpha_{2,2}^B = 1$.

**Why CYK is parallelizable.** The crucial property enabling parallelization is that **all spans of the same length are independent of each other**. Computing any length-2 span requires only length-1 spans, and computing any length-3 span requires only length-1 and length-2 spans. This means that once we have computed all spans of a given length, we can compute *all* spans of the next length simultaneously, in parallel. There are no "backward" dependencies—longer spans never depend on even longer spans. This hierarchical, bottom-up structure makes CYK naturally suited for parallel architectures like Transformers, where each layer can aggregate information from progressively larger spans.

**Limitation of CYK.** While CYK efficiently tells us *what* constituents exist (which nonterminals can generate which spans), it does not tell us *where we are* in the parsing process. Given only a prefix of a string, CYK cannot directly tell us what tokens

might come next—it only computes information about complete spans. This limitation motivates the Earley algorithm.

## B.3. Earley Algorithm

The Earley algorithm answers a different question from CYK: *given the tokens we have seen so far, what are all the possible ways the parse could be progressing, and what tokens could legally come next?* Unlike CYK, which processes complete spans bottom-up, Earley reads the input left-to-right, maintaining a running summary of all partial parses consistent with the prefix seen so far.

**Data structures: charts and items.** Earley uses position-based indexing where position $k$ refers to the point *after* reading $k$ tokens: position 0 is before any input, position 1 is after the first token, and so on. For a prefix of length $t$, the algorithm maintains charts $\texttt{Chart}[0], \texttt{Chart}[1], \dots, \texttt{Chart}[t]$, each storing a set of *Earley items* that represent partial parse states.

An Earley item is written $(i, A \to \alpha \bullet \beta)$, where $i$ is the *origin position* indicating where this constituent began, $A \to \alpha\beta$ is a production rule, and the dot $\bullet$ marks how much we have recognized. Concretely, an item $(i, A \to \alpha \bullet \beta)$ stored in $\texttt{Chart}[k]$ means: "We are trying to build nonterminal $A$ starting from position $i$. We have successfully matched $\alpha$ against input positions $i + 1$ through $k$. We are now at position $k$ and still need to match $\beta$." For example, $(0, S \to A \bullet B)$ in $\texttt{Chart}[1]$ means we started parsing $S$ at position 0, have recognized $A$ covering position 1, and now need to find $B$.

**The three operations.** Earley advances the parse through three operations, each handling a different situation based on what symbol appears immediately after the dot:

- PREDICT (dot before a nonterminal): Suppose $\texttt{Chart}[k]$ contains $(i, A \to \alpha \bullet B\beta)$ where $B$ is a nonterminal. This item needs a $B$ to appear starting at position $k$. To find such a $B$, we must try all grammar rules that produce $B$. For each rule $B \to \gamma$, we add $(k, B \to \bullet\gamma)$ to $\texttt{Chart}[k]$. The new item has origin $k$ because we start looking for $B$ at the current position, and the dot is at the beginning because we have not yet matched any part of $\gamma$.

- SCAN (dot before a terminal): Suppose $\texttt{Chart}[k]$ contains $(i, A \to \alpha \bullet a\beta)$ where $a$ is a terminal, and the next input token is $x_{k+1} = a$. Since the expected terminal matches the actual input, we advance past it: add $(i, A \to \alpha a \bullet \beta)$ to $\texttt{Chart}[k + 1]$. The origin $i$ stays the same because we are still building the same constituent; only the dot advances to reflect consuming one more token.

- COMPLETE (dot at the end): Suppose $\texttt{Chart}[k]$ contains $(j, B \to \gamma\bullet)$ where the dot is at the end, meaning we have finished recognizing $B$ spanning positions $j + 1$ through $k$. We must now "callback" to inform all items that were waiting for this $B$. We examine $\texttt{Chart}[j]$ (where $B$ started) and find all items $(i, A \to \eta \bullet B\theta)$ that were waiting for a $B$ at position $j$. For each such item, we add $(i, A \to \eta B \bullet \theta)$ to $\texttt{Chart}[k]$, advancing their dot past $B$ since that $B$ has now been found.

**Algorithm flow.** The algorithm initializes $\texttt{Chart}[0]$ with the seed item $(0, S' \to \bullet S)$, representing the goal of parsing start symbol $S$ from position 0. It then processes positions $k = 0, 1, \dots, t$ in order. At each position $k$, we repeatedly apply PREDICT and COMPLETE to items in $\texttt{Chart}[k]$ until no new items appear (reaching a fixed point). Then, if $k < t$, we apply SCAN to move items into $\texttt{Chart}[k + 1]$ based on input token $x_{k+1}$.

After processing all positions, we can read out which terminals are valid next tokens: any terminal $a$ such that some item $(i, A \to \alpha \bullet a\beta)$ exists in $\texttt{Chart}[t]$ is a grammatically valid continuation of the prefix.

**A Concrete example.** Let us trace Earley on "ab" with grammar $S \to AB$, $A \to a$, $B \to b$. At position 0, we start with $(0, S \to \bullet AB)$. PREDICT sees the dot before nonterminal $A$, so it adds $(0, A \to \bullet a)$.

At position 1, after reading "a", SCAN advances $(0, A \to \bullet a)$ to $(0, A \to a\bullet)$ in $\texttt{Chart}[1]$. Now COMPLETE processes this finished item: $A$ started at position 0, so we check $\texttt{Chart}[0]$ and find $(0, S \to \bullet AB)$ waiting for $A$. We advance it to $(0, S \to A \bullet B)$ in $\texttt{Chart}[1]$. Then PREDICT sees the dot before $B$ and adds $(1, B \to \bullet b)$.

At position 2, after reading "ab", SCAN advances $(1, B \to \bullet b)$ to $(1, B \to b\bullet)$. COMPLETE processes this: $B$ started at position 1, so we check $\texttt{Chart}[1]$ and find $(0, S \to A \bullet B)$ waiting for $B$. We advance it to $(0, S \to AB\bullet)$ in $\texttt{Chart}[2]$. The parse is complete.

**Extension to probabilistic parsing.** The basic Earley algorithm determines *which* tokens can come next, but for language modeling we need *probabilities*. Since Transformers are probabilistic prediction machines, this extension is essential for

understanding the connection.

We augment each item with a weight $w$ representing the total probability mass of all derivations leading to that item. The item $(i, A \to \alpha \bullet \beta, w)$ now means: "The sum of probabilities of all derivation paths that reach this partial parse state is $w$." The three operations propagate weights as follows:

- PREDICT: When creating $(k, B \to \bullet\gamma)$ from an item with weight $w$, the new item receives weight $w \cdot p(B \to \gamma)$, where $p(B \to \gamma)$ is the rule probability. This multiplication accounts for choosing which rule to expand $B$.

- SCAN: When advancing $(i, A \to \alpha \bullet a\beta, w)$ to $(i, A \to \alpha a \bullet \beta, w')$, we set $w' = w$. In standard PCFGs, terminal emission probabilities are absorbed into the rule probabilities, so the weight passes through unchanged.

- COMPLETE: When a completed item $(j, B \to \gamma\bullet, w_B)$ advances a waiting item $(i, A \to \eta \bullet B\theta, w_A)$, the new item receives weight $w_A \cdot w_B$—the product of reaching the waiting state and completing the constituent. Crucially, if multiple derivation paths produce the same item, we *sum* their weights. This summation is what makes the algorithm compute true prefix probabilities rather than Viterbi (max) approximations.

To compute the next-token distribution $P(x_{t+1} \mid x_{1:t})$, we examine $\texttt{Chart}[t]$ and, for each terminal $a$, sum the weights of all items waiting for $a$. Normalizing these sums yields the conditional distribution. This procedure, the probabilistic Earley algorithm (Stolcke, 1995), correctly computes prefix probabilities for PCFGs. In practice, computations use log-space to avoid numerical underflow.

We note that computing weights correctly for grammars with left-recursion or unit productions requires care to avoid infinite loops or double-counting; Stolcke (1995) addresses these cases. For the bounded-depth grammars in this paper, such complications do not arise.

**Why Earley is sequential (not parallelizable).** The COMPLETE operation creates a fundamental sequential dependency: when we finish a constituent at position $k$ that originated at position $j$, we must examine $\texttt{Chart}[j]$ to find items waiting for that constituent. Since $j$ can be *any* earlier position from 0 to $k - 1$, we cannot process position $k$ until all earlier positions are fully processed.

We emphasize that "sequential" refers to the dependency structure relevant for parallel hardware, not overall efficiency. Earley often runs faster than CYK: $O(L^2)$ for unambiguous grammars versus CYK's $O(L^3)$. The point is that CYK's computation parallelizes across span lengths, whereas Earley's backward-looking COMPLETE creates position-by-position dependencies.

**Why Earley is useful for generation.** Despite being sequential, Earley directly supports autoregressive generation: at any position, items of the form $(i, A \to \alpha \bullet a\beta)$ tell us which terminals can appear next, and the probabilistic extension tells us their probabilities. This makes Earley the natural algorithm for understanding grammar-consistent generation.

**Summary: CYK vs. Earley.** CYK and Earley offer complementary strengths. CYK identifies *what* constituents exist through parallelizable bottom-up computation, but cannot directly support prefix-conditioned generation. Earley tracks *where* we are in the parse and supports next-token prediction, but requires sequential processing due to COMPLETE. The central finding of this paper is that Transformers combine both: they use CYK-style parallel aggregation to build constituent information, which then supports Earley-style prefix probabilities for next-token prediction.

## C. Experimental Settings and Additional Results

### C.1. Grammars Used in Experiments

**Random CFG family.** We consider a family of finite-depth layered PCFGs, i.e., a subclass of the grammars in Definition 2.1, with fixed height $H = 4$ and a fixed level decomposition. The start symbol is $S = 16$, and the levels are

$$N_4 = \{16\}, \quad N_3 = \{13, 14, 15\}, \quad N_2 = \{10, 11, 12\}, \quad N_1 = \{7, 8, 9\}, \quad N_0 = \Sigma = \{'1', '2', '3'\},$$

where $N_1, \ldots, N_4$ are nonterminal layers and $N_0$ is the terminal alphabet. Thus, every production expands a nonterminal in $N_k$ into a string over $N_{k-1}$ for some $k \in \{1, 2, 3, 4\}$, exactly as required by Definition 2.1. In particular, derivations are acyclic and have maximum depth 4 from the start symbol to terminals.

**Sample procedure.** We sample 50 grammars independently from this family for the experiments. For each nonterminal $A \in N_1 \cup N_2 \cup N_3 \cup N_4$, we sample between 2 and 4 distinct production rules. Each rule has the form

$$A \to X_1 X_2 \cdots X_m,$$

where $m \in \{2, 3\}$ and each symbol $X_j$ is drawn uniformly from the immediately lower level $N_{k-1}$ when $A \in N_k$. If two sampled right-hand sides for the same left-hand side coincide, we keep only one copy, so the productions associated with each nonterminal are distinct. Finally, for each fixed left-hand side $A$, we assign a uniform distribution over its sampled rules, so that $\sum_{r \in \mathcal{R}_A} p(r) = 1$.

**Some Examples of the Grammars** Figures 7 and 8 show five representative sampled grammars, labeled CFG-1 to CFG-5, from the 50-grammar pool used in the empirical study. Figure 9 also includes CFG-6, the balanced full-binary grammar satisfying Definition 4.1 that is used in Section 4.3.

**CFG 1**

```
16 → 15 13 | 13 14 15 | 13 15 14
13 → 11 11 | 11 12 11 | 11 11 10
14 → 12 11 11 | 10 12 10 | 12 12 10 | 10 11 11
15 → 12 11 | 12 12 10
10 → 9 9 | 7 8 | 7 7 | 8 8
11 → 7 8 7 | 7 9 7 | 9 8 | 8 7 8
12 → 9 9 | 8 7 8 | 7 9
 7 → '2' '1' '2' | '2' '2' '2' | '2' '3'
 8 → '3' '3' | '3' '2' '1' | '1' '3'
 9 → '1' '1' | '2' '3' '3' | '3' '2' | '1' '3'
```

**CFG 2**

```
16 → 14 15 13 | 13 14 15 | 14 14 | 14 14 15
13 → 12 10 12 | 10 11 11 | 10 11
14 → 12 11 | 11 10 | 12 10 11
15 → 11 12 12 | 11 12 10 | 12 11 11
10 → 8 9 | 9 8 | 8 7
11 → 8 8 7 | 9 7 8 | 9 7 | 8 8
12 → 8 7 | 8 9 | 8 8 9
 7 → '3' '2' '1' | '3' '1' '3'
 8 → '2' '1' '3' | '2' '1'
 9 → '1' '3' '3' | '2' '1'
```

*Figure 7.* Sampled example grammars CFG-1 and CFG-2 from the 50-grammar pool used in the experiments.

**CFG 3**

```
16 → 15 13 14 | 13 14
13 → 10 12 | 11 11 | 11 11 12 | 10 11 10
14 → 12 11 | 12 11 11 | 11 10 10
15 → 10 10 11 | 12 10 12 | 10 11 12 | 12 11
10 → 8 8 | 7 7 7
11 → 9 9 | 7 7 | 7 8 | 9 7
12 → 8 7 | 8 9 8
 7 → '3' '3' '3' | '1' '3' '3'
 8 → '2' '1' '2' | '1' '1' | '3' '2' | '3' '2' '2'
 9 → '1' '1' | '3' '3'
```

**CFG 4**

```
16 → 13 14 13 | 13 15
13 → 11 10 12 | 11 11 11 | 11 12
14 → 12 10 | 10 11
15 → 10 10 | 12 11 10 | 11 11 11
10 → 7 9 | 8 7 | 9 7 8 | 7 8
11 → 7 9 9 | 7 7 | 9 8
12 → 7 9 | 9 9
 7 → '3' '3' | '3' '2' | '1' '2'
 8 → '2' '3' '3' | '3' '3' '1'
 9 → '1' '1' | '1' '2' | '2' '3' | '2' '1' '2'
```

*Figure 8.* Sampled example grammars CFG-3 and CFG-4 from the 50-grammar pool used in the experiments.

**CFG 5**

```
16 → 13 14 14 | 13 15 | 15 13 | 13 14
13 → 10 11 12 | 12 10 10 | 11 12
14 → 11 12 | 12 12 | 11 10 12
15 → 11 12 10 | 10 11
10 → 7 8 | 8 9 7
11 → 9 8 7 | 8 7 8 | 7 7 | 7 8 9
12 → 9 9 | 8 8 | 7 9 9 | 7 9
 7 → '2' '3' '2' | '3' '2' | '1' '1'
 8 → '3' '1' | '3' '2'
 9 → '1' '1' '1' | '3' '3' | '2' '1' '2' | '2' '1'
```

**CFG 6**

```
16 → 15 13 | 13 14 | 14 15
13 → 12 11 | 11 10 | 10 10
14 → 11 12 | 10 11 | 12 12
15 → 10 12 | 11 11 | 12 11
10 → 8 9 | 9 7 | 9 8
11 → 8 7 | 7 7 | 9 9
12 → 7 9 | 8 7 | 8 8
 7 → '2' '2' | '2' '3' | '3' '2'
 8 → '1' '2' | '3' '1' | '1' '1'
 9 → '2' '1' | '1' '3' | '3' '3'
```

*Figure 9.* Sampled example grammar CFG-5 from the 50-grammar pool and the balanced full-binary PCFG CFG-6 used in the experiments to validate the construction.

## C.2. Settings and training details

**General Depth-bounded Grammar.** We train 50 independent autoregressive language models, each on sequences generated by one of the 50 grammars described in Section C.1. At every training step, we randomly generate a fresh batch of 256 sentences by sampling from the current grammar, and wrap each sentence with special tokens [BOS] and [EOS], padding to the grammar-specific context length required to cover the longest generated sequence. The model

is a `GPT2LMHeadModel` with 5 layers, 4 heads, and hidden size dimension 512 (GPT-2 default MLP width is 4 times the hidden dimension), with positional length matched to the context length. We train a separate model for each sampled grammar by minimizing the standard cross-entropy loss $\mathcal{L}(\theta) = -\sum_{t=0}^{T} \log p_\theta(x_{t+1} \mid x_{1:t})$ where $x_0 = $ `[BOS]` and $x_{T+1} = $ `[EOS]`. We use AdamW(Loshchilov & Hutter, 2017) with learning rate 0.002, $\beta_1, \beta_2 = (0.9, 0.98)$, and weight decay 0.1, together with a cosine learning-rate schedule with 4000 warmup steps over 80,000 total training steps.

**Balanced Full-binary Grammar.** We use CFG-6 in Figure 9 and follow the same sampling and packing strategy above, but pack each batch with 96 sequences. We use a `GPT2LMHeadModel` with 5 layers, 1 head (for clearer attention map visualization), and hidden size dimension is 256 because the grammar is small. We optimize the standard next-token cross-entropy objective. We uses AdamW(Loshchilov & Hutter, 2017) with learning rate 0.001, $\beta_1, \beta_2 = (0.9, 0.98)$, and weight decay 0.1, together with a cosine learning-rate schedule with 1000 warmup steps over 20,000 total training steps.

**Probing model.** All probing tasks are formulated as multi-label classification problems. Accordingly, we employ a two-layer MLP with ReLU activations for all probes. The input dimension matches the hidden dimension of the Transformer, while the output dimension corresponds to the size of the encoded target space (i.e., the number of Earley states or completed nonterminals in CKY). We fix the hidden layer dimension to 256 and optimize the probes model using AdamW (Loshchilov & Hutter, 2017) with batch size 256. A learning rate of 0.003 is used for tasks based on the sampled random grammars, and 0.001 for tasks based on `CFG-6` (binary). All probes are trained for 100 epochs on 16,000 sequences and evaluated on a held-out set of 4,000 sequences.

**Attention visualization.** The attention maps shown in Figures 5 and 6 are obtained using the balanced full-binary grammar model described above (`CFG-6`, 5 layers, 1 head, $d = 256$). Each attention map is computed by averaging the per-position attention weights over 10,000 sequences independently sampled from the grammar, so that the displayed patterns reflect the model's typical behavior rather than any single sequence.

**Causal intervention.** The generation accuracy results reported in Table 1 are obtained on the same balanced full-binary grammar model. For each layer $k \in 1, 2, 3, 4$ and each tested offset $\Delta \in 1, 2, 4, 8$, we identify the dyadic boundary positions satisfying $t \equiv 0 \pmod{2^k}$ and mask the attention edge from each such position $t$ to the preceding position $t - \Delta$. We then run generation with the intervened attention pattern and measure the resulting next-token prediction accuracy. For each condition, accuracy is evaluated over 10,000 sequences independently sampled from the grammar.

### C.3. Details of KL Distance Computation.

To quantitatively assess this agreement, we compute the position-wise average Kullback–Leibler (KL) divergence between the ground-truth distribution and the Transformer's predicted distribution, $\mathrm{KL}(p_\mathcal{G} \,||\, p_{\mathrm{Tr}})$. For comparison, we include a uniform random-guess baseline, $\mathrm{KL}(p_\mathcal{G} || p_{\mathrm{Rg}})$, which serves as a sanity check on the magnitude of the divergence.

Let $\mathcal{D}$ denote a finite test set of sequences sampled from a PCFG $\mathcal{G}$, where each sequence $x_{1:T} = (x_1, \ldots, x_T)$ has length $T$. Let $p_\mathcal{G}(\cdot \mid x_{1:t-1})$ be the grammar-induced next-token distribution computed by probabilistic Earley parsing, and $p_{\mathrm{Tr}}(\cdot \mid x_{1:t-1})$ be the Transformer prediction, $p_{\mathrm{Rg}}(\cdot \mid x_{1:t-1})$ be the Random Guess (uniform distribution).

$$\mathrm{KL}_{\mathrm{avg}}(p_\mathcal{G} \,||\, p_{\mathrm{alg}}) = \frac{1}{|\mathcal{D}|} \sum_{x_{1:T} \in \mathcal{D}} \frac{1}{T} \sum_{t=1}^{T} \sum_{a \in \Sigma} p_\mathrm{G}(a \mid x_{1:t-1}) \log \frac{p_\mathrm{G}(a \mid x_{1:t-1})}{p_{\mathrm{alg}}(a \mid x_{1:t-1})}$$

where alg can be random guess or Transformer. We compute these empirical results on 20000 different sequences for each grammar, and report the final number as the average KL divergence over 50 randomly sampled grammars.

Results are reported in Figure 10. Averaged over 50 random grammars, the Transformer achieves a substantially smaller KL divergence to the exact grammar-induced next-token distribution than the uniform random-guess baseline. Specifically, the average KL divergence is 0.0881 for the Transformer, compared with 0.5862 for the baseline, yielding about a 6.7× reduction. This indicates that the learned model does not merely outperform a naive predictor, but closely tracks the grammar-implied distribution across diverse random grammars.

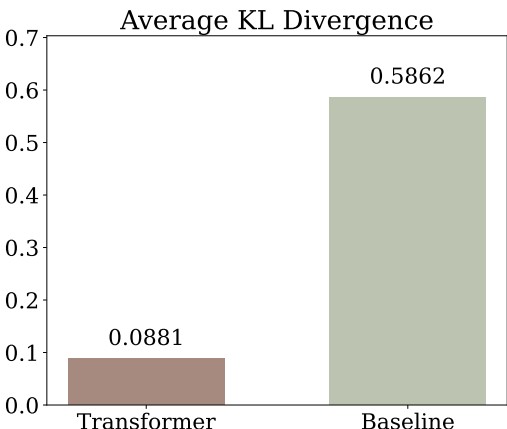

*Figure 10.* KL divergence comparisons of next-token prediction distributions on 50 sampled grammars. We primarily compare the Transformer against the Earley parser, with uniform prediction (random guess) included as a baseline. Lower values indicate closer agreement with the Earley distribution.

## C.4. Details of CYK Chart Probing

In Section 3.3, we employ the CYK algorithm to record completed constituents (i.e., recognized ancestors) at their corresponding ending positions. To apply CYK, we convert each grammar into Chomsky Normal Form (CNF) (Chomsky, 1956), which is equivalent to the original grammar but introduces an extended set of nonterminals $\tilde{\mathcal{N}}$, with $\mathcal{N} \subseteq \tilde{\mathcal{N}}$. For clarity, we suppress this distinction in the main text. All recognized ancestors therefore belong to $\tilde{\mathcal{N}}$, which does not affect any of our results.

## C.5. Additional Results of Comparison of Earley algorithm and Transformer

In Figure 11, we provide additional comparisons of next-token distributions between the exact Earley algorithm and the Transformer outputs for several sampled grammars from the 50-grammar pool (`CFG-2` to `CFG-5`). Figure 12 presents the same comparison for the binary grammar used in Section 4. Across these additional examples, the Transformer predictions remain closely aligned with the exact grammar-induced distributions, consistent with the agreement observed in Section 3.1.

## C.6. Additional Experiments Results for Probing

In Section 3, we report the probing results using an MLP probe. We include here a comparison between linear and MLP probes to clarify this design choice. Our probing targets are multi-label prediction tasks, so a linear probe may be too restrictive to reliably recover the relevant structured features from hidden states. We therefore use the MLP probe in the main text to provide sufficient expressivity, following prior probing work that also adopts nonlinear probes for structured targets. At the same time, the probe architecture is fixed in advance, and the train and test losses remain close at every layer, which indicates that the MLP probe generalizes well rather than merely overfitting.

Table 2 shows that both probe families recover the same qualitative layer-wise trend: for both CYK-father and Earley-state prediction, accuracy improves substantially from shallow to intermediate layers, consistent with the view that the Transformer progressively constructs hierarchical information across depth. However, the MLP probe is consistently stronger in absolute accuracy. For CYK-father probing, the MLP reaches 0.9124 and 0.9264 at Layers 3 and 4, compared with 0.8616 and 0.8802 for the linear probe; for Earley-state probing, the corresponding numbers are 0.8522 and 0.9361 for the MLP versus 0.7785 and 0.9025 for the linear probe. Thus, linear probes do not overturn the main conclusion, but they under-estimate how clearly the relevant parsing-style information is represented. We therefore use the MLP probe in the main experiments, while reporting the linear comparison here for completeness.

## C.7. Additional Probing Results for Balanced Full-Binary PCFGs

We also run the probing experiments on the Transformer trained on `CFG-6`, the balanced full-binary grammar. In this setting, the information of completed constituents is concentrated at dyadic boundaries, as shown by the construction in

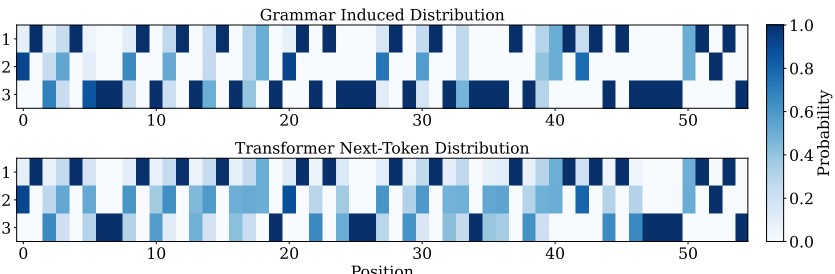

*(a)* Visualization for sampled example grammar `CFG-2`

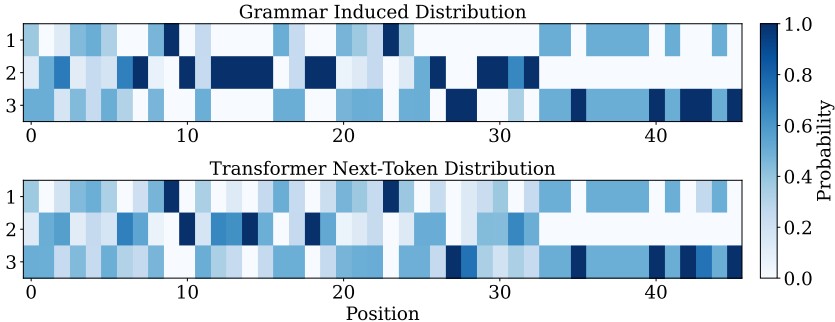

*(b)* Visualization for sampled example grammar `CFG-3`

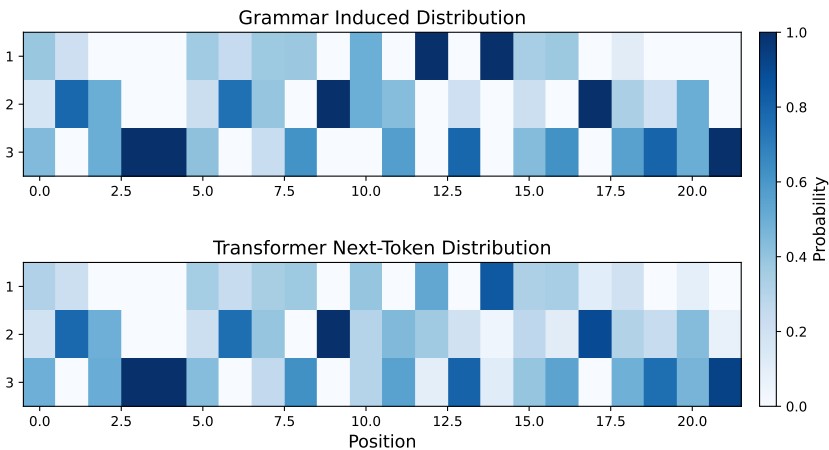

*(c)* Visualization for sampled example grammar `CFG-4`

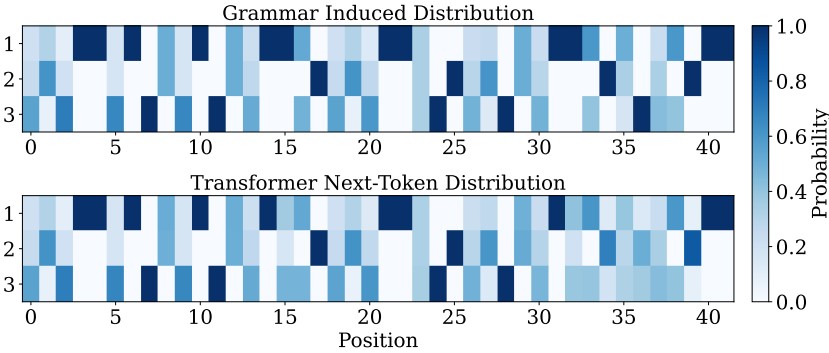

*(d)* Visualization for sampled example grammar `CFG-5`

*Figure 11.* Visualization of the next-token distribution at every position of sample sequences from different CFGs (2-5), comparing Earley algorithm and Transformer that trained on these grammars.

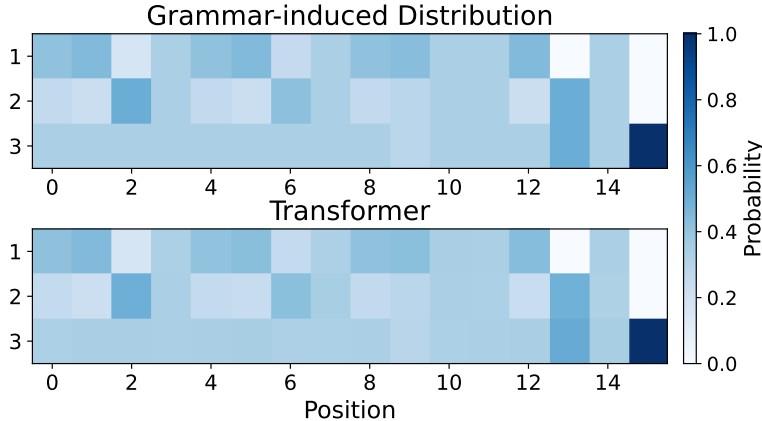

*Figure 12.* Visualization of the next-token distribution at every position of a sample sequence from CFG 6 (binary), comparing Earley algorithm and Transformer that learned this grammar.

*Table 2.* Average probing accuracy, train loss and test loss across 50 random CFG grammars with depth 4. Columns denote probe layers.

| Method | Metric | Layer 1 | Layer 2 | Layer 3 | Layer 4 | Layer 5 |
|---|---|---|---|---|---|---|
| CYK Father (MLP) | Accuracy | 0.6358 | 0.8404 | 0.9124 | 0.9264 | 0.8818 |
| | Train Loss | 0.0819 | 0.0146 | 0.0022 | 0.0014 | 0.0120 |
| | Test Loss | 0.0891 | 0.0282 | 0.0187 | 0.0178 | 0.0172 |
| CYK Father (Linear) | Accuracy | 0.5755 | 0.7817 | 0.8616 | 0.8802 | 0.7253 |
| | Train Loss | 0.1032 | 0.0350 | 0.0177 | 0.0145 | 0.0529 |
| | Test Loss | 0.1040 | 0.0358 | 0.0189 | 0.0160 | 0.0539 |
| Earley State (MLP) | Accuracy | 0.4931 | 0.7110 | 0.8522 | 0.9361 | 0.9274 |
| | Train Loss | 0.0718 | 0.0215 | 0.0038 | 0.0010 | 0.0026 |
| | Test Loss | 0.0794 | 0.0472 | 0.0400 | 0.0262 | 0.0073 |
| Earley State (Linear) | Accuracy | 0.4315 | 0.6343 | 0.7785 | 0.9025 | 0.8936 |
| | Train Loss | 0.0872 | 0.0427 | 0.0195 | 0.0053 | 0.0073 |
| | Test Loss | 0.0883 | 0.0446 | 0.0219 | 0.0076 | 0.0083 |

Section 4. We therefore probe CYK-style completed constituents only at those boundary positions. As in the main text, one may also separately probe CYK-father labels at these positions. However, we omit that additional panel here because it is already subsumed by the max-span probing results: the CYK-father task corresponds to the last column, i.e., the largest-span nonterminals, in the span-restricted probing heatmap.

Figure 13 shows a qualitatively similar pattern to the random-grammar results in Section 3. In the left panel, probing accuracy for Earley states improves substantially across the early and intermediate layers, indicating that the model progressively accumulates prefix-conditioned parsing information. In the right panel, probing accuracy for span-restricted completed constituents is uniformly high, but the same coarse layer-to-span alignment remains visible: shorter-span constituents are recovered reliably in shallower layers, while larger-span constituents are most clearly represented in deeper layers. Compared with the random-grammar family, however, the constituent probe is less diagnostically sharp here because the balanced full-binary grammar is more regular, making it easier for the probe to exploit boundary-local shortcuts.

## C.8. Additional Quantitative Results for Attention Pattern in Section 4

To complement the qualitative attention visualizations in Figure 5 in Section 4, we quantify how much attention mass in each layer falls inside the theory-predicted local window. For layer $k$, the construction predicts an effective window size of $2^k$, corresponding to window sizes $2, 4, 8, 16$ for Layers 1–4. Table 3 reports the average fraction of attention mass that remains inside this predicted window.

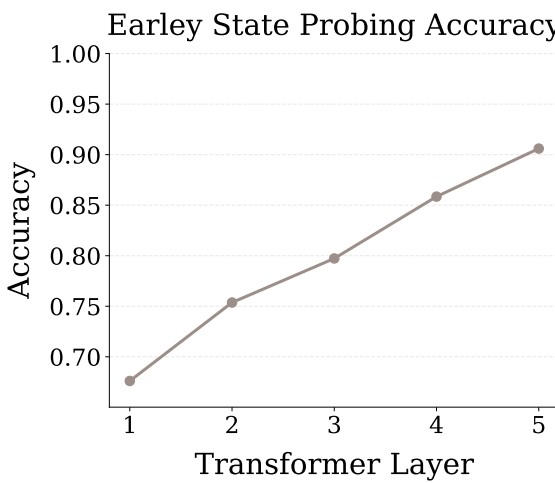
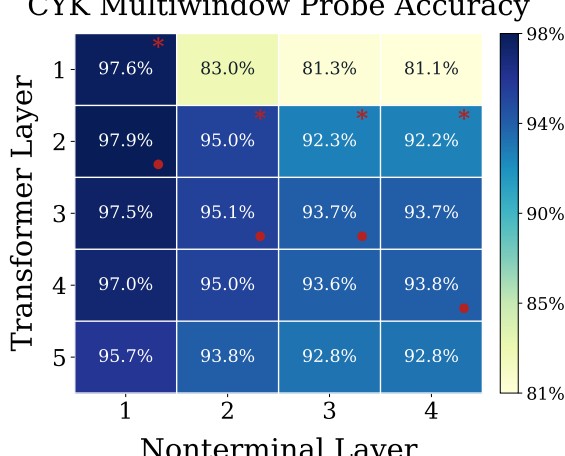

*Figure 13.* **Left.** Layer-wise probing accuracy of hidden states for Earley states across positions. **Right.** Layer-wise probing accuracy for completed constituents at different window sizes (max span of nonterminals in each layer) for CFG 6 (binary) used in experiments. ⋆ marks the first layer where accuracy exceeds 90%, and ● marks the layer with the highest accuracy.

| Layer | Window Size | Avg Attention Ratio |
|-------|-------------|---------------------|
| 1 | 2 | 0.6727 |
| 2 | 4 | 0.8123 |
| 3 | 8 | 0.9227 |
| 4 | 16 | 1.0000 |

*Table 3.* Average fraction of attention mass inside the theory-predicted window at each layer for the balanced full-binary grammar in Section 4.

The pattern is strongly consistent with the theoretical construction. Across all four layers, most of the attention mass falls inside the theory-predicted local window: $67\%$ in Layer 1, $81\%$ in Layer 2, and $92\%$ in Layer 3. In Layer 4, this ratio reaches $100\%$. This is expected because, at the final boundary layer in the balanced full-binary setting considered here, the longest relevant dyadic attention window has size 16, so the theory-predicted window already covers the full effective receptive field used by the construction. Overall, the trained Transformer places the large majority of its attention inside the windows prescribed by the local-to-global mechanism, providing quantitative support for the qualitative alignment in theoretical construction in Figure 4 and Theorem 4.2.

### C.9. Additional Results for Deeper Grammar

To test whether the local-to-global mechanism persists beyond the depth-4 setting used in the main experiments, we repeat the probing analysis on the depth-6 grammar shown in Figure 14. This grammar goes beyond the balanced binary setting of our construction: it includes ternary productions in addition to binary ones, and therefore provides a stricter test of whether the observed pattern is tied to binary structure. We again omit a separate CYK-father panel. As in the balanced full-binary case above, it is already covered by the max-span probing results, since probing the highest-layer nonterminals is equivalent to the last-column prediction task in the span-restricted heatmap.

Figure 15 shows that the same qualitative picture continues to hold. In the left panel, Earley-state probing accuracy improves across layers before peaking in the deeper part of the network, indicating that prefix-conditioned parsing information is again accumulated progressively rather than being present uniformly from the start. In the right panel, the span-restricted CYK-style probing exhibits the same depth-aligned pattern as in the depth-4 grammars: probes for smaller windows succeed earlier, whereas larger-window probes become accurate only in deeper layers. This indicates that longer-range structural dependencies are resolved later in the network.

Overall, the depth-6 experiment confirms that the local-to-global mechanism is not an artifact of the shallower depth-4 setup. Since this grammar includes ternary productions, the persistence of the same probing pattern also suggests that the mechanism extends beyond the binary structure covered by our explicit construction.

$$\text{Layer 6} \left\{ 22 \rightarrow 21\,20 \mid 20\,19 \right.$$

$$\text{Layer 5} \left\{ \begin{array}{l} 19 \rightarrow 16\,17\,18 \mid 17\,18\,16 \\ 20 \rightarrow 17\,16\,18 \mid 16\,17 \\ 21 \rightarrow 18\,16 \mid 16\,18\,17 \end{array} \right.$$

$$\text{Layer 4} \left\{ \begin{array}{l} 16 \rightarrow 15\,13 \mid 13\,15\,14 \\ 17 \rightarrow 14\,13\,15 \mid 15\,13\,14 \\ 18 \rightarrow 15\,14\,13 \mid 14\,13 \end{array} \right.$$

$$\text{Layer 3} \left\{ \begin{array}{l} 13 \rightarrow 11\,12 \mid 12\,11 \\ 14 \rightarrow 11\,10\,12 \mid 10\,11\,12 \\ 15 \rightarrow 12\,11\,10 \mid 11\,12\,10 \end{array} \right.$$

$$\text{Layer 2} \left\{ \begin{array}{l} 10 \rightarrow 7\,9\,8 \mid 9\,8\,7 \\ 11 \rightarrow 8\,7\,9 \mid 7\,8\,9 \\ 12 \rightarrow 8\,9\,7 \mid 9\,7\,8 \end{array} \right.$$

$$\text{Layer 1} \left\{ \begin{array}{l} 7 \rightarrow '3'\,'1' \mid '1'\,'2'\,'3' \\ 8 \rightarrow '3'\,'2' \mid '3'\,'1'\,'2' \\ 9 \rightarrow '3'\,'2'\,'1' \mid '2'\,'1' \end{array} \right.$$

*Figure 14.* The PCFG with depth 6 used in experiments of Table 9 and 10, where nonterminals are partitioned into six layers and each production expands symbols from one layer to the immediately lower layer. Each rule provides alternative expansions separated by |. The terminal alphabet is $\{1, 2, 3\}$.

## D. A Constructive Transformer for Balanced Full-Binary PCFGs

### D.1. Problem Setting: Balanced Full-Binary PCFGs

The Balanced Full-Binary PCFG is defined in Definition 4.1. Now we recall it here.

Let $G = (\mathcal{N}, \Sigma, \mathcal{R}, S, p)$ be a PCFG, where:

- Terminals $\Sigma$ and nonterminals $\mathcal{N}$, with a *layered* partition $\mathcal{N} = \sqcup_{k=1}^{K} \mathcal{N}_k$ and $\Sigma = \mathcal{N}_0$. Thus $\mathcal{N}_i \cap \mathcal{N}_j = \emptyset$ for $i \neq j$ and $\sqcup_{k=1}^{K} \mathcal{N}_k = \mathcal{N}$. We denote the number of nonterminals in layer $k$ as $n_k = |\mathcal{N}_k|$, and the number of terminals $n_0 = |\mathcal{N}_0|$.

- Binary rules $\mathcal{R} = \{C \rightarrow AB : C \in \mathcal{N}_k, \ A, B \in \mathcal{N}_{k-1}, \ k = 1, \ldots, K\}$ with probabilities $p(r)$. We write $\mathcal{R}_k := \{C \rightarrow AB : C \in \mathcal{N}_k, \ A, B \in \mathcal{N}_{k-1}\}$ for the rules at level $k$, and $\mathcal{R} = \sqcup_{k=1}^{K} \mathcal{R}_k$.

We assume the grammar is **balanced full-binary of depth** $K$:

- **Binary branching:** every rule is $C \rightarrow AB$.

- **Perfect balance:** every constituent at level $k$ spans length $2^k$.

- **Boundary alignment:** a level-$k$ span ends at $t$ iff $t \equiv 0 \pmod{2^k}$.

Thus for a boundary $t \equiv 0 \pmod{2^k}$, the unique split is fixed:

$$\text{left child ends at } t - 2^{k-1}, \qquad \text{right child ends at } t.$$

**Inside probabilities.** Let $\alpha_k(C, t)$ denote the inside probability that nonterminal $C$ generates the span $(t - 2^k + 1, \ldots, t)$ at level $k$. The (specialized) inside recursion is:

$$\alpha_k(C, t) = \sum_{(C \rightarrow AB) \in \mathcal{R}_k} p(C \rightarrow AB)\, \alpha_{k-1}(A, t - 2^{k-1})\, \alpha_{k-1}(B, t), \qquad t \equiv 0 \pmod{2^k}. \tag{2}$$

At non-boundary positions $t \not\equiv 0 \pmod{2^k}$, no level-$k$ constituent ends at $t$, so the model should behave as an identity map.

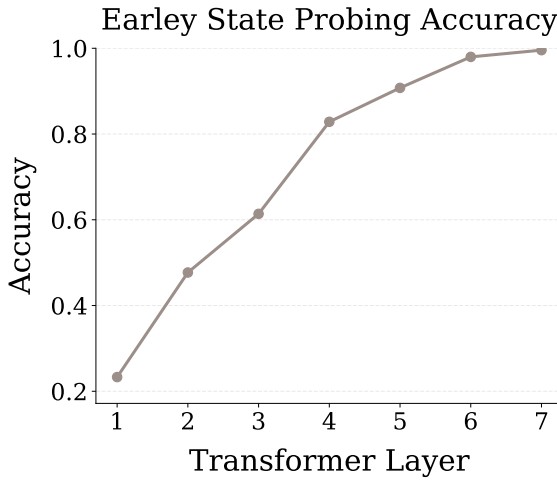
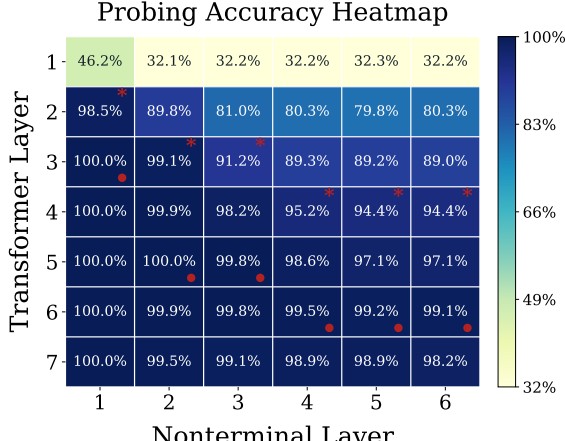

*Figure 15.* **Left.** Layer-wise probing accuracy of hidden states for Earley states across positions for the depth-6 grammar in Figure 14. **Right.** Layer-wise probing accuracy for completed constituents at different window sizes (max span of nonterminals in each layer) for the same grammar. ⋆ marks the first layer where accuracy exceeds 90%, and • marks the layer with the highest accuracy.

## D.2. Fixed-Dimension Representation and Invariants in First $K$ Layers

Let $n := |\mathcal{N} \cup \Sigma| = \sum_{k=0}^{K} n_k$ be the size of the symbol vocabulary and let $\ell := T + 1$ be the positional embedding dimension with one-hot vectors $e_t^{\text{pos}} \in \{0, 1\}^{\ell}$. We include the position $t = 0$ to represent `[BOS]`.

**Fixed model dimension.** We choose a constant $d_{\text{model}} := 3n + \ell$ for *all layers*. At each layer $k$, position $t$ holds:

$$h_t^{(k)} = \begin{bmatrix} x_t^{(k)} \\ u_{L,t}^{(k)} \\ u_{R,t}^{(k)} \\ e_t^{\text{pos}} \end{bmatrix} \in \mathbb{R}^{3n+\ell}, \tag{3}$$

where:

- $x_t^{(k)} \in \mathbb{R}^n$ is the **main symbol block** (stores inside probabilities);

- $u_{L,t}^{(k)}, u_{R,t}^{(k)} \in \mathbb{R}^n$ are **buffers** that temporarily store left/right child vectors;

- $e_t^{\text{pos}}$ is the one-hot positional encoding (invariant across layers).

**Notation for Index / Coordination** Let $\phi$ map symbols in $\mathcal{N} \cup \Sigma$ to indices in $\{1, \dots, n\}$. The type of notation: $\tilde{u}_L$-coord $\phi(A)$ denotes the global coordinate within the whole representation that corresponds to the local index $\phi(A)$ within block $\tilde{u}_L$.

**Initialization ($k = 0$).** Given input terminals $(w_1, \dots, w_T)$, set:

$$x_t^{(0)}[\phi(a)] = \mathbf{1}\{w_t = a\} \quad (a \in \Sigma), \qquad x_t^{(0)}[\phi(C)] = 0 \quad (C \in \mathcal{N}),$$

and initialize buffers to zero:

$$u_{L,t}^{(0)} = 0, \qquad u_{R,t}^{(0)} = 0,$$

Fix $e_t^{\text{pos}}$ as one-hot encoding for each position.

**Remark.** For $t = 0$, we have $x_0^{(0)} = \mathbf{0}$ and $e_0^{\text{pos}}[0] = 1$.

**Target invariant.** For each $k \geq 0$ and each boundary $t \equiv 0 \pmod{2^k}$,

$$x_t^{(k)}[\phi(C)] = \alpha_k(C, t) \quad \forall C \in \mathcal{N},$$

and for non-boundaries $t \not\equiv 0 \pmod{2^k}$,

$$x_t^{(k)} = x_t^{(k-1)}.$$

Moreover, at the *end* of each layer, buffers reset:

$$u_{L,t}^{(k)} = u_{R,t}^{(k)} = 0.$$

### D.3. Layer $k$: Two-Head Attention with Pure Positional QK

At each layer $k \in \{1, \ldots, K\}$ we use two heads $H \in \{L, R\}$. The heads are purely **structural routers**:

- head-$R$ routes the *right child* (always from position $t$);
- head-$L$ routes the *left child* (from $t - 2^{k-1}$ at boundaries; otherwise self).

Crucially, QK depends *only* on position and contains *no $I_\ell$* term.

#### D.3.1. QK DEFINITION

For each head $H \in \{L, R\}$, define:

$$W_{QK,H}^{(k)} = \begin{bmatrix} 0_{3n \times 3n} & 0 \\ 0 & U_H^{(k)} \end{bmatrix},$$

so the attention logit from query position $t$ to key position $u$ is:

$$l_{t,u,H}^{(k)} = \langle e_t^{\text{pos}}, U_H^{(k)} e_u^{\text{pos}} \rangle = U_H^{(k)}[t, u].$$

All symbol/buffer blocks are ignored in QK.

#### D.3.2. POSITIONAL MASKS

Fix a large $M \gg 1$.

**Head-$R$ mask (right child).**

$$U_R^{(k)}[t, u] = \begin{cases} 0, & u = t, \\ -M, & u \neq t. \end{cases}$$

Thus head-$R$ always selects $u = t$.

**Head-$L$ mask (left child at boundaries, otherwise self).**

$$U_L^{(k)}[t, u] = \begin{cases} 0, & t \equiv 0 \pmod{2^k} \text{ and } u = t - 2^{k-1} \text{ and } t \neq 0, \\ 0, & (t \not\equiv 0 \pmod{2^k} \text{ and } u = t) \text{ or } t = u = 0, \\ -M, & \text{otherwise.} \end{cases}$$

#### D.3.3. ATTENTION PROBABILITIES

Let

$$s_{t,u,H}^{(k)} = \text{softmax}_u([l_{t,0,H}^{(k)}, \cdots, l_{t,t,H}^{(k)}])[u].$$

Because each head has exactly one key with logit 0 and all others $-M$, in the limit $M \to \infty$ we obtain deterministic one-hot weights:

$$s_{t,u,R}^{(k)} = \begin{cases} 1, & u = t, \\ 0, & \text{otherwise,} \end{cases} \qquad s_{t,u,L}^{(k)} = \begin{cases} 1, & u = t - 2^{k-1}, \ t \equiv 0 \pmod{2^k} \text{ and } t \neq 0, \\ 1, & (u = t, \ t \not\equiv 0 \pmod{2^k}) \text{ or } t = u = 0, \\ 0, & \text{otherwise.} \end{cases}$$

## D.4. Value Maps: Routing Child Inside Vectors into Fixed Buffers

Recall the fixed block structure (Eq. (3)):

$$h_t^{(k)} = \begin{bmatrix} x_t^{(k)} \\ u_{L,t}^{(k)} \\ u_{R,t}^{(k)} \\ e_t^{\text{pos}} \end{bmatrix} \in \mathbb{R}^{3n+\ell}, \qquad x_t^{(k)}, u_{L,t}^{(k)}, u_{R,t}^{(k)} \in \mathbb{R}^n, \ e_t^{\text{pos}} \in \mathbb{R}^{\ell}.$$

In the attention stage of layer $k$, we want to *copy* the child inside vectors from the main block $x^{(k-1)}$ into two disjoint buffer subspaces:

$$a_t := x_{t-2^{k-1}}^{(k-1)} \text{ (left child)} \quad \mapsto \quad u_{L,t}, \qquad b_t := x_t^{(k-1)} \text{ (right child)} \quad \mapsto \quad u_{R,t},$$

while writing zeros into the other blocks. Importantly, this is done with *fixed $d_{\text{model}} = 3n + \ell$* and without mixing $a_t$ and $b_t$.

### D.4.1. VALUE MATRICES $W_{V,L}$ AND $W_{V,R}$

We now write the value maps explicitly as block matrices in $\mathbb{R}^{(3n+\ell)\times(3n+\ell)}$. Partition rows/columns according to the block order $(x, u_L, u_R, \text{pos})$.

**Head-$L$ value matrix.** Head-$L$ should take the input $x_u^{(k-1)}$ and write it into the $u_L$-*block* of the output:

$$W_{V,L} = \begin{bmatrix} \mathbf{0}_{n\times n} & \mathbf{0}_{n\times n} & \mathbf{0}_{n\times n} & \mathbf{0}_{n\times\ell} \\ I_n & \mathbf{0}_{n\times n} & \mathbf{0}_{n\times n} & \mathbf{0}_{n\times\ell} \\ \mathbf{0}_{n\times n} & \mathbf{0}_{n\times n} & \mathbf{0}_{n\times n} & \mathbf{0}_{n\times\ell} \\ \mathbf{0}_{\ell\times n} & \mathbf{0}_{\ell\times n} & \mathbf{0}_{\ell\times n} & \mathbf{0}_{\ell\times\ell} \end{bmatrix}.$$

Thus, for any input state $h_u^{(k-1)} = [x_u^{(k-1)}; u_{L,u}^{(k-1)}; u_{R,u}^{(k-1)}; e_u^{\text{pos}}]$,

$$W_{V,L} h_u^{(k-1)} = \begin{bmatrix} 0 \\ x_u^{(k-1)} \\ 0 \\ 0 \end{bmatrix}.$$

**Head-$R$ value matrix.** Head-$R$ should write the input $x_u^{(k-1)}$ into the $u_R$-*block*:

$$W_{V,R} = \begin{bmatrix} \mathbf{0}_{n\times n} & \mathbf{0}_{n\times n} & \mathbf{0}_{n\times n} & \mathbf{0}_{n\times\ell} \\ \mathbf{0}_{n\times n} & \mathbf{0}_{n\times n} & \mathbf{0}_{n\times n} & \mathbf{0}_{n\times\ell} \\ I_n & \mathbf{0}_{n\times n} & \mathbf{0}_{n\times n} & \mathbf{0}_{n\times\ell} \\ \mathbf{0}_{\ell\times n} & \mathbf{0}_{\ell\times n} & \mathbf{0}_{\ell\times n} & \mathbf{0}_{\ell\times\ell} \end{bmatrix},$$

so that

$$W_{V,R} h_u^{(k-1)} = \begin{bmatrix} 0 \\ 0 \\ x_u^{(k-1)} \\ 0 \end{bmatrix}.$$

### D.4.2. PER-HEAD ATTENTION OUTPUTS

Define each head output by the standard value aggregation:

$$o_{t,H}^{(k)} := \sum_{u=1}^{T} s_{t,u,H}^{(k)} W_{V,H} h_u^{(k-1)}, \qquad H \in \{L, R\},$$

where the attention weights $s_{t,u,H}^{(k)}$ are the deterministic one-hot distributions from the pure positional QK masks in the previous section.

**Boundary case ($t \equiv 0 \pmod{2^k}$ and $t \neq 0$).** Head-$L$ attends to $u = t - 2^{k-1}$ and head-$R$ attends to $u = t$, hence

$$o_{t,L}^{(k)} = \begin{bmatrix} 0 \\ x_{t-2^{k-1}}^{(k-1)} \\ 0 \\ 0 \end{bmatrix}, \qquad o_{t,R}^{(k)} = \begin{bmatrix} 0 \\ 0 \\ x_t^{(k-1)} \\ 0 \end{bmatrix}.$$

Summing heads gives the attention output:

$$\text{AttnOut}_t^{(k)} := o_{t,L}^{(k)} + o_{t,R}^{(k)} = \begin{bmatrix} 0 \\ a_t \\ b_t \\ 0 \end{bmatrix}, \quad a_t = x_{t-2^{k-1}}^{(k-1)}, \ b_t = x_t^{(k-1)}.$$

Crucially, $a_t$ and $b_t$ are stored in disjoint subspaces ($u_L$ vs. $u_R$), so no linear mixing occurs in the attention stage.

**Non-boundary case ($t \not\equiv 0 \pmod{2^k}$ or $t = 0$).** Both heads fall back to self, so

$$\text{AttnOut}_t^{(k)} = \begin{bmatrix} 0 \\ x_t^{(k-1)} \\ x_t^{(k-1)} \\ 0 \end{bmatrix}.$$

The subsequent MLP uses positional gating to ensure that no level-$k$ parent is created at non-boundary positions (i.e., it outputs $\Delta x_t^{(k)} = 0$ there) and resets the buffers.

$$\tilde{h}_t^{(k)} = h_t^{k-1} + \text{AttnOut}_t^{(k)} = \begin{bmatrix} x_t^{(k-1)} \\ \tilde{u}_{L,t}^{(k)} \\ \tilde{u}_{R,t}^{(k)} \\ e_t^{\text{pos}} \end{bmatrix} \in \mathbb{R}^{3n+\ell}.$$

### D.5. MLP Construction: Inside Update + Buffer Reset with Fixed Dimension

In this section we give an *explicit* two-layer MLP construction that simultaneously:

1. computes the inside update $\Delta x_t^{(k)}$ at level-$k$ boundaries;

2. outputs $-\tilde{u}_{L,t}^{(k)}$ and $-\tilde{u}_{R,t}^{(k)}$ to *reset* both buffers via the residual connection;

3. outputs 0 on the positional block.

Crucially, this MLP has *fixed* input/output dimension equal to that of $h_t$ (no runtime change in hidden-state dimension).

D.5.1. INPUTS AND DESIRED OUTPUTS

Recall that after attention and its residual, the layer-$k$ pre-MLP state is

$$\tilde{h}_t^{(k)} = \begin{bmatrix} x_t^{(k-1)} \\ \tilde{u}_{L,t}^{(k)} \\ \tilde{u}_{R,t}^{(k)} \\ e_t^{\text{pos}} \end{bmatrix} \in \mathbb{R}^{3n+\ell}.$$

At a boundary $t \equiv 0 \pmod{2^k}, t \neq 0$ we have

$$\tilde{u}_{L,t}^{(k)} = x_{t-2^{k-1}}^{(k-1)}, \qquad \tilde{u}_{R,t}^{(k)} = x_t^{(k-1)}.$$

We want an MLP output of the form

$$\mathrm{MLP}^{(k)}(\tilde{h}_t^{(k)}) = \begin{bmatrix} \Delta x_t^{(k)} \\ -\tilde{u}_{L,t}^{(k)} \\ -\tilde{u}_{R,t}^{(k)} \\ 0 \end{bmatrix},$$

where $\Delta x_t^{(k)} \in \mathbb{R}^n$ satisfies the inside recursion on boundaries and is 0 off boundaries:

$$\Delta x_t^{(k)}[\phi(C)] = g_k(t) \cdot \sum_{(C \to AB) \in \mathcal{R}_k} p(C \to AB) \, \tilde{u}_{L,t}^{(k)}[\phi(A)] \, \tilde{u}_{R,t}^{(k)}[\phi(B)], \qquad \forall C \in \mathcal{N}.$$

Here $g_k(t) \in \{0,1\}$ is the boundary indicator

$$g_k(t) = \begin{cases} 1, & t \equiv 0 \pmod{2^k} \text{ and } t \neq 0, \\ 0, & \text{otherwise.} \end{cases}$$

Because $e_t^{\mathrm{pos}}$ is one-hot, $g_k(t)$ is a linear functional of position:

$$g_k(t) = \langle w_k, e_t^{\mathrm{pos}} \rangle, \qquad w_k[j] = \mathbf{1}\{j \equiv 0 \pmod{2^k} \text{ and } j \neq 0\}.$$

### D.5.2. MLP ARCHITECTURE AND ACTIVATION

We use a two-layer MLP with elementwise square activation $\sigma(z) = z^2$:

$$\mathrm{MLP}^{(k)}(\tilde{h}) = W_2^{(k)} \, \sigma\big(W_1^{(k)} \tilde{h} + b_1^{(k)}\big) + b_2^{(k)}.$$

We explicitly construct the parameters by partitioning the hidden units into *four* groups:

$$\text{hidden} = \underbrace{\mathcal{H}_{\mathrm{prod}}}_{\text{products } \tilde{u}_L[A]\tilde{u}_R[B]} \cup \underbrace{\mathcal{H}_L}_{\text{reset } -\tilde{u}_L} \cup \underbrace{\mathcal{H}_R}_{\text{reset } -\tilde{u}_R} \cup \underbrace{\mathcal{H}_{\mathrm{gate}}}_{\text{boundary gate } g_k(t)} .$$

Each group serves a distinct role, and no hidden unit is reused across groups.

### D.5.3. GROUP I: EXPLICIT PRODUCT FEATURES VIA $(x \pm y)^2$

Fix $(A, B) \in \mathcal{N} \times \mathcal{N}$. Define two hidden neurons:

$$h_{A,B}^+(t) := \left(\tilde{u}_{L,t}^{(k)}[\phi(A)] + \tilde{u}_{R,t}^{(k)}[\phi(B)]\right)^2, \qquad h_{A,B}^-(t) := \left(\tilde{u}_{L,t}^{(k)}[\phi(A)] - \tilde{u}_{R,t}^{(k)}[\phi(B)]\right)^2. \tag{4}$$

**How $W_1$ realizes (4).** Let $\tilde{h} = [x; \tilde{u}_L; \tilde{u}_R; e^{\mathrm{pos}}]$. For each pair $(A, B)$ we add *two rows* to $W_1^{(k)}$:

- A "plus" row $r_{A,B}^+$ with exactly two nonzero entries:

$$r_{A,B}^+[\tilde{u}_L\text{-coord } \phi(A)] = 1, \qquad r_{A,B}^+[\tilde{u}_R\text{-coord } \phi(B)] = 1.$$

  Then $(r_{A,B}^+ \cdot \tilde{h}) = \tilde{u}_L[\phi(A)] + \tilde{u}_R[\phi(B)]$ and after $\sigma$ we get $h_{A,B}^+$.

- A "minus" row $r_{A,B}^-$ with:

$$r_{A,B}^-[\tilde{u}_L\text{-coord } \phi(A)] = 1, \qquad r_{A,B}^-[\tilde{u}_R\text{-coord } \phi(B)] = -1.$$

  Then $(r_{A,B}^- \cdot \tilde{h}) = \tilde{u}_L[\phi(A)] - \tilde{u}_R[\phi(B)]$ and after $\sigma$ we get $h_{A,B}^-$.

All other coordinates of these rows are 0, and we set the corresponding entries of $b_1^{(k)}$ to 0.

**Exact product extraction.** Using the identity

$$xy = \frac{1}{4}\big((x+y)^2 - (x-y)^2\big), \tag{5}$$

we define the product feature

$$m_{A,B}(t) := \frac{1}{4}\Big(h_{A,B}^+(t) - h_{A,B}^-(t)\Big) = \tilde{u}_{L,t}^{(k)}[\phi(A)] \cdot \tilde{u}_{R,t}^{(k)}[\phi(B)].$$

D.5.4. GROUP II/III: EXPLICIT RESET FEATURES FOR $-\tilde{u}_L$ AND $-\tilde{u}_R$ USING SQUARE ACTIVATION

To reset buffers we need to output $-\tilde{u}_L$ and $-\tilde{u}_R$. Since $\sigma(z) = z^2$ is even, we cannot obtain $-x$ directly from $x^2$ without a bias trick. We therefore use the exact identity (valid for all real $x$):

$$x = \frac{(x+1)^2 - x^2 - 1}{2}.$$

Thus

$$-x = -\frac{(x+1)^2 - x^2 - 1}{2}.$$

**Reset hidden neurons for each coordinate of $\tilde{u}_L, \tilde{u}_R$.** For each $i \in \{1, \dots, n\}$, $H \in \{L, R\}$, introduce two hidden neurons:

$$q_{i,+}^{(H)}(t) := \big(\tilde{u}_{H,t}^{(k)}[i] + 1\big)^2, \qquad q_{i,0}^{(H)}(t) := \big(\tilde{u}_{H,t}^{(k)}[i]\big)^2.$$

These are realized by two rows in $W_1^{(k)}$:

- row selects $\tilde{u}_H[i]$ with bias $+1$ (i.e., $b_1 = 1$ for that row);

- row selects $\tilde{u}_H[i]$ with bias $0$.

Then the linear combination

$$-\tilde{u}_{H,t}^{(k)}[i] = -\frac{q_{i,+}^{(H)}(t) - q_{i,0}^{(H)}(t) - 1}{2}.$$

D.5.5. GROUP IV: BOUNDARY GATING VIA POSITIONAL HIDDEN UNITS

We now make explicit how the positional encoding $e_t^{\text{pos}}$ controls whether the inside aggregation is written to $\Delta x_t^{(k)}$.

**Construction of a gating hidden unit.** We add a single hidden neuron whose pre-activation is

$$z_t^{\text{gate}} := \sum_{j \in \mathcal{B}_k} e_t^{\text{pos}}[j], \qquad \mathcal{B}_k := \{j : j \equiv 0 \pmod{2^k} \text{ and } j \neq 0\}.$$

This is realized by one row of $W_1^{(k)}$ with coefficients $1$ on boundary position indices and $0$ elsewhere, and bias $0$.

Since $e_t^{\text{pos}}$ is one-hot, we have

$$z_t^{\text{gate}} \in \{0, 1\}, \qquad z_t^{\text{gate}} = g_k(t).$$

Applying $\sigma(z) = z^2$ leaves it unchanged:

$$h_t^{\text{gate}} := \sigma(z_t^{\text{gate}}) = g_k(t).$$

**Role of the gate.** The gate $h_t^{\text{gate}}$ depends *only* on position and is independent of symbol values. All product features $m_{A,B}(t)$ are computed regardless of position; the gate determines whether these features are written to the symbol block.

D.5.6. OUTPUT LAYER $W_2$: AGGREGATION, RESET, AND GATING

The output matrix $W_2^{(k)}$ is defined so that:

- the symbol block $\Delta x_t^{(k)}$ is obtained by aggregating product features and applying the gate $h_t^{\text{gate}}$;

- the buffer blocks output $-\tilde{u}_{L,t}^{(k)}$ and $-\tilde{u}_{R,t}^{(k)}$;

- the positional block outputs $0$.

**Gated inside aggregation.** For each $C \in \mathcal{N}$, define the ungated sum

$$s_C^{(k)}(t) := \sum_{(C \to AB) \in \mathcal{R}_k} p(C \to AB)\, m_{A,B}(t).$$

The $\phi(C)$-th output coordinate is set to

$$\Delta x_t^{(k)}[\phi(C)] = h_t^{\text{gate}} \cdot s_C^{(k)}(t).$$

Since $h_t^{\text{gate}} \in \{0, 1\}$, this operation is purely selective: it passes $s_C(t)$ unchanged at boundary positions and suppresses it elsewhere.

**Buffer reset and positional outputs.** The remaining outputs are as in Groups II/III:

$$\left[ \text{MLP}^{(k)}(\tilde{h}_t^{(k)}) \right]_{u_L} = -\tilde{u}_{L,t}^{(k)}, \qquad \left[ \text{MLP}^{(k)}(\tilde{h}_t^{(k)}) \right]_{u_R} = -\tilde{u}_{R,t}^{(k)},$$

and the positional block outputs $0$.

D.5.7. SUMMARY OF SECTION D.5

The hidden layer consists of:

- Group I: product units $h_{A,B}^+, h_{A,B}^-$ yielding $m_{A,B}(t) = \tilde{u}_L[A]\tilde{u}_R[B]$;

- Groups II/III: reset units yielding $-\tilde{u}_L$ and $-\tilde{u}_R$;

- Group IV: a single positional gating unit $h_t^{\text{gate}} = g_k(t)$.

The output layer linearly combines these units so that:

$$\text{MLP}^{(k)}(\tilde{h}_t^{(k)}) = \begin{bmatrix} \Delta x_t^{(k)} \\ -\tilde{u}_{L,t}^{(k)} \\ -\tilde{u}_{R,t}^{(k)} \\ 0 \end{bmatrix},$$

thereby completing the inside update, enforcing boundary constraints, and resetting buffers, all within a fixed-dimensional MLP.

**Design rationale.** Attention writes child representations into buffers additively. Since Transformer layers do not permit in-place overwriting, the only way to ensure that these buffers are ephemeral is to output their exact negatives in the subsequent MLP. The explicit boundary gate guarantees that inside probabilities are created *only* at valid level-$k$ boundaries.

**D.6. Inductive Proof of Exact Inside Simulation**

We restate and prove the main claim in a form aligned with the explicit Section D.5 construction (Groups I–IV).

D.6.1. STATEMENT OF THE INVARIANT

Let the layer-$k$ hidden state at position $t$ be

$$h_t^{(k)} = \begin{bmatrix} x_t^{(k)} \\ u_{L,t}^{(k)} \\ u_{R,t}^{(k)} \\ e_t^{\mathrm{pos}} \end{bmatrix} \in \mathbb{R}^{3n+\ell},$$

where $x_t^{(k)} \in \mathbb{R}^n$ is the *symbol/inside block*, $u_{L,t}^{(k)}, u_{R,t}^{(k)} \in \mathbb{R}^n$ are *left/right buffers*, and $e_t^{\mathrm{pos}} \in \{0,1\}^\ell$ is the one-hot positional block.

For each level $k \geq 0$, define the boundary indicator

$$g_k(t) = \mathbf{1}\{t \equiv 0 \pmod{2^k} \text{ and } t \neq 0\},$$

and let $\alpha_k(C, t)$ denote the inside probability that nonterminal $C$ spans the length-$2^k$ segment ending at $t$, i.e. $(t - 2^k + 1, \ldots, t)$.

**Theorem D.1** (Exact inside recursion with vanishing buffers). *For every $k = 0, 1, \ldots, K$ and every position $t$:*

(I1) **Positional invariance:** *the positional block is unchanged,*

$$h_t^{(k)}[\mathrm{pos}] = e_t^{\mathrm{pos}}.$$

(I2) **Buffer reset:** *after the MLP and residual, both buffers vanish,*

$$u_{L,t}^{(k)} = 0, \qquad u_{R,t}^{(k)} = 0.$$

(I3) **Inside semantics on boundaries:** *at boundary positions $t \equiv 0 \pmod{2^k}$, the symbol block equals the level-$k$ inside distribution:*

$$x_t^{(k)}[\phi(C)] = \alpha_k(C, t) \qquad \forall C \in \mathcal{N}.$$

*At non-boundary positions ($g_k(t) = 0$), no new level-$k$ constituent is created and the update for level $k$ is suppressed.*

*Proof.* We prove the three invariants by induction on the layer index $k$.

**Base case ($k = 0$).** At layer 0, the model is initialized with

$$h_t^{(0)} = \begin{bmatrix} x_t^{(0)} \\ 0 \\ 0 \\ e_t^{\mathrm{pos}} \end{bmatrix},$$

where $x_t^{(0)}$ encodes the leaf-level inside distribution (e.g. a one-hot vector on the observed terminal, or the corresponding preterminal distribution). Thus (I1) holds by construction and (I2) holds because the buffers are initialized to 0. Moreover, the leaf inside recursion is satisfied by definition of $x_t^{(0)}$, so (I3) holds for $k = 0$.

**Inductive step.** Assume (I1)–(I3) hold for layer $k - 1$ at all positions. We prove they hold for layer $k$.

Fix a position $t$. Consider the attention sublayer of layer $k$ (two heads $L$ and $R$) with the positional QK masks from the previous section and the explicit value maps. By the construction of the QK masks:

- If $t \equiv 0 \pmod{2^k}$ and $t \neq 0$ (a level-$k$ boundary), then head-$L$ attends to $t - 2^{k-1}$ and head-$R$ attends to $t$.

- If $t \not\equiv 0 \pmod{2^k}$ or $t = 0$ (non-boundary), then both heads attend to $t$ (self).

Moreover, by the value matrices designed in Section D.4, head-$L$ writes the attended token's $x$-block into the $u_L$ buffer and head-$R$ writes it into the $u_R$ buffer, while outputting $0$ on the $x$ and pos blocks. Hence, the *attention output* at position $t$ has the form

$$
\text{AttnOut}_t^{(k)} = \begin{bmatrix} 0 \\ \star \\ \star \\ 0 \end{bmatrix}.
$$

After adding the attention residual, the pre-MLP state becomes

$$
\tilde{h}_t^{(k)} = h_t^{(k-1)} + \text{AttnOut}_t^{(k)} = \begin{bmatrix} x_t^{(k-1)} \\ \tilde{u}_{L,t}^{(k)} \\ \tilde{u}_{R,t}^{(k)} \\ e_t^{\text{pos}} \end{bmatrix}.
$$

We now analyze boundary vs. non-boundary cases.

**Case 1:** $t \equiv 0 \pmod{2^k}$ **and** $t \neq 0$ **(boundary).** By the attention routing above,

$$
\tilde{u}_{L,t}^{(k)} = x_{t-2^{k-1}}^{(k-1)}, \qquad \tilde{u}_{R,t}^{(k)} = x_t^{(k-1)}.
$$

By the induction hypothesis (I3) applied at level $k-1$ and at the two child boundaries, the child symbol blocks encode the level-$(k-1)$ inside distributions:

$$
\tilde{u}_{L,t}^{(k)}[\phi(A)] = \alpha_{k-1}(A, t - 2^{k-1}), \qquad \tilde{u}_{R,t}^{(k)}[\phi(B)] = \alpha_{k-1}(B, t), \qquad \forall A, B \in \mathcal{N}_{k-1}.
$$

Now apply the explicit MLP construction in Section D.5:

- **Group I** forms product features $m_{A,B}(t) = \tilde{u}_{L,t}^{(k)}[\phi(A)]\tilde{u}_{R,t}^{(k)}[\phi(B)]$ exactly via $(x \pm y)^2$.

- **Group IV** forms a hidden gating unit $h_t^{\text{gate}} = g_k(t) = 1$ from $e_t^{\text{pos}}$.

- The output layer aggregates these to produce the symbol update

$$
\Delta x_t^{(k)}[\phi(C)] = h_t^{\text{gate}} \cdot \sum_{(C \to AB) \in \mathcal{R}_k} p(C \to AB)\, m_{A,B}(t) = \sum_{(C \to AB) \in \mathcal{R}_k} p(C \to AB)\, \tilde{u}_{L,t}^{(k)}[\phi(A)]\, \tilde{u}_{R,t}^{(k)}[\phi(B)].
$$

  Substituting the induction hypothesis for $\tilde{u}_{L,t}^{(k)}$ and $\tilde{u}_{R,t}^{(k)}$ yields

$$
\Delta x_t^{(k)}[\phi(C)] = \sum_{(C \to AB) \in \mathcal{R}_k} p(C \to AB)\, \alpha_{k-1}(A, t - 2^{k-1})\, \alpha_{k-1}(B, t) = \alpha_k(C, t),
$$

  which is exactly the inside recursion for the fixed split point of the balanced grammar. The last equality holds because of Eq. (2).

- **Groups II/III** output $-\tilde{u}_{L,t}^{(k)}$ and $-\tilde{u}_{R,t}^{(k)}$ exactly (using the square-activation bias trick), and the positional output is $0$.

Therefore the MLP output satisfies

$$
\text{MLP}^{(k)}(\tilde{h}_t^{(k)}) = \begin{bmatrix} \Delta x_t^{(k)} \\ -\tilde{u}_{L,t}^{(k)} \\ -\tilde{u}_{R,t}^{(k)} \\ 0 \end{bmatrix}.
$$

Applying the MLP residual gives the post-MLP state

$$
h_t^{(k)} = \tilde{h}_t^{(k)} + \text{MLP}^{(k)}(\tilde{h}_t^{(k)}) = \begin{bmatrix} x_t^{(k-1)} + \Delta x_t^{(k)} \\ 0 \\ 0 \\ e_t^{\text{pos}} \end{bmatrix}.
$$

Thus (I1) and (I2) hold at layer $k$ on boundaries. Moreover, the increment satisfies $\Delta x_t^{(k)}[\phi(C)] = \alpha_k(C, t)$, so the symbol block encodes the level-$k$ inside distribution as required (I3).

**Case 2:** $t \not\equiv 0 \pmod{2^k}$ **or** $t = 0$ **(non-boundary).** The attention routing reduces to self-attention, hence $\tilde{u}_{L,t}^{(k)}$ and $\tilde{u}_{R,t}^{(k)}$ are copies of $x_t^{(k-1)}$ (up to the fixed buffer routing). However, Group IV again produces $h_t^{\text{gate}} = g_k(t) = 0$ from $e_t^{\text{pos}}$. Therefore, the symbol update is suppressed:

$$\Delta x_t^{(k)} = 0.$$

Groups II/III still output $-\tilde{u}_{L,t}^{(k)}$ and $-\tilde{u}_{R,t}^{(k)}$, so the buffers are reset after the MLP residual, and the pos block stays unchanged. Hence (I1) and (I2) hold, and (I3) holds trivially since no level-$k$ constituent should be created off boundaries.

Since both cases satisfy (I1)–(I3), the induction closes. $\qquad\square$

## D.7. The Hidden State after the First $K$ Layers

After the first $K$ layers, we have

$$h_t^{(K)} = \begin{bmatrix} x_t^{(K)} \\ 0 \\ 0 \\ e_t^{\text{pos}} \end{bmatrix} \in \mathbb{R}^{3n+\ell},$$

For $t = 0$, the first three blocks are initialized to $\mathbf{0}$ and remain so after the first $K$ layers, hence $h_0^{(K)} = [\mathbf{0}; \mathbf{0}; \mathbf{0}; e_0^{\text{pos}}]$.

## D.8. Final Layer: Next-Token Prediction via Canonical Earley Rollout

This section constructs the final prediction module that maps a prefix $x_{1:t}$ to the next-token distribution $p(x_{t+1} \mid x_{1:t})$. The module *does not* re-parse the prefix. Instead it: (i) extracts a canonical sequence of *maximal completion summaries* from the prefix using deterministic attention, and (ii) advances a probabilistic Earley state by consuming these summaries in order. Balanced full-binary structure yields a unique dyadic consumption order, so no backtracking is needed.

### D.8.1. WHAT IS AVAILABLE AFTER $K$ LAYERS: PACKED COMPLETION VECTORS

After the first $K$ layers, each position $j \in \{0, 1, \ldots, T\}$ carries

$$x_j^{(K)} \in \mathbb{R}^n.$$

We interpret $x_j^{(K)}$ as a *packed completion representation*:

$$x_j^{(K)} = \begin{bmatrix} c_{j,0}, \, c_{j,1}, \, \ldots, \, c_{j,K} \end{bmatrix}, \qquad c_{j,k} \in \mathbb{R}^{n_k}, \qquad n = \sum_{k=0}^{K} n_k \tag{6}$$

For $j \geq 1$, the block $c_{j,k}$ is nonzero iff $j \equiv 0 \pmod{2^k}$, and $c_{j,k}[\phi(B)]$ equals the total probability mass of completing a level-$k$ constituent rooted at nonterminal $B$ whose right boundary is $j$ (the span is uniquely determined by balance and boundary alignment). For $j = 0$ we set $x_0^{(K)} = \mathbf{0}$. Under this representation, $c_{j,k}$ is the complete constituents (inside probabilities) at position $j$ of the symbol in layer $k$, i.e. $\mathcal{N}_k$.

### D.8.2. HIGHEST-LEVEL COMPLETION SUMMARY AT A BOUNDARY

In this section, we identify the specific information required to map completed constituents directly to Earley states without invoking the COMPLETE operation described in Lemma D.2. We then provide a formal proof for this conversion.

**Lemma D.2** (Canonical Dyadic Consumption). *Let $\mathcal{G}$ be a balanced full-binary PCFG and let $t$ be a prefix length with binary expansion $t = \sum_{r=1}^{m(t)} 2^{\beta_r}$, $\beta_1 > \cdots > \beta_{m(t)} \geq 0$. Define the dyadic boundary chain $b_0(t) = 0$, $b_r(t) = \sum_{q=1}^{r} 2^{\beta_q}$. If the inside probabilities of the dyadic blocks $[\, b_{r-1}(t) + 1, \, b_r(t) \,]$ are available at positions $b_r(t)$, then the Earley state at position $t$ can be obtained by consuming these blocks in decreasing span order using only PREDICT and SCAN, without any COMPLETE operation.*

To ensure coherence and clarity, we maintain the notation established in the previous sections throughout our proof. For any boundary $j \geq 1$, define its 2-adic valuation

$$\nu(j) := \max\{k \geq 0 : 2^k \mid j\}. \tag{7}$$

If $j = 0$, $\nu(j) = 0$. Then $\nu(j)$ is exactly the *highest level* at which a constituent can complete at $j$. We define the *maximal completion summary* at $j$ as the vector

$$u(j) := S_{\nu(j)}\, x_j^{(K)} \in \mathbb{R}^n, \tag{8}$$

which keeps only the block $c_{j,\nu(j)}$ and zeroes all other blocks. The Thus $u(j)[\phi(B)]$ is the probability mass of completing the maximal constituent rooted at $B$ whose right boundary is $j$.

**Lemma D.3** (Maximal completion suffices). *Fix a prefix index $t$ and its dyadic boundary chain $\{b_r(t)\}_{r=0}^{R}$ (where $b_0(t) = 0 < b_1(t) < \cdots < b_R(t) = t$). For each boundary $j = b_r(t)$, consuming only the* maximal completion summary *$u(j)$ is sufficient to advance a probabilistic Earley state from $b_{r-1}(t)$ to $b_r(t)$. In particular, no backtracking over lower-level completions ending at the same $j$ is required.*

*Proof.* We spell out the key geometric fact behind the lemma and then connect it to the Earley consumption step.

**Dyadic spans ending at a fixed boundary are nested.** Fix a boundary position $j \geq 1$. Let $\nu(j)$ denote the 2-adic valuation of $j$, i.e., $\nu(j) = \max\{m \geq 0 : 2^m \mid j\}$. In a balanced dyadic (full-binary) decomposition, the canonical dyadic spans ending at $j$ are

$$I_k(j) := [j - 2^k + 1,\ j], \qquad k = 0, 1, \ldots, \nu(j). \tag{9}$$

These spans form a strictly nested chain:

$$I_0(j) \subset I_1(j) \subset \cdots \subset I_{\nu(j)}(j). \tag{10}$$

We call $I_{\nu(j)}(j)$ the *maximal dyadic span ending at $j$*.

**What $u(j)$ summarizes.** Let $u(j)$ denote the *completion summary* associated with the maximal span $I_{\nu(j)}(j)$ (ending at $j$). Conceptually, $u(j)$ is the inside-style aggregate over *all* derivations that produce exactly the substring on that maximal span: it sums (or log-sum-exp's) the probabilities of all parse fragments whose yield is $x_{I_{\nu(j)}(j)}$ and whose root nonterminal matches the completion being performed. Thus $u(j)$ marginalizes out *every* internal choice made strictly inside $I_{\nu(j)}(j)$.

**Lower-level completions are internal to the maximal span.** Any other completion that ends at the same boundary $j$ but at a lower level $k < \nu(j)$ corresponds to the strict subspan $I_k(j)$. By (10), this subspan lies entirely inside the maximal span:

$$I_k(j) \subsetneq I_{\nu(j)}(j).$$

Hence, any derivational alternative that completes on $I_k(j)$ is an *internal* event within the larger parse fragment over $I_{\nu(j)}(j)$. When we compute the maximal completion summary $u(j)$ for $I_{\nu(j)}(j)$, all such internal completions (and the branching choices they enable) are already accounted for via the inside aggregation inside $I_{\nu(j)}(j)$.

**Dyadic rollout consumes only maximal blocks.** Now consider the dyadic chain $\{b_r(t)\}$ for the fixed $t$. By construction of the dyadic decomposition order, the step from $b_{r-1}(t)$ to $b_r(t) = j$ consumes exactly one dyadic block, namely the maximal dyadic span ending at $j$:

$$[b_{r-1}(t) + 1,\ b_r(t)] = I_{\nu(j)}(j). \tag{11}$$

Crucially, the rollout never consumes any strict subspan $I_k(j)$ with $k < \nu(j)$ as a separate step; such subspans are *always* contained within a maximal block that is consumed in one shot.

**Conclusion: no backtracking.** Therefore, to advance the probabilistic Earley state across the boundary $j = b_r(t)$, it suffices to apply the completion/consumption update using the summary $u(j)$ for the single block (11). All lower-level completions ending at the same $j$ correspond to internal subspans and have already been marginalized into $u(j)$; they do not create competing "choices of what to consume" at this step. Hence the update at $j$ is deterministic given $u(j)$, and no backtracking is required. $\square$

Therefore, we have proved Lemma D.2, which is equivalent to Lemma D.3.

**Example 1 (Boundary $j = 8$).** Take $j = 8$. Then $\nu(8) = 3$, and the dyadic spans ending at 8 are

$$I_0(8) = [8, 8], \quad I_1(8) = [7, 8], \quad I_2(8) = [5, 8], \quad I_3(8) = [1, 8].$$

They satisfy $[8, 8] \subset [7, 8] \subset [5, 8] \subset [1, 8]$. The maximal span is $I_3(8) = [1, 8]$, so $u(8)$ summarizes all derivations whose yield is the entire block $x_{1:8}$. Any completion on $[7, 8]$ or $[5, 8]$ is strictly internal to $[1, 8]$ and is already accounted for inside $u(8)$. Thus when the dyadic rollout crosses the boundary at 8, it consumes the whole block $[1, 8]$ using $u(8)$ and never needs to separately consume $[7, 8]$ or $[5, 8]$.

**Example 2 (Boundary $j = 12$).** Take $j = 12$. Then $\nu(12) = 2$, and the dyadic spans ending at 12 are

$$I_0(12) = [12, 12], \quad I_1(12) = [11, 12], \quad I_2(12) = [9, 12].$$

The maximal span is $I_2(12) = [9, 12]$, so $u(12)$ aggregates all derivations yielding $x_{9:12}$. A completion ending at 12 on the shorter span $[11, 12]$ is internal to $[9, 12]$ and is included in the computation of $u(12)$; therefore the rollout step that advances across boundary 12 only needs $u(12)$.

**Example 3 (Prefix with length $12$)** Let $t = 12$. Its dyadic decomposition is $12 = 8 + 4$, hence the dyadic boundary chain is

$$b_0(t) = 0, \qquad b_1(t) = 8, \qquad b_2(t) = 12.$$

Accordingly, the rollout advances the probabilistic Earley state in two block consumption steps:

$$0 \longrightarrow 8 \longrightarrow 12.$$

At boundary $j = 8$, we have $\nu(8) = 3$, so the maximal dyadic span ending at 8 is $[1, 8]$ and the block update consumes *only* the maximal completion summary $u(8)$ (which aggregates all internal derivations within $[1, 8]$). After applying the usual predict/closure at 8, we proceed to the next boundary.

At boundary $j = 12$, we have $\nu(12) = 2$, so the maximal dyadic span ending at 12 is $[9, 12]$ and the second block update consumes *only* $u(12)$ (the maximal completion summary for $[9, 12]$). Applying predict/closure at 12 yields the Earley state at position $t = 12$.

No additional branching over smaller completions ending at 12 (e.g. on $[11, 12]$ or $[12, 12]$) is necessary: these correspond to strict subspans inside $[9, 12]$ and are already marginalized in $u(12)$.

**Remark** The core idea of these results are not limited to balanced full binary grammar. For general layerd grammar, the types of decomposition of prefix are actually deterministic when grammar is fixed. When knowing these decomposition, one can also rollout Earley algorithm without COMPLETE operations.

### D.8.3. PROBABILISTIC EARLEY STATE, THE SCAN OPERATOR, AND THE PREDICT OPERATOR

In this part, we demonstrate how to encode probabilistic Earley state, and introduce the SCAN and PREDICT operator for the encoded Earley state. The SCAN operator works with the complete constituents to push forward the $\bullet$. The PREDICT operator works with the SCAN results to predict the next / lower Earley State. By our previous statement, it is sufficient that these operation works on $u_r(t)$, the complete summary of the boundaries.

Let $\mathcal{I}$ be a finite set of dotted Earley items sufficient for prediction, and fix a bijection

$$\eta : \mathcal{I} \to \{1, \ldots, d_E\}.$$

A probabilistic Earley state is a vector $E \in \mathbb{R}^{d_E}$ where $E[\eta(i)]$ denotes the probability mass assigned to item $i$.

**Start state $E_0$.** The grammar-defined start state encodes the start-symbol predictor with PCFG probabilities: for each start rule $S \to AB$ with probability $p(S \to AB)$,

$$E_0[\eta(S \Rightarrow \bullet AB)] := p(S \to AB),$$

and all other coordinates are zero.

**Scan relation and matrices.** For each nonterminal $B \in \mathcal{N}$, define a SCAN relation $\mathcal{T}_B \subset \mathcal{I} \times \mathcal{I}$ by

$$(i, i') \in \mathcal{T}_B \quad \Longleftrightarrow \quad i = (A \Rightarrow \alpha \bullet B\,\beta), \; i' = (A \Rightarrow \alpha B \bullet \beta).$$

Define the sparse matrix $M_B \in \mathbb{R}^{d_E \times d_E}$ by

$$(M_B)_{p,q} = \begin{cases} 1, & \text{if } (\eta^{-1}(q), \eta^{-1}(p)) \in \mathcal{T}_B, \\ 0, & \text{otherwise}, \end{cases}$$

where $\eta^{-1}(q) \in \mathcal{I}$ denotes the unique item whose index is $q$.

**Scan operator $F$.** Given an Earley state $E$ and a completion summary $u \in \mathbb{R}^{|\mathcal{N}|}$, define

$$F(E, u) := \sum_{B \in \mathcal{N}} u[\phi(B)]\, M_B\, E \in \mathbb{R}^{d_E}. \tag{12}$$

Since $\mathcal{N}$ and $\mathcal{I}$ are finite, $F$ is a well-defined finite-dimensional map. It is bilinear in $(E, u)$. When a completion summary is represented as a vector in $\mathbb{R}^n$, we identify it with its restriction to nonterminal coordinates (terminal coordinates are 0).

**Scan operator $F$** Let $\mathcal{I}$ be the chosen finite set of dotted Earley items and $\eta : \mathcal{I} \to \{1, \ldots, d_E\}$ a bijection. For any item $i = (A \Rightarrow \alpha \bullet C\,\beta) \in \mathcal{I}$ with $C \in \mathcal{N}$, and any rule $(C \to DE) \in \mathcal{R}$ with probability $p(C \to DE)$, let $i' = (C \Rightarrow \bullet DE) \in \mathcal{I}$ be the predicted item. Define the predictor matrix $P \in \mathbb{R}^{d_E \times d_E}$ by

$$P_{\eta(i'),\eta(i)} \mathrel{+}= p(C \to DE) \qquad \text{whenever} \qquad i = (A \Rightarrow \alpha \bullet C\beta), \; i' = (C \Rightarrow \bullet DE).$$

Since the grammar is layered and acyclic with depth $K$, repeated prediction terminates after at most $K$ steps. We therefore define the (finite) closure matrix

$$C := \sum_{j=0}^{K} P^j \in \mathbb{R}^{d_E \times d_E}, \qquad \text{so that} \qquad \text{Close}(E) = C\,E.$$

### D.8.4. FINAL-LAYER ATTENTION CONSTRUCTION

The final-layer attention is constructed to aggregate the information at the dyadic chain, and provide the complete constituents information for the MLP to SCAN and PREDICT.

The final prediction layer must provide, for each query position $t$, (i) the fixed start Earley state $E_0 \in \mathbb{R}^{d_E}$ and (ii) an *ordered* list of packed completion vectors

$$z_1(t), z_2(t), \ldots, z_K(t) \in \mathbb{R}^n,$$

where $z_r(t) = x_{b_r(t)}^{(K)}$ for $r \le m(t)$ and $z_r(t) = \mathbf{0}$ for $r > m(t)$. A subsequent SliceMLP converts each $z_r(t)$ into its maximal summary $u_r(t) = S_{\beta_r(t)} z_r(t)$. It is essential that these vectors remain *separated by index $r$* (rather than summed), because the rollout consumes them sequentially.

We realize this using $(K + 2)$ deterministic attention heads: one position head (to preserve $e_t^{\text{pos}}$), one BOS head that injects $E_0$, and $K$ dyadic heads, one per rollout step. Crucially, the representation dimension change happens only in the value maps: the input to the final attention is still in $\mathbb{R}^{3n+\ell}$, but each head outputs into the enlarged space

$$\mathbb{R}^{D_{\text{final}}}, \qquad D_{\text{final}} := d_E + Kn + \ell.$$

We interpret this space as three blocks:

$$\underbrace{\mathbb{R}^{d_E}}_{\text{Earley state } E} \quad \oplus \quad \underbrace{\mathbb{R}^{Kn}}_{K \text{ completion blocks } (u_1, \ldots, u_K)} \quad \oplus \quad \underbrace{\mathbb{R}^{\ell}}_{\text{pos}}.$$

**Input convention.** The input to the final layer is the fixed-dimension vector

$$h_u^{(K)} = \begin{bmatrix} x_u^{(K)} \\ 0 \\ 0 \\ e_u^{\mathrm{pos}} \end{bmatrix} \in \mathbb{R}^{3n+\ell},$$

where $x_u^{(K)} \in \mathbb{R}^n$ is the symbol/inside block produced by the first $K$ layers. (For $u = 0$, $x_0^{(K)} = \mathbf{0}$.)

(A) POSITION HEAD: KEEP THE POSITIONAL VECTOR

To preserve positional information across the dimension change, we add a self-attention head that copies $e_t^{\mathrm{pos}}$ into the final positional block.

$$s_{t,u}^{\mathrm{pos}} = \begin{cases} 1, & u = t, \\ 0, & \text{otherwise.} \end{cases}$$

This is implemented by the following QK matrix:

$$W_{QK,\mathrm{pos}} = \begin{bmatrix} 0 & 0 \\ 0 & U_{\mathrm{pos}} \end{bmatrix}$$

where

$$U_{\mathrm{pos}}[t, u] = \begin{cases} 0, & u = t, \\ -M, & \text{otherwise.} \end{cases}$$

for $M \gg 1$. And we can construct the value matrix

$$W_{V,\mathrm{pos}} = \begin{bmatrix} \mathbf{0} & \mathbf{0} & \mathbf{0} & \mathbf{0} \\ \mathbf{0} & \mathbf{0} & \mathbf{0} & \mathbf{0} \\ \mathbf{0} & \mathbf{0} & \mathbf{0} & I_\ell \end{bmatrix} \in \mathbb{R}^{(d_E + Kn + \ell) \times (3n + \ell)}$$

so that

$$W_{V,\mathrm{pos}}[x; 0; 0; e_t^{\mathrm{pos}}] = \begin{bmatrix} 0 \\ 0 \\ e_t^{\mathrm{pos}} \end{bmatrix}$$

Therefore, the attention output is

$$o_t^{pos} = \begin{bmatrix} 0 \\ 0 \\ e_t^{\mathrm{pos}} \end{bmatrix}$$

(B) BOS HEAD: INJECT $E_0$ INTO THE EARLEY BLOCK

We define one attention head bos that deterministically attends to key $u = 0$ for every query $t$:

$$s_{t,u}^{\mathrm{bos}} = \begin{cases} 1, & u = 0, \\ 0, & \text{otherwise.} \end{cases}$$

This is implemented by a purely positional QK mask:

$$W_{QK,\mathrm{bos}} = \begin{bmatrix} 0_{n \times n} & 0 & 0 & 0 \\ 0 & 0_{n \times n} & 0 & 0 \\ 0 & 0 & 0_{n \times n} & 0 \\ 0 & 0 & 0 & U_{\mathrm{bos}} \end{bmatrix}, \qquad U_{\mathrm{bos}} \in \mathbb{R}^{\ell \times \ell},$$

where $U_{\text{bos}}$ is chosen so that the unique finite-score key for any query is $u = 0$ (all other keys get $-M \ll -1$). Because positions are finite, such a deterministic routing mask exists.

$$U_{\text{bos}}[t, u] = \begin{cases} 0, & u = 0, \\ -M, & \text{otherwise.} \end{cases}$$

**BOS head $W_V$: the only source of $E_0$.** The BOS head value map converts $\mathbb{R}^{3n+\ell} \to \mathbb{R}^{d_E+Kn+\ell}$ and writes only into the Earley block:

$$W_{V,\text{bos}} = \begin{bmatrix} 0 & 0 & 0 & A_{\text{bos}} \\ 0 & 0 & 0 & 0 \\ 0 & 0 & 0 & 0 \end{bmatrix},$$

where:

$$A_{\text{bos}} \in \mathbb{R}^{d_E \times \ell}, \qquad W_{V,\text{bos}} : \mathbb{R}^{3n+\ell} \to \mathbb{R}^{d_E+Kn+\ell}.$$

We choose $A_{\text{bos}}$ so that

$$A_{\text{bos}} e_0^{\text{pos}} = E_0, \qquad A_{\text{bos}} e_u^{\text{pos}} = \mathbf{0} \quad (u \neq 0),$$

which is possible because $e_0^{\text{pos}}$ is a fixed one-hot vector for the [BOS] position and the set of positions is finite. Hence the BOS head output is

$$o_t^{\text{bos}} = \sum_u s_{t,u}^{\text{bos}} W_{V,\text{bos}} h_u^{(K)} = \begin{bmatrix} E_0 \\ \mathbf{0}_{Kn} \\ \mathbf{0}_\ell \end{bmatrix}.$$

(C) $K$ DYADIC HEADS: ROUTE PACKED COMPLETION VECTORS INTO SLOTS

Each dyadic head produces a *packed* vector $z_r(t)$ in slot $r$. If $r \leq m(t)$ then $z_r(t) = x_{b_r(t)}^{(K)}$; otherwise $z_r(t) = \mathbf{0}$. The SliceMLP in part (E) will convert $z_r(t)$ into the maximal summary $u_r(t) = S_{\beta_r(t)} z_r(t)$.

**Dyadic head routing.** For each $r \in \{1, \ldots, K\}$ we use one attention head that deterministically routes query position $t$ to key position $u = b_r(t)$ if $r \leq m(t)$, and outputs $\mathbf{0}$ otherwise. Concretely, its weights satisfy

$$s_{t,u}^{(r)} = \begin{cases} 1, & u = b_r(t) \text{ and } r \leq m(t) \text{ and } t \neq 0 \\ 1, & (r > m(t) \text{ or } t = 0) \text{ and } u = 0 \\ 0, & \text{otherwise.} \end{cases}$$

As in earlier sections, we implement $s^{(r)}$ by a purely positional QK mask:

$$W_{QK}^{(r)} = \begin{bmatrix} 0 & 0 & 0 & 0 \\ 0 & 0 & 0 & 0 \\ 0 & 0 & 0 & 0 \\ 0 & 0 & 0 & U^{(r)} \end{bmatrix}, \qquad U^{(r)} \in \mathbb{R}^{\ell \times \ell},$$

where $U^{(r)}$ is chosen so that the unique finite-score key for query $t$ is $u = b_r(t)$ when $r \leq m(t)$, and the fallback key is $u = 0$ when $r > m(t)$ (or $t = 0$). (The existence follows from finiteness of positions and one-hot positional codes.)

$$U^{(r)}[t, u] = \begin{cases} 0, & u = b_r(t) \text{ and } r \leq m(t) \text{ and } t \neq 0 \\ 0, & (r > m(t) \text{ or } t = 0) \text{ and } u = 0 \\ -M, & \text{otherwise.} \end{cases}$$

**Dyadic head value maps (write full packed vector into slot $r$).** Each dyadic head uses a value map

$$W_V^{(r)} : \mathbb{R}^{3n+\ell} \to \mathbb{R}^{d_E+Kn+\ell},$$

that writes *only* the main block $x_u^{(K)} \in \mathbb{R}^n$ into the $r$-th packed slot $z_r$ and writes zeros elsewhere. In block form (rows grouped as $(E; z_1; \ldots; z_K; \mathrm{pos})$ and columns as $(x; u_L; u_R; \mathrm{pos})$),

$$W_V^{(r)} = \begin{bmatrix} 0 & 0 & 0 & 0 \\ \hline 0 & 0 & 0 & 0 \\ \vdots & \vdots & \vdots & \vdots \\ I_n & 0 & 0 & 0 \\ \vdots & \vdots & \vdots & \vdots \\ 0 & 0 & 0 & 0 \\ \hline 0 & 0 & 0 & 0 \end{bmatrix},$$

where $I_n$ is placed in the row block corresponding to slot $z_r$. Thus

$$W_V^{(r)} h_u^{(K)} = \begin{bmatrix} \mathbf{0}_{d_E} \\ \mathbf{0} \\ \vdots \\ x_u^{(K)} \\ \vdots \\ \mathbf{0} \\ \mathbf{0}_\ell \end{bmatrix}.$$

and

$$o_t^r = \sum_u s_{t,u}^r W_V^{(r)} h_u^{(K)} = \begin{bmatrix} \mathbf{0}_{d_E} \\ z_1(t) \\ \vdots \\ z_K(t) \\ \mathbf{0}_\ell \end{bmatrix}, \qquad z_r(t) = \begin{cases} x_{b_r(t)}^{(K)}, & r \le m(t), \\ \mathbf{0}, & r > m(t). \end{cases}$$

**Remark.** When a dyadic head falls back to $u = 0$, we have $x_0^{(K)} = \mathbf{0}$. Since $W_V^{(r)}$ ignores the positional block, it follows that $W_V^{(r)} h_0^{(K)} = \mathbf{0}$, so this fallback contributes nothing.

(D) FINAL ATTENTION OUTPUT (POS, BOS, AND DYADIC HEADS).

Summing the position head, the BOS head (injecting $E_0$ into the Earley block), and the $K$ dyadic heads yields

$$\tilde{h}_t^{(K+1)} := o_t^{pos} + o_t^{bos} + \sum_{r=1}^K o_t^r = \begin{bmatrix} E_0 \\ z_1(t) \\ \vdots \\ z_K(t) \\ e_t^{\mathrm{pos}} \end{bmatrix} \in \mathbb{R}^{d_E+Kn+\ell}.$$

At this stage each $z_r(t)$ still contains all blocks $[c_{b_r(t),0}; \ldots; c_{b_r(t),K}]$. The next Slice-MLP will deterministically keep only the maximal block $c_{b_r(t),\beta_r(t)}$.

(E) MLP TO SLICE OUTPUT.

**Setup and objective.** Recall that each dyadic head outputs, for slot $r$,

$$z_r(t) = \begin{cases} x_{b_r(t)}^{(K)} = [c_{b_r(t),0}; \ldots; c_{b_r(t),K}]^\top \in \mathbb{R}^n, & r \le m(t), \\ \mathbf{0}, & r > m(t), \end{cases}$$

where $n = \sum_{k=0}^{K} n_k$. The goal of slicing is to produce

$$u_r(t) \in \mathbb{R}^n \quad \text{by keeping only the maximal block} \quad k = \beta_r(t) := \nu\big(b_r(t)\big),$$

i.e.

$$u_r(t) = S_{\beta_r(t)} z_r(t),$$

with all other blocks set to zero.

Equivalently,

$$u_r(t) = \sum_{k=0}^{K} g_{r,k}(t) \, S_k z_r(t), \qquad g_{r,k}(t) := \mathbf{1}\{r \le m(t) \text{ and } \beta_r(t) = k \text{ and } t \ne 0\}.$$

where $S_k$ is the block wise selection matrix.

**Blockwise selector matrices.** Let

$$n = \sum_{k=0}^{K} n_k, \qquad x = \big[x^{(0)}; \, x^{(1)}; \, \ldots; \, x^{(K)}\big], \quad x^{(k)} \in \mathbb{R}^{n_k}.$$

For each $k \in \{0, \ldots, K\}$, define the blockwise selector $S_k \in \mathbb{R}^{n \times n}$ by the block matrix

$$S_k = \begin{bmatrix} 0_{n_0 \times n_0} & & & & & \\ & \ddots & & & & \\ & & 0_{n_{k-1} \times n_{k-1}} & & & \\ & & & I_{n_k} & & \\ & & & & 0_{n_{k+1} \times n_{k+1}} & \\ & & & & & \ddots \end{bmatrix},$$

where the $k$-th diagonal block is the identity matrix $I_{n_k}$ and all other diagonal and off-diagonal blocks are zero matrices of compatible sizes.

By construction,

$$S_k \, x = \big[0; \, \ldots; \, 0; \, {x^{(k)}}^\top; \, 0; \, \ldots; \, 0\big]^\top \in \mathbb{R}^n,$$

i.e. $S_k$ preserves the $k$-th block of $x$ and zeros out all other blocks.

**Positional gates.** Since $e_t^{\text{pos}}$ is one-hot and the position set is finite, for each $(r, k)$ there exists a fixed vector $w_{r,k} \in \{0, 1\}^{\ell}$ such that

$$g_{r,k}(t) = \langle w_{r,k}, e_t^{\text{pos}} \rangle \in \{0, 1\}.$$

For each fixed $r$ and $t$, exactly one $k$ satisfies $g_{r,k}(t) = 1$ if $r \le m(t)$, and all $g_{r,k}(t) = 0$ if $r > m(t)$.

**Slice-MLP definition.** We implement slicing by a two-layer MLP with square activation,

$$\text{SliceMLP}(z) = W_2^{\text{sl}} \, \sigma\big(W_1^{\text{sl}} z + b_1^{\text{sl}}\big) + b_2^{\text{sl}}, \qquad \sigma(x) = x^2,$$

mapping $\mathbb{R}^{d_E + Kn + \ell} \to \mathbb{R}^{d_E + Kn + \ell}$. No residual connection is used.

**Hidden units: gated blockwise products.** Fix a slot $r$, a level $k \in \{0, \ldots, K\}$, and a coordinate $i \in \{1, \ldots, n\}$. Define the scalar

$$y_{r,k,i}(t) := [S_k z_r(t)]_i, \qquad g_{r,k}(t) \in \{0, 1\}.$$

We introduce two hidden units

$$h_{r,k,i}^+(t) := \big(g_{r,k}(t) + y_{r,k,i}(t)\big)^2, \qquad h_{r,k,i}^-(t) := \big(g_{r,k}(t) - y_{r,k,i}(t)\big)^2,$$

realized by rows of $W_1^{\text{sl}}$ selecting exactly the two input coordinates $(g_{r,k}, y_{r,k,i})$ with coefficients $(+1, +1)$ and $(+1, -1)$, and zero bias.

**Exact gated copy via square identity.** Using the same identity as in earlier product constructions Eq. (5),

$$xy = \frac{(x+y)^2 - (x-y)^2}{4},$$

we obtain

$$g_{r,k}(t)\, y_{r,k,i}(t) = \frac{h^+_{r,k,i}(t) - h^-_{r,k,i}(t)}{4}.$$

**Output wiring: blockwise injection and summation.** The output matrix $W_2^{\mathrm{sl}}$ is defined so that:

- for each $(r, k, i)$, the scalar $\frac{h^+_{r,k,i} - h^-_{r,k,i}}{4}$ is written into the $(r, k, i)$ coordinate of the packed output via the fixed injector $J_k$;

- all other coordinates receive zero contribution from this group.

Summing over $k$ yields, for each slot $r$,

$$u_r(t) = \sum_{k=0}^{K} g_{r,k}(t)\, S_k z_r(t) \in \mathbb{R}^n.$$

**Identity on Earley and positional blocks.** The Earley block $E_0$ and positional block $e_t^{\mathrm{pos}}$ are copied identically using the same square-based linear reconstruction as in previous sections, without interacting with the slicing logic. (This can use the $y = \frac{(y+1)^2 - y^2 - 1}{2}$ technique, which is trivial and we do not elaborate further here.).

**Summary of the current output**

$$\hat{h}_t^{(K+1)} = \begin{bmatrix} E_0 \\ u_1(t) \\ \vdots \\ u_K(t) \\ e_t^{\mathrm{pos}} \end{bmatrix} \in \mathbb{R}^{d_E + Kn + \ell}$$

## D.9. Final Rollout: Predictor Closure and $K$ Overwrite-MLPs

### D.9.1. SCAN AND PREDICT MLPS

The SliceMLP produces, for each position $t$, the vector

$$h_{t,0} = \hat{h}_t^{(K+1)} = \begin{bmatrix} E_{t,0} \\ u_1(t) \\ \vdots \\ u_K(t) \\ e_t^{\mathrm{pos}} \end{bmatrix} \in \mathbb{R}^{d_E + Kn + \ell}, \qquad E_{t,0} = E_0$$

where each $u_r(t) \in \mathbb{R}^n$ is the maximal completion summary for the $r$-th dyadic block.

We perform a fixed $K$-step rollout. At each step $r$, only the $r$-th completion summary $u_r(t)$ may be consumed. The target Earley update is

$$E_{t,r} := \begin{cases} \mathrm{Close}\left( \sum_{B \in \mathcal{N}} u_r[\phi(B)]\, M_B\, E_{t,r-1} \right) = \sum_{B \in \mathcal{N}} u_r[\phi(B)]\, (C M_B)\, E_{t,r-1}, & r \le m(t), \\ E_{t,r-1}, & r > m(t), \end{cases}$$

where $C = \sum_{j=0}^{K} P^j$ is the finite closure matrix.

**Overwrite information.** Crucially, the rollout MLPs do **not** use residual connections. Instead, each step-$r$ MLP directly outputs the next hidden state, overwriting the Earley block when $r \leq m(t)$ and acting as the identity otherwise. All completion slots and the positional block are always passed through unchanged.

Formally, the desired target map is

$$\mathrm{MLP}_r^{(K+1)}(h_{t,r-1}) = \begin{bmatrix} E_{t,r} \\ u_1(t) \\ \vdots \\ u_K(t) \\ e_t^{\mathrm{pos}} \end{bmatrix}, \qquad h_{t,r} := \mathrm{MLP}_r^{(K+1)}(h_{t,r-1}).$$

**Two-layer square-activation construction.** Each $\mathrm{MLP}_r^{(K+1)}$ is implemented as a two-layer network

$$\mathrm{MLP}_r^{(K+1)}(z) = W_{2,r}\, \sigma\big(W_{1,r}z + b_{1,r}\big) + b_{2,r}, \qquad \sigma(x) = x^2,$$

with fixed input/output dimension $d_E + Kn + \ell$. The hidden units are partitioned into the following groups.

**(I) Bilinear completion features.** For each Earley coordinate $q \in \{1, \ldots, d_E\}$ and each nonterminal $B \in \mathcal{N}$, we introduce two hidden units

$$h_{q,B}^+ = (E_q + u_r[\phi(B)])^2, \qquad h_{q,B}^- = (E_q - u_r[\phi(B)])^2.$$

These are realized by rows of $W_1^{(r)}$ selecting exactly the two input coordinates $(E_q,\ u_r[\phi(B)])$ with coefficients $\pm 1$ and zero bias. Their linear combination yields the exact product using the trick in Eq. (5)

$$m_{q,B} := \tfrac{1}{4}\big(h_{q,B}^+ - h_{q,B}^-\big) = E_q \cdot u_r[\phi(B)].$$

**(II) Step-activation gate.** Using the one-hot positional encoding $e_t^{\mathrm{pos}}$, we define a binary gate

$$g_r(t) := \mathbf{1}\{r \leq m(t)\} \in \{0,1\},$$

realized as a linear functional of $e_t^{\mathrm{pos}}$. This gate determines whether step $r$ performs a completion update ($g_r(t) = 1$) or acts as the identity ($g_r(t) = 0$), keeping the Earley block unchanged.

**(III) Identity-preserving features.** To allow exact pass-through of all non-Earley blocks, we include, for each coordinate $y$ of every completion slot and of the positional block, hidden units implementing

$$y = \tfrac{1}{2}\big((y+1)^2 - y^2 - 1\big),$$

so that these coordinates can be copied exactly through the MLP. These features are independent of $r$ and shared across all rollout steps. Concretely, for each coordinate of $u_s(t)$ ($s = 1, \ldots, K$) and $e_t^{\mathrm{pos}}$, we include the two pre-activations $(y+1)$ and $y$ in $W_1^{(r)}$ and combine them in $W_2^{(r)}$ to reproduce $y$ exactly.

**Output layer and conditional overwrite.** The output weights $W_2^{(r)}$ are chosen as follows.

*Earley block.* For each output coordinate $p \in \{1, \ldots, d_E\}$, the MLP computes

$$E_{t,r}[p] = g_r(t) \sum_{B \in \mathcal{N}} \sum_{q=1}^{d_E} (CM_B)_{p,q}\, m_{q,B}\ +\ \big(1 - g_r(t)\big) E_{t,r-1}[p].$$

Thus, when $r \leq m(t)$, the Earley block is overwritten by the completed-and-closed state, and when $r > m(t)$ it is passed through unchanged.

*Completion slots and positional block.* For every completion slot $u_s(t)$ ($s = 1, \ldots, K$) and for the positional block, the output layer selects only the identity-preserving features, so that these components are copied exactly:

$$u_{s,r}(t) = u_{s,r-1}(t), \qquad e_t^{\mathrm{pos}} \text{ unchanged.}$$

**Resulting rollout.** Starting from $h_{t,0}$, the $K$ conditional-overwrite MLPs are applied sequentially:

$$h_{t,r} := \mathrm{MLP}_r^{(K+1)}(h_{t,r-1}), \qquad r = 1, \ldots, K.$$

The final Earley state is the Earley block of $h_{t,K}$ and is used to compute the next-token distribution.

### D.10. Approximating the Final $K + 1$ MLP Composition by a Single MLP

**Lemma D.4** (Approximation of Composed MLPs). *Let $f_{\mathrm{MLP}} : X \to \mathbb{R}^{d_E+Kn+\ell}$ be a function defined by the composition of $K + 1$ finite-layer MLPs: $f_{\mathrm{MLP}} = \mathrm{MLP}_K^{(K+1)} \circ \cdots \circ \mathrm{MLP}_1^{(K+1)} \circ \mathrm{SliceMLP}$, where the domain $X \subset \mathbb{R}^{d_E+Kn+\ell}$ is a compact set. Given any $\epsilon > 0$ and a non-polynomial continuous activation function $\sigma(\cdot)$, there exists a single two-layer MLP, denoted as $\widehat{f}$, such that:*

$$\sup_{h \in X} \|f_{\mathrm{MLP}}(h) - \widehat{f}(h)\| < \epsilon$$

*Proof.* We first establish the compactness of the domain $X$. According to the explicit construction in Sections D.5, D.8.4, and D.9.1, any input vector $h \in X$ consists of three components:

1. Unnormalized Earley state probabilities, which are bounded within the unit hypercube $[0,1]^{d_E}$.

2. Inside probabilities for complete constituents, which are bounded within $[0,1]^{Kn}$.

3. Positional embeddings $\{e_t^{\mathrm{pos}}\}_{t=0}^T$, which constitute a finite set of discrete one-hot vectors in $\mathbb{R}^\ell$.

The domain $X$ is a subset of the Cartesian product $[0,1]^{d_E+Kn} \times \{e_t^{\mathrm{pos}}\}_{t=0}^T$. Since it is a finite union of compact sets (or equivalently, a closed and bounded subset of $\mathbb{R}^{d_E+Kn+\ell}$), $X$ is compact by the Heine-Borel theorem.

Next, we address the continuity of $f_{\mathrm{MLP}}$. Each constituent MLP (including SliceMLP and $\mathrm{MLP}_r^{(K+1)}$) is constructed using linear transformations and the activation function $\phi(z) = z^2$. Since linear maps and polynomial activations are continuous, each MLP defines a continuous mapping. Consequently, their composition $f_{\mathrm{MLP}}$ is continuous on the compact set $X$.

Finally, by the Universal Approximation Theorem (Hornik, 1991, Theorem 2), the set of functions represented by a two-layer MLP with a non-polynomial continuous activation $\sigma(\cdot)$ is dense in $C(X)$ under the supremum norm. Therefore, for any $\epsilon > 0$, there exists a $\widehat{f}$ that approximates $f_{\mathrm{MLP}}$ with error at most $\epsilon$ for all $h \in X$. $\square$

### D.11. Final Readout

Let $E_t \in \mathbb{R}^{d_E}$ denote the final Earley state. Define a readout matrix $R \in \mathbb{R}^{(|\Sigma|+1) \times d_E}$ by

$$R_{a,\eta(i)} = \begin{cases} 1, & a \in \Sigma \text{ and } i = (A \Rightarrow \alpha \bullet a\beta), \\ 1, & a = [\text{EOS}] \text{ and } i = (S \Rightarrow \gamma\bullet), \\ 0, & \text{otherwise.} \end{cases}$$

The unnormalized next-token scores are $\tilde{p}_t = RE_t$, and the next-token distribution is obtained by normalizing $\tilde{p}_t$.

### D.12. Summary of the Model Construction

For the first $K$ layers, for each $k = 1, \ldots, K$,

$$\tilde{h}_t^{(k)} = h_t^{(k-1)} + \sum_{H \in \{L,R\}} \mathrm{Attn}_H^{(k)}(h^{(k-1)})_t,$$

$$h_t^{(k)} = \tilde{h}_t^{(k)} + \mathrm{MLP}^{(k)}(\tilde{h}_t^{(k)}).$$

For the final layer,

$$\tilde{h}_t^{(K+1)} = \text{Attn}_{\text{pos}}^{(K+1)}(h^{(K)})_t + \text{Attn}_{\text{bos}}^{(K+1)}(h^{(K)})_t + \sum_{r=1}^K \text{Attn}_r^{(K+1)}(h^{(K)})_t,$$

$$h_{t,0} = \text{SliceMLP}(\tilde{h}_t^{(K+1)}),$$

$$h_{t,r} = \text{MLP}_r^{(K+1)}(h_{t,r-1}), \qquad r = 1, \ldots, K,$$

where $\text{SliceMLP} = \text{MLP}_0^{(K+1)}$ and $h_{t,0} = \hat{h}_t^{(K+1)}$.

Finally, the unnormalized next-token scores are

$$\tilde{p}_t = RE_t = R\,[h_{t,K}]_{1:d_E}.$$

## D.13. Main Theorems: Exact Next-Token Prediction

Now we can conclude the above results and formally state our main theorem for the construction. The following Theorem D.5 is the formal version of Theorem 4.2. The Theroem D.6 is a stronger version that uses only one MLP in the final layer.

### D.13.1. MAIN THEOREM: END-TO-END CONSTRUCTIVE TRANSFORMER FOR EXACT NEXT-TOKEN PREDICTION

**Theorem D.5** (Constructive Transformer achieves exact next-token prediction). *Fix a maximum length $T$ and let $\ell := T + 1$ with one-hot positional encodings $e_t^{\text{pos}} \in \{0,1\}^\ell$ for $t \in \{0, 1, \ldots, T\}$. Let $\mathcal{G} = (\mathcal{N}, \Sigma, \mathcal{R}, S, p)$ satisfy Definition 4.1 with depth $K$, and set $n := |\mathcal{N} \cup \Sigma|$. Then there exists an explicit finite-depth Transformer-like network $\mathcal{T}$ such that for every prefix length $t \leq T$,*

$$\mathcal{T}(x_{1:t}) = \big(p_G(a \mid x_{1:t})\big)_{a \in \Sigma \cup \{[EOS]\}}.$$

*Moreover, $\mathcal{T}$ can be chosen with the following concrete architecture:*

**(A) Inside stack (depth $K$; fixed dimension).** *There are $K$ standard Transformer blocks indexed by $k = 1, \ldots, K$, all operating on sequence length $T + 1$ and model dimension*

$$d_{\text{in}} := 3n + \ell.$$

*At each layer $k$ and position $t$, the hidden state is partitioned as*

$$h_t^{(k)} = \big[x_t^{(k)};\, u_{L,t}^{(k)};\, u_{R,t}^{(k)};\, e_t^{\text{pos}}\big] \in \mathbb{R}^{3n+\ell}, \qquad x_t^{(k)}, u_{L,t}^{(k)}, u_{R,t}^{(k)} \in \mathbb{R}^n.$$

*Each inside layer uses:*

- **Attention:** *exactly 2 heads $(L, R)$ with pure positional QK logits in $\{0, -M\}$ (for a fixed large $M$), hence deterministic routing in the limit $M \to \infty$. The value maps are fixed linear maps $W_{V,L}, W_{V,R} \in \mathbb{R}^{(3n+\ell)\times(3n+\ell)}$ that write only into the buffers: head $L$ copies the attended $x$-block into $u_L$, head $R$ copies the attended $x$-block into $u_R$.*

- **Residuals:** *the inside layer uses two residual connections: one after attention and one after the MLP:*

$$\tilde{h}_t^{(k)} = h_t^{(k-1)} + \text{Attn}^{(k)}(h^{(k-1)})_t, \qquad h_t^{(k)} = \tilde{h}_t^{(k)} + \text{MLP}^{(k)}(\tilde{h}_t^{(k)}).$$

- **MLP:** *a two-layer position-wise MLP with square activation $\sigma(z) = z^2$,*

$$\text{MLP}^{(k)}(z) = W_2^{(k)}\, \sigma(W_1^{(k)} z + b_1^{(k)}) + b_2^{(k)},$$

*mapping $\mathbb{R}^{3n+\ell} \to \mathbb{R}^{3n+\ell}$. It (i) computes the level-$k$ inside update on boundaries using $(x \pm y)^2$ product features, (ii) outputs exact negatives to reset both buffers, and (iii) outputs $0$ on the positional block.*

**(B) Final prediction module (dimension change).** *After $K$ inside layers, a single* final attention stage *maps from $d_{\text{in}}$ to the enlarged dimension*

$$d_{\text{final}} := d_E + Kn + \ell,$$

*where $d_E := |\mathcal{I}|$ is the dimension of a fixed finite Earley-item set $\mathcal{I}$.*

*This final attention stage uses exactly $(K + 2)$ deterministic heads:*

- **pos-head (1 head):** *self-attends and copies $e_t^{\text{pos}}$ to the final positional block.*

- **BOS-head (1 head):** *attends to key position $0$ for all queries and injects the fixed start Earley vector $E_0 \in \mathbb{R}^{d_E}$ into the Earley block via a value map $A_{\text{bos}} \in \mathbb{R}^{d_E \times \ell}$ satisfying $A_{\text{bos}} e_0^{\text{pos}} = E_0$.*

- $K$ **dyadic heads:** *for each $r \in \{1, \ldots, K\}$, head $r$ routes query $t$ to key $b_r(t)$ (if $r \leq m(t)$, otherwise to key $0$) using a pure positional QK mask, and writes the key's $x^{(K)}$ vector into the $r$-th slot $z_r \in \mathbb{R}^n$ via a value map that places $I_n$ in that slot and $0$ elsewhere.*

*Importantly, this is the* only *place where the representation dimension changes (from $3n + \ell$ to $d_E + Kn + \ell$), and the change is realized solely by the value maps.*

**(C) SliceMLP (no residual; same $d_{\text{final}}$).** *A two-layer square-activation MLP*

$$\text{SliceMLP} : \mathbb{R}^{d_{\text{final}}} \to \mathbb{R}^{d_{\text{final}}}$$

*(with* no residual*) replaces each $z_r(t)$ by the maximal-level completion summary $u_r(t)$ by blockwise selection, while copying the Earley and positional blocks exactly.*

**(D) Scan and Predict MLP (depth $K$; no residual).** *There are $K$ position-wise overwrite MLPs $\{\text{MLP}_r^{(K+1)}\}_{r=1}^K$, each a two-layer square-activation map*

$$\text{MLP}_r^{(K+1)} : \mathbb{R}^{d_{\text{final}}} \to \mathbb{R}^{d_{\text{final}}}$$

*(with* no residual*) that updates only the Earley block by*

$$E \leftarrow \text{Close}\big(F(E, u_r(t))\big) \quad \text{if } r \leq m(t), \qquad \text{and} \qquad E \leftarrow E \text{ otherwise,}$$

*while passing through all slots $(u_1, \ldots, u_K)$ and $e_t^{\text{pos}}$ unchanged.*

**(E) Readout.** *Finally, a linear map $R \in \mathbb{R}^{(|\Sigma|+1) \times d_E}$ (followed by normalization) maps the final Earley block $E_t$ to the next-token distribution.*

*All parameters are explicit functions of $(\mathcal{G}, T, K)$.*

*Proof.* By Theorem D.1, with the structure in **(A)**, after the first $K$ Transformer layers, the hidden states contain the insides probabilities at every boundary. From Section D.8.4, D.8.4, and D.9.1, we know the structure in **(B)**, **(C)** and **(D)** can perform Earley SCAN and PREDICT using the information at the maximal-level completion summary. By Lemma D.3, we know this is sufficient to get the Earley state for the Balanced Full-binary grammar. The readout operation in **(E)** perform exactly the computation of next-token probabilities conditioned on Earley State.

Therefore, this construction can achieves exact next-token prediction. $\square$

The following theorem is a stronger version that uses a more standard Transformer structure with only one MLP in the final layer.

**Theorem D.6** (Constructive Transformer with only 1 MLP in the last layer)**.** *Fix a maximum length $T$ and let $\ell := T + 1$ with one-hot positional encodings $e_t^{\text{pos}} \in \{0, 1\}^\ell$ for $t \in \{0, 1, \ldots, T\}$. Let $\mathcal{G} = (\mathcal{N}, \Sigma, \mathcal{R}, S, p)$ satisfy Definition 4.1 with depth $K$, and set $n := |\mathcal{N} \cup \Sigma|$. Then there exists an explicit finite-depth Transformer-like network $\mathcal{T}$ such that for every prefix length $t \leq T$,*

$$\mathcal{T}(x_{1:t}) \approx \big(p_G(a \mid x_{1:t})\big)_{a \in \Sigma \cup \{[EOS]\}}.$$

*Moreover, $\mathcal{T}$ can be chosen with the same architecture **(A)**, **(B)**, **(E)** as those in Theorem D.5, but approximate the $K + 1$ MLP in **(C)**, **(D)** with a 2-layer MLP $\hat{f}$ with a non-polynomial continuous activation function $\sigma(\cdot)$.*

*Proof.* From Lemma D.4, we know that there exists a 2-layer MLP $\hat{f}$ with a non-polynomial continuous activation function $\sigma(\cdot)$ that can approximate the $K + 1$ MLP in **(C), (D)** in Theorem D.5 with error $\epsilon$. □

