# OpenReview forum: "How Transformers Represent Hierarchies: A Local-to-Global Mechanism"
_ICML.cc/2026/Conference — ICML 2026 regular_

### Official Review · Reviewer_LXUf · 2026-02-25

**Soundness:** 2
**Presentation:** 3
**Significance:** 2
**Originality:** 2
**Overall Recommendation:** 4
**Confidence:** 2

**Summary:**

This paper explores how autoregressive Transformer models represent and update hierarchical information when processing sequences with latent hierarchical structures, such as natural language. The authors establish a controlled experimental environment using Probabilistic Context-Free Grammars (PCFGs) to investigate the internal computational mechanisms of Transformers. They construct a Transformer instance for balanced full-binary grammars, demonstrating its capability to achieve hierarchical parsing with a depth of $O(\log  L)$, where $L$ represents the sequence length. Furthermore, the attention patterns exhibited by the trained model closely align with the characteristics of this theoretical construction.

**Compliance With Llm Reviewing Policy:**

Affirmed.

**Final Justification:**

The author's rebuttal addressed my concerns, and I will keep my original rating unchanged. However, I am not an expert in this field. Therefore, after considering the comments raised by the other reviewers, I have lowered my confidence score to 2.

**Key Questions For Authors:**

1. I believe this paper constitutes a piece of work on the interpretability of deep learning, and as I understand, research in this area is quite extensive. I would suggest that the authors explicitly add a dedicated related work section to systematically and concisely outline the perspectives and contributions of different researchers in this field, as well as the development of the work most relevant to theirs. This would be beneficial for helping newcomers to the field quickly grasp the significance of the research.

2. The paper demonstrates that $O(\log L)$ depth is sufficient for parsing. However, during practical training, models often require a parameter count far exceeding the theoretical prediction to converge to this theoretically predicted ideal state. May I ask why this is the case?

**Limitations:**

See Weakness.

**Strengths And Weaknesses:**

$\textbf{Strength}$

1. The paper is professionally written, with sufficient experiments that effectively address the research questions it sets out to investigate.

2. The authors propose and validate a clear mechanistic interpretation: lower layers perform local modeling by aggregating short-span constituents (following CYK-like logic), while deeper layers integrate these local features into global Earley states. This finding provides a granular perspective on how autoregressive models handle long-range dependencies.

3. The paper explains how Transformers leverage parallel computation to accomplish syntactic derivations within a limited number of layers that would logically require more steps—a quite interesting insight.

$\textbf{Weakness}$

1. While the study is primarily based on balanced full-binary grammars—which are ideal for theoretical validation—natural language in real-world scenarios often involves a substantial number of unbalanced structures, center embeddings, or highly ambiguous syntax. It remains unclear whether the proposed mechanism remains robust when handling asymmetric or deeply nested hierarchical structures.

---

> ### Author Rebuttal · Authors · 2026-03-31
>
> Thank you for your review.
>
> **Q1: Unclear if mechanism is robust for asymmetric or deeply nested structures.**
>
> **A1:** While our theoretical construction focuses on **balanced full-binary grammars**, the empirical grammars (CFG1--5) feature **variable-length and non-binary rules** with diverse branching factors, repetition patterns, and ambiguity levels (see Appendix C). Furthermore, the new **depth-6 grammar** [Supplementary](https://anonymous.4open.science/r/cfg-transformer-56DD/rebuttal_results.pdf) Figure 1 includes **ternary productions**, going beyond the binary setting of the theory. The consistent local-to-global pattern across all these grammars suggests the mechanism is **not restricted to perfectly balanced derivations**. That said, our goal is not to directly model all of natural language, whose structure is too rich to permit rigorous mechanistic analysis with reliable ground truth. We study a controlled setting that is rich enough to capture **nontrivial hierarchical structure and long-range dependence**, while remaining tractable for precise interpretation.
>
> ---
>
> **Q2: Add a dedicated related work section.**
>
> **A2:** In the Introduction, we discuss the related work most directly connected to our paper. Due to space limitations, Appendix A contains a broader discussion of interpretability-related work. We will **add more related-work discussion to the main text** in the revision to help readers better appreciate the context and significance.
>
> ---
>
> **Q3: $O(\log L)$ depth suffices theoretically, but practical models need far more parameters -- why?**
>
> **A3:** The $O(\log L)$-layer result comes from our theoretical construction for the balanced full-binary grammars and should be interpreted as an upper bound on **depth**. Empirically, for grammars with maximum sequence length up to 81, a **5-layer Transformer** already suffices, consistent with the $O(\log L)$ bound. Regarding **parameter count** (width): our theoretical construction requires only a small number of heads and moderate hidden dimension, while the practical model uses $d=512$ and 4 heads -- substantially more parameters. This gap is a **well-known phenomenon** in neural network theory: **gradient-based optimization typically requires overparameterization** to navigate the loss landscape and converge, even when the target function admits a compact representation. Understanding this gap connects to the broader open question of learning dynamics, which we leave as an important question for future work.

---

> > ### Author Rebuttal · Reviewer_LXUf · 2026-04-01
> >
> > Thank you for the authors' response.
> >
> > I will maintain my original score.

---

> > > ### Author Response · Authors · 2026-04-06
> > >
> > > Thank you for your positive evaluation of our work.
> > >
> > > We would like to highlight an additional experiment ([Supplementary](https://anonymous.4open.science/r/cfg-transformer-56DD/rebuttal_results_2.pdf)) that further addresses your concern about robustness against unbalanced structures and structurally diverse grammars. We have run experiments on **50 randomly generated depth-4 PCFGs**, where for each nonterminal the number of production rules is drawn uniformly from $\\{2, 3, 4\\}$, the RHS length from $\\{2, 3\\}$, and each RHS symbol uniformly from the nonterminals of the next layer. A **4-head, 5-layer** Transformer is trained on each grammar under the same setup as the main paper.
> > >
> > > This design directly targets your concern: because both the number of rules and the RHS length are random, the resulting grammars span a wide range of **branching factors, ambiguity levels, and derivation widths** — including non-binary and unbalanced productions that go beyond both our theoretical construction and the hand-designed CFG1--5. No grammar is curated or selected for clean results.
> > >
> > > Across all 50 random grammars, we consistently observe:
> > >
> > > - **CYK-father and Earley-state probing accuracy increase monotonically layer by layer**, showing that hierarchical constituent information is progressively constructed across depth.
> > > - **Larger-window CYK-father probes succeed only in deeper layers**, confirming that longer-range structural dependencies are resolved later in the network.
> > > - **The CYK/Earley correspondence holds throughout**: early layers recover bottom-up completed constituents (CYK-style), while the final layer encodes prefix-consistent Earley states.
> > >
> > > These findings **mirror the main paper's results precisely**. The fact that the same local-to-global pattern emerges consistently across 50 randomly structured grammars — without any selection or tuning — provides strong evidence that the mechanism is a general property of autoregressive Transformers processing hierarchical structure, not an artifact of any particular grammar design. We look forward to incorporating these results in the revision.

---

### Official Review · Reviewer_bGv3 · 2026-03-07

**Soundness:** 2
**Presentation:** 2
**Significance:** 3
**Originality:** 3
**Overall Recommendation:** 3
**Confidence:** 3

**Summary:**

The authors design probing tasks to find out how transformer decoders process hierarchical structure. Their results reveal that the transformer hidden states encode information similar to that computed by classic parsing algorithms such as CKY and Earley's algorithm. More specifically, the authors first probe whether the hidden state at each position encodes the corresponding Earley chart state, and find that the accuracy increases monotonically across the first 4 layers, then drops at the final layer, suggesting that the first layers encode information similar to that of an Earley chart, while the final layer aggregates information for next-token prediction. Next, the authors probe whether transformers learn completed constituents like CKY, and find that transformers indeed learn completed constituents like in the CKY algorithm. Finally, the authors use span-restricted probing to test whether transformers transformers build this information hierarchically, building larger constituents from smaller ones, and find that this is the case. To complement their empirical findings, the authors give a theoretical construction showing how transformers implement local-to-global parsing.

**Compliance With Llm Reviewing Policy:**

Affirmed.

**Final Justification:**

The authors addressed some of my concerns in the rebuttal, therefore I increased my score slightly. Some of my concerns still remain, though.

**Key Questions For Authors:**

1. Could you elaborate on the choice of bounded PCFGs? I feel like this choice deserves some more discussion in the paper. The paragraph from lines 129-135 mentions that these grammars have the advantage that models cannot use shortcuts such as simple bracket counting for bounded Dyck languages, but it's not clear to me what's the relationship between the two. For instance, where exactly are these languages placed in the Chomsky hierarchy? As far as I know, bounded Dyck languages are regular languages, but the bounded depth PCFGs you consider seem finite.

2. This is perhaps related to my previous question, but could you comment on the fact that, in your experiments, transformers learn the underlying grammars with 99\% accuracy? There are many theoretical studies showing that transformers can only learn a subset of the regular languages. Do you think this would decrease, maybe as a function of grammar depth?

3. Did you test with transformers with more layers? You claim that 4 transformer layers are sufficient for a depth-4 grammar, and that the first 4 layers are used similarly to an Early chart while the final layer is used the aggregate the previous representations for next-token prediction. Would a 6-layer transformer show similar behavior?

**Limitations:**

Yes.

**Strengths And Weaknesses:**

Strenghts:

1. The paper correctly identifies a gap in the current understanding of how autoregressive transformers process hierarchical structure, and sheds some light into the mechanisms used to model depth-bounded PCFGs, showing similar behavior to that of classic parsing algorithms. This is highly relevant as modern LLMs, which are transformer-based, are increasingly being used for tasks that inherently have hierarchical structure, such as writing code.

2. The authors propose an interesting and original hypothesis about how transformers process hierarchical structure, framing it as a local-to-global processing mechanism. Their methods, based on classic parsing algorithms, are innovative and show some evidence for this hypothesis.


Weaknesses:
1. The experiments seem rather limited. In most experiments, the authors use 5 PCFGs only, all of depth 4, and train a single transformer model on each of them. Therefore, I am unsure about the generalizability of the results.

2. For a paper that is primarily empirical, there are surprisingly few details about the experimental design and data. For instance, how are the 5 PCFGs constructed? Even after going over appendix C.2. it is unclear to me how these grammars were chosen. Also, it is not clear why the authors focused only on depth-4 grammars. It would be interesting to see if/how the findings change for smaller or larger depths. On a similar note, I think there should be some more discussion in the main text about the contributions of the paper vs. prior work.

3. The paper often lacks clarity or omits important details. See my questions below for details.

---

> ### Author Rebuttal · Authors · 2026-03-31
>
> Thank you for your review.
>
> **Q1: Experiments limited -- only 5 PCFGs, all depth 4, single model each.**
>
> **A1:** CFGs are a simplified setting, but serve as a **fundamental benchmark** that captures the core challenge of natural language -- **nontrivial hierarchical structure and long-range dependence** -- while avoiding the confounding complexity of natural language token embeddings and semantics. These grammars still contain nontrivial structure: for instance, CFG1 alone generates $2^{36}$ unique sequences. The goal is to test whether the **same mechanism** appears across different grammars in a controlled setting, and empirically it does. The differences among the five grammars are discussed in A2.
>
> ---
>
> **Q2: How are the 5 PCFGs constructed? Why only depth 4? Choice deserves more discussion.**
>
> **A2:** We will clarify Appx C.2. Our setting is similar to Allen-Zhu and Li (2023): the chosen CFGs are **challenging** for a Transformer to learn, while still yielding clean hierarchical interpretability results. The five grammars are all bounded-depth layered PCFGs, but differ in **branching, rule templates, repetition, and induced ambiguity/length**. CFG1--5 progressively increase RHS repetition and rules per node, so **learning difficulty varies** even at fixed depth. We use bounded PCFGs so that the generated language is finite but extremely large. We chose **depth 4**: shallower grammars generate too few sequences, while deeper ones require larger models and much costlier probing. This balances difficulty and interpretability.
>
> We further added a new experiment in[Supplementary](https://anonymous.4open.science/r/cfg-transformer-56DD/rebuttal_results.pdf) (Figure 1, Tables 9--10), where we repeat the probing experiments of Section 3 on a grammar of depth 6 under a similar setup. Table 9 shows CYK-father and Earley-state accuracy **increase layer-wise**; Table 10 shows larger-window probes accurate only in deeper layers, matching depth-4 results. Depth-6 includes **ternary productions**, beyond our binary construction. This shared local-to-global pattern confirms the **mechanism is not an artifact of binary structure**.
>
> ---
>
> **Q3: More discussion of contributions vs prior work needed.**
>
> **A3:** We discuss prior work in the Introduction and Appendix A, and will make this clearer in the main text. The main distinction is that most prior work does not study **autoregressive generation**; even those that do, they do not provide the same **algorithmic/mechanistic account** aligned with a constructive theory. See our responses A5--A8 to Reviewer 2N2C for detailed comparisons with Allen-Zhu & Li, Cagnetta et al., Zhao et al., and Garnier-Brun et al.
>
> ---
>
> **Q4: Chomsky hierarchy placement -- bounded Dyck vs your grammars, both seem finite/regular.**
>
> **A4:** Yes: both bounded Dyck languages and our grammars are finite, and therefore **regular**. However, the key difference is not the Chomsky class but the **kind of state that must be tracked**. In prior work on bounded Dyck (Yao et al., 2021, Wen et al., 2023), depth is recognized by **bracket matching/counting**, reducing learning to tracking a scalar quantity. In contrast, our grammars cannot be parsed by such simple matching. To generate correctly, the model must track a **combinatorial symbolic state** -- the possible parent constituents in the latent tree. This is why we use bounded PCFGs: they remain controllable but require **genuinely hierarchical state tracking**.
>
> ---
>
> **Q5: 99% accuracy -- surprising given theoretical limits on Transformers for regular languages?**
>
> **A5:** 99% means the model learned to autoregressively generate **grammar-consistent sequences**. This is also reflected in Appx C.3 Table 1: the KL divergence shows the learned distribution is very close to the ground truth. We further added a new experiment in [Supplementary](https://anonymous.4open.science/r/cfg-transformer-56DD/rebuttal_results.pdf) Table 8: on a **grammar of depth 6**, we train 4-head Transformers with 5, 6, and 7 layers, which shows deeper models succeed while the while the **shallower model is worse** (97.67 vs. 99.46/99.69 accuracy, with higher KL). This is consistent with the fact that deeper grammars are more complex and suggests that **accuracy decrease with grammar depth** when model depth is insufficient.
>
> ---
>
> **Q6: Did you test with more layers? Would a 6-layer Transformer show similar behavior?**
>
> **A6:** We show additional results in [Supplementary](https://anonymous.4open.science/r/cfg-transformer-56DD/rebuttal_results.pdf) Tables 4, 5, and 6, where we train a **6-layer, 4-head** Transformer on CFG1 and repeat the probing experiments from Section 3. The results are very close to those of the 5-layer model, indicating that once the model has **sufficient expressive power**, its behavior is similar: the early layers recover **CYK-style completed constituents**, leaving enough information for the final layer to perform **Earley-style generation**.

---

> > ### Author Rebuttal · Reviewer_bGv3 · 2026-04-01
> >
> > I thank the authors for their response; they have addressed most of my comments. However, my main concern, namely the limited experiments, remains. I totally agree that, while being a simplification, bounded-depth CFGs are a suitable testbed for modeling hierarchy in natural language. I also agree that the binary structure is not a limitation. However, I am concerned that 5 is too small of a number to make general claims about bounded-depth CFGs in general. I am not convinced that these 5 grammars cover all the patterns that can arise in such a grammar. You mentioned that you discussed differences among these grammars in A2, yet I see no Appendix A2. I suppose you meant C2, but I also don't see any details there. The following comment brings some clarity, although not fully, and I think it should be included in the main text of the paper: "differ in branching, rule templates, repetition, and induced ambiguity/length. CFG1--5 progressively increase RHS repetition and rules per node, so learning difficulty varies". I think it wouldn't be difficult to generate *random* grammars somehow, and experiment with a larger number of grammars. That said, I understand it might not be computationally feasible to experiment with too many grammars, but some attempt should be made to ensure the results are generalizable enough.

---

> > > ### Author Response · Authors · 2026-04-06
> > >
> > > Thank you for the follow-up.
> > >
> > > **On the "A2" reference.** When we wrote "the differences among the five grammars are discussed in A2," we meant **our response A2 above** (the answer to **Q2** in this rebuttal), not an appendix section -- we apologize for the confusion. The grammar differences (branching, rule templates, repetition, induced ambiguity/length) are described there. We will also expand **Appendix C.2** with these details in the revision so they are accessible to readers of the paper directly.
> > >
> > > **On including grammar differences in the main text.** We agree this description should appear in the paper itself. We will add the following (or similar) to the main text: *"CFG1--5 are all bounded-depth layered PCFGs but differ in branching, rule templates, repetition, and induced ambiguity/length; from CFG1 to CFG5 we progressively increase RHS repetition and rules per node, so learning difficulty varies substantially even at fixed depth."*
> > >
> > > **On the number of grammars and generalizability.** We have now conducted experiments on **50 randomly generated depth-4 PCFGs** ([Supplementary](https://anonymous.4open.science/r/cfg-transformer-56DD/rebuttal_results_2.pdf)) to directly address this concern. The five grammars in the main paper, while structurally diverse, are hand-designed and cannot alone rule out the possibility that the observed mechanism is specific to curated choices. Random generation provides a principled coverage test: grammars are sampled from a well-defined distribution without any selection or filtering based on experimental outcome, offering an unbiased assessment of whether the local-to-global mechanism generalizes across the space of bounded-depth PCFGs.
> > >
> > > Each grammar is generated as follows: for each nonterminal, the number of production rules is drawn uniformly from $\\{2, 3, 4\\}$, the RHS length from $\\{2, 3\\}$, and each RHS symbol is drawn uniformly from the nonterminals of the next layer. We train a **4-head, 5-layer** Transformer on each grammar under the same training setup as the main paper, and run **CYK-father probing**, **Earley-state probing**, and **CYK-father window probing** at each layer, reporting averages across all 50 grammars.
> > >
> > > Across all 50 random grammars, we consistently observe:
> > >
> > > - **CYK-father and Earley-state probing accuracy increase monotonically layer by layer**, showing that hierarchical constituent information is progressively constructed across depth.
> > > - **Larger-window CYK-father probes succeed only in deeper layers**, confirming that longer-range structural dependencies are resolved later in the network.
> > > - **The CYK/Earley correspondence holds throughout**: early layers recover CYK-style completed constituents; the final layer encodes prefix-consistent Earley states.
> > >
> > > These findings **mirror the main paper's results precisely**. The consistency across 50 unselected random grammars confirms that the local-to-global mechanism is **not an artifact of the five hand-chosen grammars**, but a robust and general property of how autoregressive Transformers process bounded-depth hierarchical structure.

---

### Official Review · Reviewer_2N2C · 2026-03-11

**Soundness:** 3
**Presentation:** 3
**Significance:** 3
**Originality:** 3
**Overall Recommendation:** 5
**Confidence:** 5

**Summary:**

This paper uses hierarchical synthetic languages to study the mechanism by which deep transformer-based language models represent hierarchical structures. The data model considered is a layered and uniform-depth family of context-free grammars. The authors compare the representations of transformers trained on such data with two different parsing paradigms: Earley (sequential/left-to-right) and CYK (hierarchical/bottom-up) algorithms. The results indicate that transformers implement a combination of the two, using their layers to aggregate local features into the global information required for next-token prediction.

**Compliance With Llm Reviewing Policy:**

Affirmed.

**Final Justification:**

This contribution fits within a growing body of literature using formal grammars and other controlled synthetic languages to understand how LLMs capture the structure of language. The reviews prompted the authors to significantly extend the breadth of the empirical result, and to provide a better comparison with existing literature. As a result, I believe this work to advance our understanding of how transformer based LLMs represent hierarchical structures, with direct implications at least in the areas of learning theory and interpretability.

**Key Questions For Authors:**

I have no further questions.

**Limitations:**

The authors did not discuss the relationship with previous works in sufficient detail (see Strengths and Weaknesses section).

**Strengths And Weaknesses:**

# Soundness

The claims are clear and easy to identify. While the experiments support the claims, they do not provide definitive evidence.
- Concerning Fig. 3, the chosen probe is a two-layer MLP, which could be made arbitrarily expressive by increasing the hidden layer dimension. Therefore, with sufficient training data, Early states and CYK fathers could be extracted directly from the raw input data. Understanding how the transformer layers simplify the extraction would require additional experiments, comparing different probes trained with different amounts of resources.
- Concerning Fig. 5, it presents only qualitative evidence. Can it be made quantitative, e.g. by having a number that quantifies how local the effective attention windows are?

# Presentation

The manuscript is well-structured and easy to follow, and clearly written in general.

The point I struggle to understand is the relationship between bottom-up algorithms, such as CYK, and incremental algorithms, such as Earley parsing. They are initially presented as two alternative approaches (introduction, paragraph 4). Right after the introduction, the authors suggest that transformers implement a bottom-up algorithm ("this rules out naive sequential emulation ..." at the end of the 5th paragraph), which is later stressed in the *main contributions* paragraph ("unlikely to arise from sequentially emulating Earley"). However, on page 3, in the last paragraph of section 2.2, the authors claim that "transformers appear to combine the best of both". Which is it? Presumably, since both algorithms eventually produce the probability of the next token, they must ultimately agree with the transformer's prediction and with each other.

In addition, the *probing Earley states* paragraph on page 5 would benefit from a short description of what the Earley items represent (such as the sentence "an Earley item ..." at the beginning of page 14).

Finally, I believe that the current writing does not position the results properly in the context of prior literature (see section below for an extended discussion).

# Significance and Originality

This is the major weakness of this submission and the main reason for my mark. While the question is interesting and the results are mostly convincing (or could be made so with minor revisions), there is not enough discussion about the relationship with previous results on the same topic. Lacking a clear understanding of how the results differ from the following works, I cannot appreciate the significance and originality of this contribution.

- **Allen-Zhu & Li (2023)**: The authors correctly point out that this work only probes the final transformer layers, but it also uses linear probes. These are less expressive than those considered in the present study, hence providing stronger evidence of the representation's content. The authors also claim (in Appendix A) that Allen-Zhu & Li's result "suggests a multi-stage interpretation", which sounds analogous to the result proposed here, but what does this sentence mean exactly?

- **Cagnetta et al. (2024)**/**Cagnetta & Wyart (2024)** also probe the structure and information content of hidden representations, despite the author's stating that "it does not explicitly address the Transformer's internal mechanism" in Appendix A. "Scaling Laws and Representation Learning in Simple Hierarchical Languages:
Transformers vs. Convolutional Architectures" (https://journals.aps.org/pre/abstract/10.1103/qtd6-nl8p), belonging to the same line of works, presents further results on the dynamics of hidden representations across training.

- **Zhao et al. (2023)** present constructions of transformers capable of implementing the Inside-Outside algorithm either exactly or approximately. What is the relationship with the construction proposed here?

- **Garnier-Brun et al. (2024)** present the very same result displayed on Figure 5 of this manuscript.

Overall, it seems that all the results presented in this paper are already present in some form in the literature.

---

> ### Author Rebuttal · Authors · 2026-03-31
>
> Thank you for your review.
>
> **Q1: MLP probe could be arbitrarily expressive; Earley/CYK info could be extracted from raw input.**
>
> **A1:** The layer-wise progression and probe design are addressed in our response to Reviewer xtb8 (A2) -- the same reasoning applies here. One additional point: the probe architecture is **fixed in advance** (Appendix C); the probe is a **finite-parameter model**, not an arbitrarily expressive extractor.
>
> ---
>
> **Q2: Fig 5 is only qualitative -- can you quantify?**
>
> **A2:** Fig. 5 already shows a quantitative pattern: attention in layers 1--4 concentrates within windows of size 2, 4, 8, 16, aligning with our theoretical construction. We further quantify this in [Supplementary](https://anonymous.4open.science/r/cfg-transformer-56DD/rebuttal_results.pdf) Table 3 by measuring **attention mass inside theory-predicted windows**: **67% (Layer 1), 81% (Layer 2), 92% (Layer 3), and 100% (Layer 4)** -- a substantial and **monotonically increasing** fraction that strongly supports our interpretation.
>
> ---
>
> **Q3: Relationship between CYK and Earley -- paper seems contradictory ("bottom-up" vs "combines the best of both").**
>
> **A3:** In short: **CYK handles bottom-up constituent building (early layers); Earley handles autoregressive generation (final layer)** -- this is what we mean by "combines the best of both".
>
> CYK can be **parallelized** but not made autoregressive; Earley is **autoregressive** but sequential. Our results support a **hybrid picture**: early layers recover **CYK-style completed constituents**, while the final layer carries out **Earley-style Scan and Predict**, avoiding explicit sequential computation.
>
> ---
>
> **Q4: Earley items need brief description in main text.**
>
> **A4:** An Earley state $(i, A \to \alpha \bullet \beta, \omega)$ means rule $A \to \alpha\beta$ **started at position** $i$, matched up to $\alpha$, with **probability** $\omega$. Full Earley introduction is in Appendix B; we will add a brief inline explanation in the revision.
>
> ---
>
> **Q5: Allen-Zhu & Li comparison; linear probes give stronger evidence; what does "multi-stage" mean?**
>
> **A5:** Allen-Zhu and Li (2023) probe only the **final layer** and assume the **global derivation tree is known**, which does not align with prefix-based generation. We probe **every layer at every position** with a harder **multi-label** target; for linear vs. MLP, see [Supplementary](https://anonymous.4open.science/r/cfg-transformer-56DD/rebuttal_results.pdf) Tables 1--2 and Reviewer xtb8 (A2).
>
> Regarding "multi-stage": they show **dynamic-programming-like behavior** but do not identify **which specific algorithm**. We identify the stages concretely: **CYK-style bottom-up parsing** followed by **Earley-style prefix rollout** in the final layer, supported by probing and theory.
>
> ---
>
> **Q6: Cagnetta et al. (2024) / Cagnetta & Wyart (2024) also probe hidden representations; new PRE paper [4] studies dynamics of hidden representations across training.**
>
> **A6:** We will cite [4]. These works are **complementary**: they study **how representations emerge during training** (scaling laws, dynamics), while we characterize **what algorithm the trained model has learned**. They differ in three ways: (1) all three use only **last-token prediction**, vs. our **full autoregressive generation** at every prefix; (2) their focus is the **data-side** (learning curves vs. hierarchy), ours is the **model-side** (matching CYK/Earley via layer-wise probing); (3) we provide an explicit **Transformer construction** provably implementing the local-to-global mechanism (Fig. 5, Appendix Fig. 12), with no counterpart in their work.
>
> ---
>
> **Q7: Zhao et al. (2023) construct Transformers for Inside-Outside. What is the relationship?**
>
> **A7:** Zhao et al. (2023) study **masked/bidirectional** settings and their construction requires **linear depth** $O(L)$. Our construction addresses the **causal setting** and requires $O(\log L)$ **depth** -- a 5-layer Transformer suffices for sequences up to length 72. Our model must produce an **Earley-style prefix summary** for valid autoregressive generation, rather than relying on full-string information. The $O(\log L)$ vs. $O(L)$ gap reflects this difference between causal and bidirectional settings.
>
> ---
>
> **Q8: Garnier-Brun et al. (2024) present the very same result as Figure 5.**
>
> **A8:** Garnier-Brun et al. (2024) use **masked language modeling**, so a similar tree-like attention pattern in an autoregressive model is **not obvious**. Beyond this observation, our paper: (1) shows how completed constituents enable **autoregressive generation** via Earley-style rollout, and (2) provides a **theoretical construction** provably implementing this with matching attention patterns (Fig. 5, Appendix Fig. 12). Neither is addressed in their work.
>
> ---
>
> **References:**
>
> [4] Cagnetta et al. (2025). Scaling laws and representation learning in simple hierarchical languages: Transformers versus convolutional architectures.

---

> > ### Author Rebuttal · Reviewer_2N2C · 2026-04-03
> >
> > Most of my concerns have been addressed in this reply (including parts from the reply to xtb8). I believe that the paper could be accepted, provided the authors:
> > * Include a discussion of linear probes and the answer A2 to reviewer xtb8 in the revised manuscript, including the table comparing linear and MLP probes. I don't think linear probes do so badly, while being much simpler than MLP probes, so I think it's worth including those results.
> > * Report answer A3 in the main manuscript to clarify the roles of CYK-like and Early-style algorithms;
> > * Include a detailed comparison with previous literature as in answers A5 and A6;
> >
> > The two following concerns remain, and they *must* be addressed for a fair comparison with the literature:
> >   * Zhao et al. **do require linear depth, but they don't assume a Balanced Full-Binary PCFG with a fixed number of layers**---a *huge* restriction from the set of generic PCFGs. While I don't believe the huge restriction to be an issue, it should be addressed, especially because it would have been impossible to find such an economical construction without the restriction.
> >   * Garnier-Brun et al. **do provide a theoretical construction based on Belief Propagation** (paragraph "Exact transformer embedding of BP." of section 4), and it is kind of obvious, or at least intuitive, that a similar tree-like attention pattern should emerge in autoregressive models.

---

> > > ### Author Response · Authors · 2026-04-06
> > >
> > > Thank you for the follow-up and for moving toward acceptance. We confirm the revision plan and address the two remaining concerns.
> > >
> > > **On the three revision requests.** We will incorporate all of these: **(1)** the linear vs. MLP probe comparison ([Supplementary](https://anonymous.4open.science/r/cfg-transformer-56DD/rebuttal_results.pdf) Tables 1--2) and the reasoning from xtb8 A2 in the main text, with the comparison table; **(2)** the clarification of the respective roles of CYK-style and Earley-style computation (A3) added inline; and **(3)** detailed comparisons with Allen-Zhu & Li, Cagnetta et al., Zhao et al., and Garnier-Brun et al. (A5--A6) in a dedicated related-work section.
> > >
> > > **On Zhao et al. and the balanced full-binary PCFG restriction.**
> > >
> > > We acknowledge this restriction and agree it should be stated more explicitly — the $O(\log L)$ depth bound benefits from the dyadic structure of balanced full-binary PCFGs. We will be explicit that removing this restriction is an open problem.
> > >
> > > The more fundamental distinction is the **task setting**: Zhao et al. (2023) work in the **masked language modeling (MLM)** setting, where the model has access to the **complete string** at every position. Our construction addresses **autoregressive generation**, where the model sees only a prefix and must produce the correct next-token distribution at each step. These are fundamentally different problems: in MLM, each position can attend bidirectionally to all others; in autoregressive generation, only past tokens are available, and the representation must encode a **sufficient statistic for all valid continuations** — precisely the Earley-style prefix summary our construction implements. The $O(\log L)$ vs. $O(L)$ gap therefore reflects a result for an entirely different task, not merely an efficiency improvement. To our knowledge, prior constructive results for Transformers on CFG languages have largely focused on non-autoregressive settings or on specific languages such as Dyck with specialized mechanisms (e.g., shortcut-based constructions); our construction is the first to implement Earley-style prefix rollout for PCFGs in a causal Transformer with $O(\log L)$ depth, even under the balanced full-binary restriction. The structural restriction we impose is what makes an $O(\log L)$-depth causal construction tractable; extending this to general PCFGs is a natural open problem.
> > >
> > > **On Garnier-Brun et al. and the Belief Propagation construction.**
> > >
> > > We agree the intuition is natural: if tree-like attention emerges in bidirectional models via BP, one might expect a similar pattern in autoregressive models. What our construction adds is a formal account of how this works in the autoregressive setting. While the intuition carries over, the technical approach is different: the causal mask prevents attending to future tokens, so BP-style inference over the complete string does not directly apply. Correct next-token prediction requires tracking **active Earley states** — which constituents are consistent with the current prefix — a computation that is structurally distinct from full-string BP. Our $O(\log L)$-depth causal Transformer makes this precise and, to our knowledge, provides a constructive result with no direct counterpart in prior work. We will incorporate this discussion and clarify the positioning of our work in the revision.

---

### Official Review · Reviewer_xtb8 · 2026-03-13

**Soundness:** 2
**Presentation:** 3
**Significance:** 2
**Originality:** 3
**Overall Recommendation:** 3
**Confidence:** 2

**Summary:**

In the paper authors study how autoregressive transformers represent hierarchical structure. Thy investigate it in a controlled formal language setting. They first train a small decoder only transformer on a specific CFGs. Then show that models closely match the grammar's next token distributions. Later use probing to argue that later layers encoder prefix states while intermediate layers build CY style constituent information in local to global fashion.

**Compliance With Llm Reviewing Policy:**

Affirmed.

**Final Justification:**

The paper studies an interesting mechanistic question in a clean and well controlled setting, and I appreciate the combination of empirical analysis and explicit construction. The local-to-global story is coherent and reasonably original, and the rebuttal was helpful in clarifying claims and adding some stronger supporting evidence, especially the causal masking result.

However, my overall recommendation remains unchanged. The rebuttal improves the paper, but it does not fully resolve my main concerns. I still view the empirical evidence as suggestive rather than definitive on causal necessity, and I still see the theory mainly as an expressivity/existence result in a restricted setting rather than an explanation of why standard training should recover this mechanism. More broadly, I recognize that there is related literature making representation-level claims of this kind, sometimes strengthened by comparisons showing that Transformers encode the relevant structure better than other architectures. My concern is partly with that broader style of evidence as well: showing that a flexible neural network can represent a desired computation, even with supportive probing and intervention results, is not always enough for a compelling account of the learned mechanism. I mention this to clarify that my reservation is partly about the standards I apply to this line of work more generally, rather than only to this paper.

Overall, I think the paper has clear strengths in originality, clarity, and careful analysis, but the remaining concerns about soundness and broader significance keep me at my original score.

**Key Questions For Authors:**

1) Can you provide a causal test of the proposed mechanisms?

2) In addition to the representation, can you provide learning theory for the proposed mechanism?

**Limitations:**

yes

**Strengths And Weaknesses:**

Strengths:
1) the question they focus is clear and and interesting. they ask a mechanistic question: "how hierarchical information is represented across layers during autoregressive generation and chooses a setting where this can be studies cleanly.

2) The empirical story is coherent that is they show that the models closely match the grammar's next token distribution and span restricted probing results. Supporting the main claim that represantations become more global in deeper layers.

3) They provide a theorem for a simplified grammar family give a concrete description of how hierarchical computation could be distributed across layers.

Weaknesses:
1) The empirical scope is toy. The authors acknowledges that transfer to unrestricted natural language is uncertain. Their approach is a reasonable starting point but it limits how strongly one can interpret the claims as explaining the hierarchical computation in transformers.

2) The mechanistic claims rely heavily on probing which only shows recoverability. It does not show any causal use. In addition using two layer probes is problematic. I also think when anyone uses a probe they should always consider and discuss about the probe capacity and how much data is used to train the probe for a consistent result.

3) Some of the conclusions are stated too strongly compared to the presented evidence, for instance the phrases like "strong evidence" make the story read as more settled that it is.

4) The theory is primarily an existence/representation result, not a learning result. It proves that a Transformer can realize the proposed parsing mechanism for a restricted grammar family, but does not explain why standard training should recover this mechanism or under what conditions it emerges in practice.

---

> ### Author Rebuttal · Authors · 2026-03-31
>
> Thank you for your review.
>
> **Q1: Empirical scope is toy.**
>
> **A1:** CFGs serve as a **fundamental benchmark** capturing the core challenge of natural language -- **nontrivial hierarchical structure and long-range dependence** -- while avoiding the confounding complexity of token embeddings and semantics. This enables **rigorous mechanistic analysis** with reliable ground truth, currently infeasible for unrestricted natural language. Our goal is to analyze a rich formal setting where precise mechanistic claims are possible.
>
> ---
>
> **Q2: Probing only shows recoverability, not causal use; two-layer probes are problematic; probe capacity/data discussion needed.**
>
> **A2:** If the probed information were extractable from raw input, **probe accuracy would not vary across layers**. Instead, it increases monotonically from Layer 1 to Layer 4, and span-restricted probing confirms **short-span constituents emerge before long-span ones** (Fig. 3). This layer-wise progression shows the Transformer is *constructing* this information. We address **causality** in A5 below.
>
> We present additional experiments in the [Supplementary](https://anonymous.4open.science/r/cfg-transformer-56DD/rebuttal_results.pdf) (Tables 1--2) showing that **linear probes are weaker**, neural probes are stable on both train and test. Since our task is **multi-label** classification, we use the neural probe to ensure sufficient expressivity. This choice is consistent with prior work: [1] and [2] use MLP probes, and [3] notes that not all features exhibit simple linear structure. At each layer, the **probe's train loss and test loss are nearly identical**, indicating probes generalize well rather than simply overfitting. Full setup is in Appendix C.2.
>
> ---
>
> **Q3: Conclusions stated too strongly ("strong evidence").**
>
> **A3:** We will soften the wording where appropriate. That said, the evidence is strong when taken together. The probing results show a **significant and nontrivial correspondence** between learned representations and both CYK-style and Earley-style computations. **Attention patterns** (Fig. 5, Appendix Fig. 12) closely match our **theoretical local-to-global construction**, with progressively expanding windows and peaks aligned with predicted dyadic boundaries. A new **causal intervention** ([Supplementary](https://anonymous.4open.science/r/cfg-transformer-56DD/rebuttal_results.pdf) Table 7) shows the largest accuracy drop at exactly the offsets our construction specifies. These **converging lines of evidence** -- probing, attention analysis, theory-experiment alignment, and causal intervention -- support a concrete mechanistic interpretation rather than a loose analogy.
>
> ---
>
> **Q4: Theory is an existence result, not a learning result.**
>
> **A4:** This work focuses on **mechanistic interpretation**, so training dynamics is outside our scope. The training setup is entirely **standard** (cross-entropy loss, AdamW optimizer), with full details in Appendix C. The constructed Transformer uses **finite width and finitely many parameters**, so standard tools such as **VC/pseudo-dimension** bounds can in principle be applied to study generalization, and one can further ask how sample complexity scales with grammar depth. The deeper question of why **gradient-based optimization** recovers parsing-like mechanisms remains largely open; we leave this to future work.
>
> ---
>
> **Q5: Can you provide a causal test of the proposed mechanisms?**
>
> **A5:** We provide two complementary forms of evidence.
>
> **Causal intervention** ([Supplementary](https://anonymous.4open.science/r/cfg-transformer-56DD/rebuttal_results.pdf) Table 7): Our theory predicts that layer $k$ attends to offset $2^{k-1}$ (positions with $t \bmod 2^k = 0$), giving offsets 1, 2, 4, 8 for layers 1--4. We mask each separately and evaluate accuracy. The **largest drop occurs exactly at the theory-predicted offset** for layers 1--3, producing a clear **diagonal pattern** matching our construction. Diminishing magnitude in deeper layers is expected: each aggregates over **exponentially longer spans**, so masking a single offset has a naturally diluted impact.
>
> **Theory-experiment alignment:** Attention patterns (Fig. 5, Appendix Fig. 12) match our theoretical construction closely. **Attention windows expand progressively** across layers exactly as the theory predicts, and **peaks align with the predicted dyadic boundaries**. The model has organized its computation to mirror the constructed mechanism.
>
> Taken together, the interventional and structural evidence provide **converging support** for the proposed local-to-global mechanism.
>
> ---
>
> **References:**
>
> [1] Hewitt and Liang (2019). Designing and Interpreting Probes with Control Tasks.
>
> [2] Tenney et al. (2019). What do you learn from context? Probing for sentence structure in contextualized word representations.
>
> [3] Engels et al. (2025). Not All Language Model Features Are One-Dimensionally Linear.

---

> > ### Author Rebuttal · Reviewer_xtb8 · 2026-04-03
> >
> > The controlled CFG setting is well motivated for mechanistic analysis, but the empirical scope remains limited, so the broader implications for natural language Transformers are uncertain. The rebuttal strengthens the probing results and adds a targeted intervention, which provides some causal support for the proposed mechanism. However, I still find the causal evidence somewhat limited, and probing remains more diagnostic of representational structure than definitive proof of computational necessity. Finally, the theory is primarily a constructive expressivity result rather than a theory of learning, so it does not explain why standard training should recover this mechanism in practice.

---

> > > ### Author Response · Authors · 2026-04-06
> > >
> > > Thank you for the follow-up. We address your two remaining concerns.
> > >
> > > **On causal evidence and computational necessity.**
> > >
> > > We agree that probing alone does not establish computational necessity, which is why we complement it with evidence from other angles. Our causal claim rests on the **attention masking experiment** ([Supplementary](https://anonymous.4open.science/r/cfg-transformer-56DD/rebuttal_results.pdf) Table 7): masking the theory-predicted offset at layer $k$ causes an accuracy drop at exactly that layer, and the diagonal pattern across layers matches our construction. This goes beyond recoverability — it shows that **removing the predicted circuit component degrades task performance in exactly the way the theory anticipates**.
> > >
> > > We note that "computational necessity" is a strong bar rarely achieved in mechanistic interpretability even for natural language models, where ground truth is unavailable. Our position is more modest: we provide **converging multi-method evidence** — layer-wise probing progression, attention window quantification (67% → 81% → 92% → 100% mass in theory-predicted windows), and targeted causal interventions — that collectively support the proposed mechanism more strongly than any single method alone. We will clarify this framing in the revision so that the strength of the claim matches the strength of the evidence.
> > >
> > > **On the theory as an expressivity result.**
> > >
> > > We agree that our construction shows a Transformer *can* implement the local-to-global parsing mechanism, but does not prove that gradient-based optimization *will* recover it. A theoretical characterization of the training dynamics is out of the scope of the current paper (which focuses on mechanistic interpretation of trained transformers), and we leave it as an important question for future work.
> > >
> > > However, we would like to emphasize that in the **balanced full-binary setting** — the exact setting our theory covers — the experiments corroborate the construction in close detail. The attention patterns (Fig. 5, Appendix Fig. 12) match the theoretical local-to-global structure almost exactly: attention windows expand as 2, 4, 8, 16 across layers 1--4, with peaks aligned at the dyadic boundaries the construction predicts. The causal masking experiment ([Supplementary](https://anonymous.4open.science/r/cfg-transformer-56DD/rebuttal_results.pdf) Table 7) further confirms this: the most disruptive offset at each layer falls on precisely the position our construction assigns to that layer, producing a diagonal pattern that a post-hoc theory would not predict. In this setting, the theory is not merely an existence result — it is a **quantitatively accurate description** of what the trained model actually learns.
> > >
> > > Beyond the balanced full-binary case, the same qualitative pattern persists across five structurally different grammars and a depth-6 extension with ternary productions, suggesting the mechanism generalizes beyond the specific setting our theory analyzes. We believe these theoretical and empirical results shed light on how to analyze the training dynamics in future work.

---

### Decision · Program_Chairs · 2026-04-30

**Decision:**

Accept (regular)

**Comment:**

The paper is technically solid and provides some insight in LLM encoding of language.  Contribution is mostly at the theoretical level.  Reviewers have criticised the paper for lack of further definitive empirical evidence that the LLM behaves to theoretical capacity; reframing the theoretical insights in that light is advised.